



# A new process-based and scale-respecting desert dust emission scheme for global climate models – Part I: description and evaluation against inverse modeling emissions

Danny M. Leung[1], Jasper F. Kok[1], Longlei Li[2], Gregory S. Okin[3], Catherine Prigent[4], Martina Klose[5], Carlos Pérez García-Pando[6,7], Laurent Menut[8], Natalie M. Mahowald[2], David M. Lawrence[9], and Marcelo Chamecki[1]

[1]Department of Atmospheric and Oceanic Sciences, University of California – Los Angeles, Los Angeles, California, USA
[2]Department of Earth and Atmospheric Sciences, Cornell University, Ithica, New York, USA
[3]Department of Geography, University of California – Los Angeles, Los Angeles, California, USA
[4]Sorbonne Université, Observatoire de Paris, Université PSL, CNRS, LERMA, Paris, France
[5]Institute of Meteorology and Climate Research (IMK-TRO), Department Troposphere Research, Karlsruhe Institute of Technology (KIT), Karlsruhe, Germany
[6]Barcelona Supercomputing Center (BSC), Barcelona, Spain
[7]Catalan Institution for Research and Advanced Studies (ICREA), Barcelona, Spain
[8]Laboratoire de Météorologie Dynamique, École Polytechnique, Institut Polytechnique de Paris, Ecole Normale Supérieure, Sorbonne Université, CNRS, Palaiseau, France
[9]Climate and Global Dynamics Laboratory, National Center for Atmospheric Research, Boulder, Colorado, USA

*Correspondence to*: Danny M. Leung (dannymleung@ucla.edu)

## Abstract

Desert dust accounts for most of the atmosphere's aerosol burden by mass and produces numerous important impacts on the Earth system. However, current global climate models (GCMs) and land surface models (LSMs) struggle to accurately represent key dust emission processes, in part because of inadequate representations of soil particle sizes that affect the dust emission threshold, surface roughness elements that absorb wind momentum, and boundary-layer characteristics that control wind fluctuations. Furthermore, because dust emission is driven by small-scale (~1 km or smaller) processes, simulating the global cycle of desert dust in GCMs with coarse horizontal resolutions (~100 km) presents a fundamental challenge. This representation problem is exacerbated by dust emission fluxes scaling nonlinearly with wind speed above a threshold wind speed that is sensitive to land surface characteristics. Here, we address these fundamental problems underlying the simulation of dust emissions in GCMs and LSMs by developing improved descriptions of (1) the effect of soil texture on the dust emission threshold, (2) the effects of non-erodible roughness elements (both rocks and green vegetation) on the surface wind stress, and (3) the effects of boundary-layer turbulence on driving intermittent dust emissions. We then use the resulting revised dust emission parameterization to simulate global dust emissions in a standalone model forced by reanalysis meteorology and land surface fields. We further propose (4) a simple methodology to scale up high-resolution gridded dust emissions to the coarse resolution of GCMs. The resulting dust emission simulation shows substantially improved agreement against regional dust emissions observationally constrained by inverse modeling. We thus find that our revised dust emission parameterization can substantially improve dust emission simulations in GCMs and LSMs.



## 1. Introduction

Desert dust accounts for more than half of the atmospheric mass loading of particulate matter (PM) (Kinne et al., 2006; Kok et al., 2017) and produces a wide range of important impacts on multiple components of the Earth system (Shao et al., 2011a). For instance, dust changes Earth's radiative budget and atmospheric dynamics directly by scattering and absorbing radiation (Sokolik and Toon, 1996; Miller and Tegen, 1998) and indirectly by mediating cloud formation (Rosenfeld et al., 2001; Shi and Liu, 2019; McGraw et al., 2020). Dust also impacts biogeochemistry by delivering nutrients such as iron and phosphorus to ocean and land ecosystems (Mahowald et al., 2010; Hamilton et al., 2020).


In recent decades, modelers have made substantial progress in developing various parameterizations for the main dust cycle processes including emission (e.g., Shao et al., 1993; Marticorena and Bergametti, 1995; Marticorena et al., 1997; Tegen and Fung, 1995; Klose and Shao, 2013), advection (e.g., Prospero, 1999; Lin, 2004; Van der Does et al., 2018), deposition (e.g., Barth et al., 2000; Liu et al., 2001; Zhang et al., 2001; Petroff and Zhang, 2010), as well as its biogeochemical effects (e.g., Jickells et al., 2005; Mahowald et al., 2005, 2010; Hamilton et al., 2020), optics (e.g., Sokolik and Golitsyn, 1993; Linke et al., 2006; Adebiyi et al., 2020), and radiative effects (e.g., Di Biagio et al., 2020; Li et al., 2021). Despite substantial progress in dust modeling, current global climate models (GCMs) and Earth System Models (ESMs) still struggle to adequately simulate the dust cycle, which impedes an accurate assessment of dust impacts (Huneeus et al., 2011; Wu et al., 2020; Zhao et al., 2022). For instance, model simulations still show large discrepancies when compared against observations of the spatial and temporal characteristics of the dust cycle, including dust emission (Kok et al., 2014a; Pierre et al., 2014a), dust PM concentration (Wu et al., 2019; Pu et al., 2020; Li et al., 2022), dust aerosol optical depth (DAOD or DOD) (Ridley et al., 2012; Kok et al., 2014b; Pu and Ginoux, 2018; Parajuli et al., 2019), dust deposition (Ginoux et al., 2001; Albani et al., 2014; Kok et al., 2014b; Li et al., 2022), and dust size distributions (Parajuli et al., 2019; Adebiyi and Kok, 2020; Li et al., 2022). Also, models struggle to capture the observed interannual and decadal variability of dust (Ridley et al., 2014; Smith et al., 2017; Evan, 2018; Kok et al., 2018) as well as the sensitivity of dust to climate changes (Evan, 2018; Kok et al., 2018). An improved quantification of dust impacts on the Earth system thus requires improvements in how dust is simulated in models.


One key piece of physics that models struggle to parameterize is the dust emission threshold. The dust emission threshold $u_{*t}$ is defined as the threshold wind stress/speed above which winds initiate, or below which winds cease, the lifting of sand particles whose impacts on the soil surface emit dust aerosols (Kok et al., 2012; Comola et al., 2019b). The dust emission threshold is a function of soil properties like particle size distribution, air density, and soil moisture. There are various reasons for the inadequate parameterization of the dust emission threshold. First, many models assume a globally constant soil particle size in calculating a spatially varying dust emission threshold (Zender et al., 2003a; Darmenova et al., 2009; Kok et al., 2014b), whereas the actual soil particle size is likely a function of space and time and could depend on soil properties (Webb et al., 2016) . Some models proposed that soil particle sizes are related to the soil texture and therefore represent the soil particle size as a global map (Tegen et al., 2002; Darmenova et al., 2009; Menut et al., 2013; Klose et al., 2021), but these maps either have not yet been thoroughly validated against observations or are based upon extrapolation of a limited amount of observations. Second, most current models use the fluid threshold (also named static threshold or initiation threshold) above which saltation is *initiated* as the dust emission threshold, but it is well known that dust emission is governed by both the larger fluid threshold and the smaller impact threshold (also named dynamic threshold or cessation threshold) below which saltation is *terminated* (Bagnold, 1941; Shao, 2008;



Martin and Kok, 2018; Comola et al., 2019a, b; Pähtz et al., 2020). Moreover, dust emission is a nonlinear process (i.e., it varies with the wind speed to the second to fifth power per Kok et al., 2014a) and the emission flux is particularly sensitive to the magnitude of the emission threshold (Kawai et al., 2021). Thus, land surface models (LSMs) within GCMs and ESMs need to parameterize the emission threshold

correctly to get an adequate spatiotemporal variability of the modeled atmospheric dust.

The second key dust emission physics that LSMs struggle to represent is the partitioning of the wind stress. Wind drag is partitioned into the part absorbed by surface roughness elements (mainly rocks and plants) and the part exerted on the bare soil that drives dust emissions. This drag partitioning effect is

modeled by several dynamical schemes (Raupach et al., 1993; Marticorena and Bergametti, 1995; Okin, 2008), and is accounted for in some models (LeGrand et al., 2019; Klose et al., 2021; Tai et al., 2021) but not others (e.g., Kok et al., 2014b; Evans et al., 2016). One major challenge in modeling drag partition is to quantify the amount of rocks (which includes rocks, pebbles, and gravel in this study) and their corresponding partition effect, because there are few measurements of rock roughness. To cope with this

issue, studies have used in-situ and/or remote sensing scatterometer measurements to quantify the small-scale land surface roughness (e.g., Greeley et al., 1997; Roujean et al., 1997; Marticorena et al., 2004; Prigent et al., 2005, 2012), especially over arid desert regions over which rocks, pebbles, and gravel dominate the roughness. However, with a few recent notable exceptions that attempted to represent both the roughness effects of rocks and vegetation (e.g., Darmenova et al., 2009; Foroutan et al., 2017; Klose

et al., 2021), studies often omitted the drag partition effect either due to vegetation (e.g., Menut et al., 2013) or due to rocks (e.g., Wu et al., 2016; LeGrand et al., 2019; Tai et al., 2021). To resolve these issues, we propose a new approach that combines the drag partition effects of both elements, leveraging satellite scatterometer measurements to quantify the surface rock roughness, and using observable vegetation and land surface variables to quantify the surface vegetation roughness.


The third key piece of fundamental dust emission physics not accounted for by many models is the effect of boundary-layer turbulence on dust emissions. Most current GCMs assume a constant wind speed (and thus a constant emission flux) within the relatively large model time step, e.g., 30 minutes (e.g., Rahimi et al., 2019; Dunne et al., 2020). However, in reality, high-frequency turbulent fluctuations cause

the wind speed to fluctuate within a time step. Because dust emissions scale nonlinearly with wind speed, this causes highly uneven and fluctuating dust emission fluxes (Durán et al., 2011). Even more importantly, wind turbulent fluctuations can sweep across the dust emission threshold multiple times and shut off dust emissions intermittently within one model time step, resulting in strong dust emission intermittency (Comola et al., 2019b). Therefore, omitting turbulence will either overestimate or underestimate dust

emissions, especially over marginal source regions where winds fluctuate around the high emission threshold, as models do not account for the cessations or initiations of dust emissions due to turbulent fluctuations. To account for the instantaneous wind fluctuations, one approach is to derive a probability density function (PDF) for the instantaneous momentum flux using large-eddy simulations (LES), which is then used for quantifying instantaneous dust emission fluctuations (Klose and Shao, 2012; Klose et al.,

2014). Another approach is to use the Monin-Obukhov similarity theory (MOST) to relate the standard deviation of the instantaneous wind to the boundary-layer dynamical variables (Comola et al., 2019b). In this study, we will account for turbulent dust emissions by following Comola et al. (2019b), which showed significant improvements in representing the small-magnitude saltation and dust fluxes that are particularly important over marginal source regions.


In addition to these issues of models missing some of the fundamental physics of dust emission, a central issue in modeling the global dust cycle is that dust emissions are grid resolution-dependent because of the nonlinear dependence of dust emissions to meteorological fields and land-surface variables. Since dust emission varies nonlinearly with wind speed and has an even more complex relation to the soil



moisture (Gillette and Passi, 1988; Fécan et al., 1999; Shao, 2001; Kok et al., 2014a), the total regional and global emissions can vary significantly with grid resolution (Ridley et al., 2013; Meng et al., 2021). For instance, modeled emissions were found to increase by ~29 % from a 1° × 1° to 0.25° × 0.25° resolution (Ridley et al., 2013; Feng et al., 2022). As a consequence, GCMs and ESMs often need to tune emissions separately for different grid resolutions to match observational dust budgets (Ginoux et al., 2001;

Zender et al., 2003a; Albani et al., 2014; Kok et al., 2014b; Chappell et al., 2021). This issue occurs because current GCM grid sizes of ~1° or 100 km cannot resolve the spatial scales of ~1 m to ~1 km over which soil properties and wind speeds change (Ridley et al., 2013). When adopting coarse grid resolutions, coarser modeled meteorological fields (for GCMs) or spatially averaged input meteorological fields (for chemical transport models or CTMs) will smooth out the local wind extrema, possibly causing wind

speeds to fall below the dust emission threshold. As a result, the coarse modeled winds usually result in strong GCM emissions underestimations (Ridley et al., 2013). The same smoothing problem also occurs for soil moisture for instance, with its maxima smoothed out leading to an overestimation of dust emissions. Thus, although some GCMs and ESMs recently implemented more physical schemes (Zhao et al., 2022), their inability to resolve the small scales still causes challenges for capturing the accurate spatial

distributions of dust emissions (Meng et al., 2021). For the same reason GCMs tend to neglect small-scale emissions over marginal source regions. In this study, we will analyze the scale-dependence of our dust emission scheme given specific input datasets and propose a method of upscaling the coarse dust emissions to alleviate the scale-dependence problem.

165        To tackle the above problems and improve simulations of the global dust cycle, we propose a new emission scheme for global models that includes key dust emission physics missing from current models. Specifically, we (1) account for the effects of the soil particle size distribution (PSD) on the dust emission threshold, (2) draw on satellite data and physically explicit models to account for wind momentum absorption by both rocks and vegetation, and (3) account for turbulence-driven intermittency in dust

emission fluxes. After we review current dust emission schemes in Sect. 2, we present our new scheme in Sect. 3. In Sect. 4, we code the new dust emission scheme as a standalone sandbox model (see Sect. 2.4) in R, a programming language for data analysis and computation, and examine the resulting spatiotemporal variability of the new dust emissions. We then examine the grid scale-dependence of dust emissions and derive a correction map for coarser dust emission simulations to correct their spatial

variability to high-resolution simulations. Sect. 5 discusses and summarizes the main findings of this paper. In our companion paper (Leung et al., in prep.), we will implement this new scheme in the state-of-the-art Community Earth System Model version 2 (CESM2) and evaluates its performance against observations.

## 2. Current dust emission schemes in climate models


        This section provides a basic review of current dust emission schemes. Broadly speaking, these schemes consist of a parameterization of the dust emission threshold (Sect. 2.1), a parameterizatation of the reduction of the wind drag on the bare soil surface due to momentum absorption by surface roughness elements (Sect. 2.2), and a parameterization of dust emission flux given wind speed, threshold wind speed,

and drag absorption (Sect. 2.3). We will develop an improved emission scheme for GCMs in Sect. 3 by improving upon each of these three core ingredients of dust emission schemes. We also provide a brief description of meteorological data used for computing our dust emission scheme in Sect. 2.4.

### 2.1. Parameterization of the dust emission thresholds

190        There have been extensive studies on the dust emission threshold $u_{*t}$, defined as the wind speed or drag that corresponds to the initiation or cessation of dust emission. Dust emission is caused by saltation, a process by which sand particles (geometric diameter > 63 $\mu$m) on the surface are lifted by wind drag into the airstream (known as aerodynamic entrainment) and undergo ballistic trajectories (Anderson, 1989;



Kok et al., 2012). Saltated sand particles subsequently impact on the granular bed (known as saltation
bombardment), and the kinetic energy incident on the granular bed may eject other particles and cause
further saltation, or may break the cohesive bonds between fine soil particles and either the saltating
particle or the granular bed, causing the emission of dust aerosols (Shao, 2008). The minimum wind
friction velocity $u_*$ required for initiating saltation through aerodynamic entrainment is called the fluid
threshold $u_{*ft}$ (McKenna Neuman and Sanderson, 2008). Once saltation is initiated, a smaller $u_*$ is
needed to maintain saltation because saltation bombardment can create further saltation, which is more
efficient than creating saltation solely through the wind drag. Saltation will be maintained at a slightly
smaller $u_*$ called the impact threshold $u_{*it}$ (Bagnold, 1941; Martin and Kok, 2018). The ratio $u_{*it}/u_{*ft}$,
is about 0.8–0.85 for loose dry sand and less for soils with other sources of cohesion (e.g., moisture,
organic matter), because cohesion rapidly increases $u_{*ft}$ but low-to-moderate levels of cohesion do not
increase $u_{*it}$ as indicated by numerical simulations (Comola et al., 2019a; Ralaiarisoa et al., 2022) . It
follows that dust emission can occur below $u_{*ft}$, especially in marginal dust source regions with high soil
moisture for which $u_{*it}$ can be much smaller than $u_{*ft}$. $u_{*t}$ is thus a general concept comprised of both
$u_{*ft}$ and $u_{*it}$. However, $u_{*it}$ is not currently accounted for in most current GCMs, which simply use $u_{*ft}$
as $u_{*t}$.

One challenge in parameterizing $u_{*ft}$ and $u_{*it}$ lies in the representation of the effect of the soil
particle size $D_p$ on both thresholds. These two thresholds are mainly governed by soil particle diameter
$D_p$, air density $\rho_a$, and soil moisture $w$ (Greeley et al., 1997; Shao and Lu, 2000). Although there are
multiple data sources of globally gridded products for $\rho_a$ and $w$, there are relatively few efforts on
obtaining globally gridded $D_p$ since there are no methods for satellites to observe and derive surface $D_p$
observations. Furthermore, current science only has a good understanding of the saltation dynamics of an
idealized, homogeneous (or monodisperse) soil bed, but natural soil beds consist of particles with a wide
range of $D_p$ (and are heterogeneous or polydispersed). With few comprehensive field studies of saltation
dynamics over mixed soils, past saltation studies either assumed that particles of different sizes saltate
independent of each other (Marticorena and Bergametti, 1995; Shao et al., 1996; Alfaro and Gomes, 2001;
Zender et al., 2003a), or assumed that a single grain size (e.g., the median) could be used to represent the
whole PSD of the soil bed (Elbelrhiti et al., 2005; Andreotti et al., 2010). The dispute of whether the
assumption of "independent" saltation (Shao, 2008) or "representative" saltation (Claudin and Andreotti,
2006) is more appropriate was informed by Martin and Kok (2019), which used both approaches to model
a series of field measurements of saltation fluxes. Their results indicated that modeling the threshold using
a single particle size ("representative" saltation) is more realistic than assuming no interactions between
saltation of different particle sizes. They argued that the median particle diameter $\overline{D}_p$ of the PSD should
be used for calculating the threshold of a mixed soil, and the emission should then be calculated for the
whole soil bed using the median instead of using a "spectral / independent" approach of calculating
emissions of different particle sizes separately.

To model $u_{*ft}$, current models assume that $u_{*ft}$ is mainly dependent on the soil PSD and soil
moisture (Iversen and White, 1982; Marticorena and Bergametti, 1995; Zender et al., 2003a):

$$u_{*ft} = u_{*ft0}(D_p, \rho_a)f_m(w) \tag{1}$$

where $u_{*ft0}$ is the "dry" fluid threshold friction velocity (in m s$^{-1}$) on a smooth and bare surface as a
function of air density $\rho_a(\text{lon}, \text{lat}, t)$ (longitude, latitude, time) and $D_p(\text{lon}, \text{lat})$, which in this study will
be the median diameter $\overline{D}_p$ of a polydispersed, mixed soil PSD. $f_m$ is the correction factor for the presence
of soil moisture $w(\text{lon}, \text{lat}, t)$; $f_m \geq 1$ such that soil moisture protects soil particles from being lifted. $u_{*ft}$
is the "wet" fluid threshold accounting for the moisture effect; dust emission occurs when $u_*$ exceeds $u_{*ft}$
(see Eq. 13). Other factors can also affect $u_{*ft}$, such as salt concentration, organic matter, electrostatic



effects, and surface crusts, but they are not included in most studies because they are not well understood
and modeled (Shao et al., 2011; Foroutan et al., 2017).

$u_{*ft0}$ is parameterized by considering the balance between aerodynamic drag and lift against
gravity and interparticle cohesion on a soil particle. An equation for the moment balance at the moment
of soil particle lifting is given by considering the wind drag $F_d$, aerodynamic lift $F_l$, interparticle force $F_{ip}$,

and gravity $F_g$ on the soil particle: $r_d F_d + r_g F_l = r_{ip} F_{ip} + r_g F_g$, where $r$ is the moment arm of any force
relative to the pivoting point $P$. The Shao and Lu (2000) (hereafter S&L00) scheme derived a simple
solution to the force balance (also see Kok et al., 2012), considering the moment balance and further
assumes that the cohesive force is proportional to particle size. Using wind tunnel measurements (e.g.,
Greeley and Iversen, 1985) they obtained an equation with fitting parameters:

$$u_{*ft0} = \sqrt{A(\rho_p g D_p + \gamma/D_p)}\, \rho_a^{-0.5} \qquad (2)$$
where $g = 9.81$ m s$^{-2}$ is the gravitational acceleration, $\rho_p = 2650$ kg m$^{-3}$ is the typical soil particle density,
$\rho_a$ is in kg m$^{-3}$, and $A = 0.0123$ as well as $\gamma = 1.65 \times 10^{-4}$ kg s$^{-2}$ are empirical constants accounting for
the aerodynamic forces and interparticle forces, respectively. Assuming an air density $\rho_a = 1.225$ kg m$^{-3}$,
Eq. 2 will yield the smallest $u_{*ft0}$ of 0.21 m s$^{-1}$ at $D_p = 80\ \mu m$. For larger sizes the particles are heavier

to lift; for smaller sizes the particles are more strongly bound by interparticle forces.

An alternative parameterization for $u_{*ft0}$ is the Iversen and White (1982) scheme (hereafter
I&W82). They derived a similar solution as S&L00 but further considered the effects of soil particle size
to the airflows, characterized by the particle Reynolds number $\mathrm{Re}_p$. They derived different solutions for
$u_{*ft0}$ in the laminar regime ($0.03 < \mathrm{Re}_p < 10$) and the turbulent regime ($\mathrm{Re}_p > 10$) (see the full solution

in I&W82 or Kok et al., 2012). I&W82 calculates $u_{*ft0} = u_{*ft0}(\rho_p, \rho_a, g, D_p, \mathrm{Re}_p)$ in a similar form to
S&L00, with
$$\mathrm{Re}_p = u_{*ft0} D_p / \nu \qquad (3)$$
where $\nu$ is the kinematic viscosity of air. Since $u_{*ft0}$ is a function of $\mathrm{Re}_p$, which itself is a function of
$u_{*ft0}$, $u_{*ft0}$ is an implicit function of itself and the calculation needs to be iterated a few times given $\rho_a$

and $D_p$ in calculation.

Regardless of whether S&L00 or I&W82 is used, different models make different assumptions for
soil particle sizes $D_p$, including a globally constant value (e.g., Zender et al., 2003a), a function of soil
texture (e.g., Menut et al., 2013), or other forms. For instance, the Community Land Model (CLM), the
land component of CESM, uses a global optimal soil diameter of $D_p = 75\ \mu m$ for the threshold calculation

following Zender et al. (2003a) (Oleson et al., 2013; also see the latest version of CLM5.0 technical note
on https://escomp.github.io/ctsm-docs/versions/release-clm5.0/html/tech_note/index.html; last access on
3 October 2022), and thus $u_{*ft0}$ becomes solely a function of $\rho_a$ in CESM (e.g., $u_{*ft0} = 0.21$ m s$^{-1}$ for
$\rho_a = 1.225$ kg m$^{-3}$ at $D_p = 75\ \mu m$).

Most models parameterize the effect of soil moisture on $u_{*ft}$ following Fécan et al. (1999):

$$f_m = \sqrt{1 + 1.21(w - w_t)^{0.68}} \qquad \text{for } w > w_t \qquad (4a)$$
$$w_t = a(0.17 f_{clay} + 0.0014 f_{clay}^2) \qquad (4b)$$
where $w(\mathrm{lon}, \mathrm{lat}, t)$ is the gravimetric soil moisture (kg / kg) in the shallowest soil layer (see Sect. S1 and
Fig. S1 for the relation between volumetric and gravimetric moisture), $w_t(\mathrm{lon}, \mathrm{lat})$ is the threshold
gravimetric water content above which $u_{*ft}$ increases, $f_{clay}(\mathrm{lon}, \mathrm{lat})$ is the percentage of clay content in

the soil, and $a$, a tunable constant usually of order one, was introduced by Zender et al. (2003a) to account
for the mismatch in the small scales for which Fécan et al. (1999) obtained their parameterization and the
large scales on which it is used in climate models (e.g., Zender et al., 2003a; Mokhtari et al., 2012; Kok
et al., 2014b). $w_t$ increases with soil clay content as water adsorbs onto clay water such that more moisture
is needed to enhance $u_{*ft}$.





Another essential dust emission threshold for this study is the dynamic or impact threshold $u_{*it}$, which is the lowest wind speed or stress to maintain saltation (Kok et al., 2012; Comola et al., 2019b):

$$u_{*it} = B_{it} u_{*ft0} \qquad (5)$$

where $B_{it} = 0.82$ is approximately constant with soil properties and particle size (Bagnold, 1937; Kok et al., 2012). In this study, we propose that dust emission models should use $u_{*it}$ instead of $u_{*ft}$ for dust

emission schemes (e.g., Eq. 10 and Eq. 13), which will cause substantial changes in the simulated spatiotemporal variability of dust emission (see Sect. 4.1). This is needed to allow dust emission when the $u_*$ is intermediate between $u_{*it}$ and $u_{*ft}$, which is especially common in marginal dust source regions. Additionally, this is more physically correct as the dust emission threshold is the minimum friction velocity at which the saltation and dust emission fluxes are non-zero, which is true at $u_{*it}$ but not true at

$u_{*ft}$ (Martin and Kok, 2018; Comola et al., 2019b; Pähtz et al., 2020).

2.2. Parameterization of the drag partition

Apart from the dust emission threshold, another essential parameter for determining the dust emission flux is the wind drag partition effect, $F_{eff}$, due to the existence of land-surface roughness

elements covering the desert surfaces (Raupach, 1992; Marticorena and Bergametti, 1995). It is crucial to account for this effect for accurately simulating the magnitude and spatial pattern of dust emissions. Many past modeling studies treated this effect as increasing the dust emission threshold $u_{*ft}$ (e.g., Raupach, 1992; Marticorena and Bergametti, 1995; Darmenova et al., 2009; Menut et al., 2013), such that the relation is expressed as (Raupach et al., 1993; Marticorena and Bergametti, 1995; Marticorena et al., 1997,

2006; Foroutan et al., 2017; Webb et al., 2020):

$$u_{*ft} = u_{*ft0}(D_p, \rho_a) f_m(w) / F_{eff} \qquad (6a)$$

where $F_{eff} < 1$ when roughness elements are present, such that roughness elements increase $u_{*ft}$ and decrease the dust emission. However, this approach is physically incorrect because roughness elements reduce the wind stress exerted on the bare soil and do not increase the forces resisting particle lifting that

determine $u_{*ft}$ (Kok et al., 2014a; Webb et al., 2020). As a consequence, Webb et al. (2020) showed that dust models using Eq. 6a will overestimate the dust emission flux compared to those using Eq. 6b. A correct emission modeling approach should instead combine Eq. 1 for $u_{*ft}$ with the effect of drag partition $F_{eff}$ to $u_*$:

$$u_{*S} = u_* F_{eff} \qquad (6b)$$

where $u_{*S}$ is called the soil surface friction velocity (Webb et al., 2020). Dust emissions should thus be a function of $u_{*S}$ instead of $u_*$.

There are different schools of drag partition schemes. A major school of drag partition parameterization originated from Arya (1975) and later Marticorena and Bergametti (1995) (hereafter

M&B95), who primarily used the roughness length $z_0$ to quantify roughness. Because of the large differences in the length scales between mountains/orography, rocks and plants, as well as down to soil particles, Menut et al. (2013) distinguished three distinct roughness lengths describing different sizes of roughness. First, the aerodynamic momentum roughness length $z_{0m}$ mainly represents roughness due to large-scale orography, forests, and/or urbanization (with sizes of $10$–$10^3$ m) with values ranging from ~1

cm to 1 m (Menut et al., 2013). Second, the aeolian roughness length $z_{0a}$ quantifies the roughness due to smaller elements such as rocks and vegetation, with a typical order of magnitude of $10^{-3}$ to 10 cm (Prigent et al., 2005; Prigent et al., 2012). $z_{0a}$ is the relevant roughness length that informs the partition of the wind stress when considering the near-surface (~ 1 m) flows in which saltation occurs,. Third, the smooth roughness length $z_{0s}$ quantifies the roughness of a bed of fine soil particles in the absence of roughness

elements. $z_{0s}$ characterizes the roughness of mobile, erodible soil particles over an exposed surface. $z_{0s}$





is directly related to the particle diameter $D_p$ by (Nikuradse, 1950; Sherman, 1992; White, 2006; Pierre et al., 2014b):

$$z_{0s} = 2D_p/30 \tag{7}$$

M&B95 proposed their drag partition scheme by applying the treatment of Arya (1975) of the wind stress over Arctic pack ice to the wind stress over desert roughness elements. M&B95 argued that behind a roughness element (obstacle), an internal boundary layer (IBL) grows and the wind within the IBL follows the log law of the wall as a function of $u_{*s}$ and the local roughness length $z_{0s}$. They then pointed out that without the obstacle, the planetary boundary layer (PBL) wind profile would follow the log law as a function of $u_*$ and $z_{0a}$. By arguing that the two wind speeds must be equal at the IBL height $\delta$, they derived $F_{eff}$ as a function of $z_{0a}$ and $z_{0s}$:

$$F_{eff} = 1 - \frac{\ln\left(\frac{z_{0a}}{z_{0s}}\right)}{\ln\left(\frac{\delta}{z_{0s}}\right)} \tag{8}$$

Later studies improved this equation based on more observations for calibrating several parameters (MacKinnon et al., 2004; King et al., 2005; Darmenova et al., 2009; see Eq. 15 in Sect. 3.2). Historically this scheme has been employed by Marticorena and others to represent the roughness due to rocks (e.g., Marticorena et al., 1997; Darmenova et al., 2009; Menut et al., 2013).

Another major school of drag partition parameterization originated from Raupach (1992) and Raupach et al. (1993) (hereafter R93), which primarily used the roughness density $\lambda$ to quantify roughness. $\lambda$ is defined as the total frontal area of roughness elements divided by the area of land $A$: $\lambda = nhb/A$, where $h$ and $b$ are the obstacle height and width and $n$ is the number of obstacles within the area. Knowing the geometric and aerodynamic properties of the roughness elements, R93 showed that the drag force of the exposed area $\tau_S'$ is related to the total drag force $\tau$, given $\lambda$, the roughness-element basal area-to-frontal area ratio $\sigma$, as well as the ratio of the roughness element-to-surface drag coefficient $\beta$. They then derived the roughness effect by geometric arguments:

$$\frac{\tau_S'}{\tau} = \frac{1}{(1-m\sigma\lambda)(1+m\beta\lambda)} \tag{9a}$$

where $m$ is a geometric parameter to account for the spatial variability of $\tau_S'$ on the erodible surface. Raupach then applied this ratio to the dust emission threshold (per Eq. 6a).

Many later studies used the R93 parameterization for plants (specifically shrubs) with prescribed $\sigma$, $m$, and $\beta$ (Darmenova et al., 2009; Xi and Sokolik, 2015). $\lambda$, however, is related to the abundance of obstacles and is thus spatially variable, and thus far there is no globally gridded datasets of $\lambda$ available. Most studies thus related grid-scale $\lambda$ to other grid-scale properties; for instance, Shao et al. (1996) linked $\lambda$ to the vegetation cover fraction $f_v$ using in-situ observations:

$$\lambda = c_\lambda \ln\left(1 - f_v\right) \tag{9b}$$

where $c_\lambda$ is a proportionality constant. Gridded $\lambda$ could thus be obtained from gridded satellite retrievals of vegetation cover (Gutman and Ignatov, 1998; Wu et al., 2016; Foroutan et al., 2017) or parameterized as a function of other gridded land-surface variables such as the leaf area index (LAI) (e.g., Klose et al., 2021). Later studies have attempted to improve Raupach's parameterization and newer schemes relating $F_{eff}$ and $\lambda$ have emerged (e.g., Okin, 2008).

With a few recent exceptions (e.g., Darmenova et al., 2009; Foroutan et al., 2017; Klose et al., 2021), many previous modeling studies have not accounted for the drag partition effects of both rocks and vegetation on dust emissions (e.g., Ginoux et al., 2001; Tegen et al., 2002; Zender et al., 2003a; Kok et al., 2014b). Many past studies either accounted for only the drag partitioning by rocks (e.g., Marticorena et al., 2006; Menut et al., 2013) or by vegetation (e.g., Shao et al., 2011b; Wu et al., 2016), mainly because it is very challenging to use proxies of both rocks and vegetation in either the M&B95 or R93 scheme. For instance, R93 was historically mostly used for modeling vegetation but not rock drag partitioning because there was no dataset of the $\lambda$ due to rocks. Similarly, vegetation roughness is historically mostly represented by $\lambda$ rather than $z_{0a}$, so there is no globally gridded $z_{0a}$ observations for vegetation that can

be fed into the M&B95 scheme. Some modeling studies (e.g., Klose et al., 2021) generated globally gridded vegetation $z_{0a}$ by relating plant $\lambda$ with $z_{0a}$ (e.g., Minvielle et al., 2003; Shao and Yang, 2005; Marticorena et al., 2006; Foroutan et al., 2017; Klose et al., 2021), but these studies all found slightly
different relations between $\lambda$ with $z_{0a}$, and often the in-situ obstacle height $h$ is required by the relation. It is thus very challenging to model vegetation drag partitioning using M&B95 by converting $\lambda$ to $z_{0a}$ when globally gridded $h$ (short vegetation height but not canopy height) is mostly unknown in GCMs or possesses strong subgrid variability.

       To our knowledge only a few approaches tried to represent both rock and vegetation roughness in
one drag partition scheme, but they all have different limitations. The first approach was proposed by Darmenova et al. (2009) and subsequently modified by Foroutan et al. (2017): They assumed rock $\lambda$ for different land types (e.g., Table 2 in Darmenova et al., 2009), and combined rock $\lambda$ with vegetation $\lambda$ (from Eq. 9b) using Darmenova's double drag partition equation (see Eq. 5 in Foroutan et al., 2017). The second approach was also proposed by Darmenova et al. (2009): They provided measured rock $z_{0a}$ and
vegetation $z_{0a}$ for different land types (Table 1 in Darmenova et al., 2009), determined the dominant land type (bare or vegetated) for each grid, and then applied M&B95 to calculate drag partitioning. The third approach was proposed by Klose et al. (2021): They provided a relation between vegetation $\lambda$, $h$, and $z_{0a}$, plus a relation between vegetation height $h$ and LAI, and thus derived the vegetation roughness length $z_{0a}$ as a function of LAI. They then used Prigent et al. (2012) satellite measurements of $z_{0a}$ (time-invariant,
static, mostly representing rocks), and took the larger $z_{0a}$ between Prigent's static $z_{0a}$ and their dynamic vegetation $z_{0a}$. They finally used M&B95 to calculate drag partitioning with $z_{0a}$. All three approaches tried to represent both rock and vegetation roughness using either $z_{0a}$ or $\lambda$, which was problematic because rocks are better measured in $z_{0a}$ by satellites and plants are better parameterized in $\lambda$. For example, globally gridded rock $\lambda$ was not available and thus the first and second approaches needed to
assign $\lambda$ for rocks as a function of land type, which can be inaccurate and highly uncertain because $\lambda$ also depends on other factors apart from the land type. On the other hand, the second and the third approaches assumed one dominant land type and used one single (either rock or vegetation) $z_{0a}$ to represent the whole grid, but ignored the fact that a grid can be partly covered by plants and partly by rocks, which was only accounted for by the first approach (the double drag partition equation is a function of rock $\lambda$, plant $\lambda$, and
$f_v$). Note that the second and the third approaches had to choose either rock or plant $z_{0a}$ because adding up the rock $z_{0a}$ and plant $z_{0a}$ is not allowed ($z_0$ is not additive, whereas $\lambda$ is additive). The same problem would not occur if there were global scale observations of rock roughness measured in density $\lambda$, which could either be directly applied to the double drag partition equation in Darmenova et al. (2009) or be added upon plant $\lambda$ and then converted to $z_{0a}$ in Klose et al. (2021). All in all, due to insufficient and
inadequate observations, all the aforementioned approaches struggle to accurately represent the combined effect of vegetation and rocks on the drag partition and dust emission. In Sect. 3.2, we will propose a novel approach that incorporates both roughness of rocks and plants and equally respects the $z_{0a}$ and $\lambda$ from both schools of drag partition parameterizations, quantifying the drag partitions of rocks and plants into one hybrid drag partition factor $F_{eff}$.


## 2.3 Parameterization of dust emission flux

       There are multiple available dust emission equations (e.g., Gillette and Passi, 1988; Shao et al., 1996; Alfaro and Gomes, 2001; Ginoux et al., 2001; Tegen et al., 2002; Zender et al., 2003a; Shao, 2004;
Kok et al., 2014b; Evans et al., 2016; and more) implemented in GCMs and ESMs to calculate dust emission fluxes. For example, the Zender et al. (2003a) scheme (hereafter Z03) is based on the Marticorena and Bergametti (1995) dust emission equation and is a popular a popular dust emission scheme adopted by many GCMs (e.g., Miller et al., 2004; Oleson et al., 2013; Meng et al., 2021). Z03 calculates dust emission as follows:

$$F_d = STC_{MB}\varphi f_{bare} \frac{\rho_a}{g} u_{*s}^3 \left(1 - \frac{u_{*t}^2}{u_{*s}^2}\right)\left(1 - \frac{u_{*t}}{u_{*s}}\right) \qquad \text{for } u_{*s} > u_{*t} \qquad (10)$$

where $F_d$ is the emission flux (in kg m$^2$ s$^{-1}$), $C_{MB}$ is a proportionality constant for bridging the gap between local-scale and large-scale dust fluxes, $\varphi = 10^{13.4f_{clay}-6}$ is the sandblasting efficiency, the amount of dust flux produced per unit of horizontal saltation flux as a function of soil clay fraction $f_{clay}$, $u_{*t}$ is the dust emission threshold (in m s$^{-1}$; Z03 used $u_{*ft}$ as $u_{*t}$), and $f_{bare}$ characterizes the fraction of land not covered by vegetation. $S(\text{lon}, \text{lat})$ is an empirical "source function" (Ginoux et al., 2001; Zender et al., 2003a, b; Koven and Fung, 2008) to characterize soil erodibility and thus preferential source regions where fluvial sediment accumulates and scale down emission flux out of desert regions. For $f_{bare}$, Mahowald et al. (2006) used a simple parameterization in which $f_{bare}$ is a pure function of LAI (neglecting the effects of other objects such as snow, rocks, buildings, etc.):

$$f_{bare} = 1 - \text{LAI}/\text{LAI}_{thr} \qquad \text{for LAI} \leq \text{LAI}_{thr} \qquad (11a)$$
$$f_{bare} = 0 \qquad \text{for LAI} > \text{LAI}_{thr} \qquad (11b)$$

While Mahowald et al. (2006) took $\text{LAI}_{thr} = 0.3$, we take $\text{LAI}_{thr} = 1$ in this study instead because 1) observations show that there could be dust emitted from semiarid regions with LAI > 0.3 (Okin, 2008); and 2) Mahowald et al. (2006) did not account for wind drag partitioning due to plants, and thus by setting

a small $\text{LAI}_{thr}$, emission ($F_d \propto f_{bare}$) drops more rapidly with LAI such that the drag partition effect is also incorporated in the $f_{bare}$ term. However, since we are considering $F_{eff}$ in this study, we can set a more realistic $\text{LAI}_{thr}$ value such that $f_{bare}$ becomes less sensitive to LAI.

In this study, we use the Kok et al. (2014b) dust emission equation (hereafter K14) which is increasingly adopted by more GCMs (e.g., Evan et al., 2015 ; Ito and Kok, 2017; Mailler et al., 2017; Tai

et al., 2021; Li et al., 2021, 2022; Klose et al., 2021). One key advance of K14 over Z03 is that K14 eliminated the need to use an empirical, time-invariant source function $S$ to tune the spatial variability of dust emissions. K14 proposed that a dynamical and time-varying soil erodibility (named $C_d$ in K14) can be physically parameterized using the standardized fluid threshold $u_{*st} = u_{*ft}\sqrt{\rho_a/\rho_{a0}}$, which is $u_{*ft}$ scaled to the standard air density of $\rho_{a0} = 1.225$ kg m$^{-3}$:

$$C_d = C_{d0} \exp\left(-C_e \frac{u_{*st}-u_{*st0}}{u_{*st0}}\right), \qquad (12)$$

where $C_d(\text{lon}, \text{lat}, t)$ is the time-varying dust emission coefficient or soil erodibility coefficient, $C_{d0} = (4.4 \pm 0.5) \times 10^{-5}$, $C_e = 2.0 \pm 0.3$, and $u_{*st0} = 0.16\, m\, s^{-1}$. Furthermore, K14 derived a new dust emission equation for $F_d$ (kg m$^{-2}$s$^{-1}$):

$$F_d = C_{tune}C_d f_{bare} f_{clay} \frac{\rho_a(u_{*s}^2-u_{*t}^2)}{u_{*st}}\left(\frac{u_{*s}}{u_{*t}}\right)^\kappa \qquad \text{for } u_* > u_{*t} \qquad (13a)$$

$$\kappa = C_\kappa \frac{(u_{*st}-u_{*st0})}{u_{*st0}} \qquad (13b)$$

where $C_\kappa = 2.7 \pm 1.0$, $C_{tune} = 0.05$ is the proportionality constant, $f_{bare}$ is modeled by Eq. 11, and $u_{*t}$ is again the emission threshold (K14 assumed for simplicity that $u_{*t} = u_{*it} = u_{*ft}$). $\kappa$ is the fragmentation exponent which quantifies the sensitivity of $F_d$ to $u_{*s}$. Here we limit the value of $\kappa$ to 3 in order to prevent excessive sensitivity of the model to wind speeds, which can be problematic around

topography. From Eq. 12, $C_d$ increases exponentially with $u_{*st}$ and thus K14 dust emission is very sensitive to $u_{*ft}$. K14 showed improvements compared with Z03 when evaluated against ground-based DAOD measurements (Kok et al., 2014a, b; Li et al., 2022).

## 2.4 Input data and model description

When calculating dust emissions and other aeolian processes, we employ the meteorological and land-surface variables as input data from the Modern-Era Retrospective Analysis for Research and Applications version 2 (MERRA-2) (Gelaro et al., 2017). MERRA-2 is a reanalysis dataset provided by NASA's Global Modelling and Assimilation Office (GMAO). MERRA-2 has a native resolution of 0.5°





× 0.625° and hourly data assimilation. All MERRA-2 modeled fields and other input variables in this study are listed in Table 1. In this study, we code the dust emission scheme with all new aeolian processes in the statistical programming language R (v4.2.1) as an offline (outputs do not feedback onto input forcings), standalone sandbox model. In this study, we use the standalone model to read in all input atmospheric and land surface forcings for year 2006 and employ equations in Sects. 2 and 3 to compute 2006 dust emissions as outputs and results for Sects. 3 and 4.


**Table 1.** Input meteorological and land-surface variables employed for running the standalone dust emission model in this study.

| Variable | Meteorological parameter (SI unit) |
| --- | --- |
| $u_*$ | MERRA-2 friction velocity (m s$^{-1}$) |
| $\theta$ | MERRA-2 volumetric soil moisture (m$^3$ water / m$^3$ soil) |
| LAI | MERRA-2 leaf area index (m$^2$ leaf / m$^2$ land) |
| $\rho_a$ | MERRA-2 air density (kg m$^{-3}$) |
| $z_i$ | MERRA-2 planetary boundary layer height (m) |
| $T$ | MERRA-2 air temperature (K) |
| $H$ | MERRA-2 Sensible heat flux over land (W m$^{-2}$) |
| $\varphi$ | MERRA-2 porosity (dimensionless) |
| $f_{clay}$ | SoilGrids clay fraction (fraction) |
| $f_{silt}$ | SoilGrids silt fraction (fraction) |
| $z_{0a}$ | Prigent et al. (2005) aeolian roughness length (m) |
| $A_r$ | Rock and bare soil land cover derived from the European Space Agency land cover dataset (fraction) |
| $A_v$ | Vegetation land cover derived from the European Space Agency land cover dataset (fraction) |
| $S$ | Ginoux et al. (2001) or Zender et al. (2003b) source function |


**3 Physics-based parameterization of dust emission threshold**



In this section, we propose additions and improvements to several parameterizations of dust emission physics, which include 1) deriving a more realistic soil median diameter map and including it in the $u_{*ft}$ calculation, 2) proposing a new hybrid approach to incorporate the drag partition parameterizations of both rocks and vegetation, and 3) implementing a parameterization of the effects of turbulence on the intermittency of dust emissions. We will use the improved model from this section to
compute hourly dust emissions in Sect. 4, driven by meteorological and land surface fields.

### 3.1 Improving the description of soil particle size parameter

The PSD of the soil bed is a critical factor to determine the dust emission threshold. In this section,
we focus on deriving a new global soil median diameter (a good proxy for the soil PSD) (Martin and Kok, 2019) as a parameter for computing the dust emission thresholds. Section 5.1 discusses the caveats and limitations of this approach.

### 3.1.1 Motivation and literature compilation of soil particle size distribution

As discussed in Sect. 2.1, Martin and Kok (2019) argued that $u_{*ft}$ of a mixed soil should be determined by the median diameter $\overline{D}_p$ of the soil PSD. Thus, we ideally need a global gridded map of $\overline{D}_p$ to calculate $u_{*it}$ and $u_{*ft}$ over the globe. However, there are only very limited in-situ measurements of soil PSDs (e.g., see Table S1) that are insufficient to compile a global $\overline{D}_p$ map. Meanwhile, extensive
studies have compiled global maps of many other soil properties, such as soil texture, soil bulk density, pH value, soil organic carbon (SOC), cation exchange capacity, and more (e.g., FAO/IIASA/ISRIC/ISS-CAS/JRC, 2012; Shangguan et al., 2014; Hengl et al., 2017; Dai et al., 2019). Therefore, to determine and predict $\overline{D}_p$, we use a compilation of literature measurements to explore and construct relationships between $\overline{D}_p$ with other soil properties such as the clay and silt fractions.
However, many past laboratory studies used the wet sedimentation or wet sieving technique to measure the texture of the soil samples. Wet sieving effectively breaks down soil microaggregates into disaggregated particles and can dissolve soluble minerals (Chatenet et al., 1996), thereby disturbing the estimations of the in-situ soil median particle sizes. In contrast, dry sieving causes a minimal disruption to soil microaggregates and thus Chatenet et al. (1996) argued that the dry-sieved soil PSDs are more
representative of the in-situ, aggregated soil PSDs. Although the soil texture is a disaggregated soil property, $\overline{D}_p$ might depend on soil texture $f_a$ and other soil properties because the strength of interparticle forces is contingent upon soil texture (the clay and silt content), which govern the extent of soil aggregation. Here, we use measurements from past laboratory studies (see Table S1), which contain site-scale, dry-sieved soil PSDs, wet-sieved soil texture $f_a$, and other soil properties to investigate their
statistical relations and infer a new global distribution of $\overline{D}_p$. All studies listed in Table S1 have dry-sieved soil PSD measurements, as well as the wet-sieved sand, silt, and clay fractions. Many studies have recorded soil organic carbon (SOC, %), as well as other properties such as calcite (CaCO₃, %), pH value, and bulk density (g cm⁻³). Figure 1a shows the locations of the measurements of the employed soil studies, and the colors show the aridity where the sites are located. Some studies obtained measurements over a
relatively large spatial domain and we plot only one symbol at the domain centroid representing multiple measurements. Many studies reported PSD measurements extending to diameters in excess of 6000 μm, but we used only PSD measurements in the diameter range of 0 and 2000 $\mu$m that is relevant to dust emission (Zender et al., 2003a). For each dry soil PSD measurement, we obtain the aggregated $\overline{D}_p$ by calculating the 50th percentile of the dry soil PSD.

### 3.1.2 Deriving a global soil median diameter map





We classify the datasets into arid and nonarid groups, since we are primarily interested in $\overline{D}_p$ over desert regions (although we also display the soil behaviours over nonarid regions). We follow past studies (Mahowald et al., 2006, 2010; Kok et al., 2014b) which defined arid (or dust emission) regions using the criterion of LAI smaller than a threshold LAI$_{thr}$, which we take to be 1 (see Sect. 2.3). Section 3.2.2 also describes the MERRA-2 LAI we used in this study to identify the world's arid regions.

After dividing the data into median dry diameters for arid and non-arid soils, we examine the statistical relationships between $\overline{D}_p$ and the soil properties (see Fig. 1b and Fig. S2). Figure 1b shows a scatterplot of $\overline{D}_p$ versus the sum of the soil component that produce substantial cohesion, namely the silt and clay fractions ($f_{silt+clay} = f_{silt} + f_{clay}$). The data exhibits distinctly different trends for non-arid versus arid soils: for non-arid soils, $\overline{D}_p$ increases from 100 μm to greater than 1000 μm with $f_{silt+clay}$ (regression $p$-value = $7.3 \times 10^{-6}$), likely due to increasing cohesion with increasing clay and silt content. In contrast, $\overline{D}_p$ for arid soils shows a small and statistically insignificant increasing trend with $f_{silt+clay}$ ($p$-value = 0.77) with a smaller $\overline{D}_p$ variability (50–250 μm). This flat trend indicates that $f_{silt+clay}$ does not effectively explain the median diameter of aggregated soil particles in arid regions. We examined the relationships of $\overline{D}_p$ with the individual fractions of sand, silt, and clay, as well as with other soil properties including SOC, pH, and CaCO$_3$ (Fig. S2), but these relationships are not statistically significant. We obtain a surprisingly simple finding from the available measurements that there is limited variability in the aggregated $\overline{D}_p$ over the arid regions across different soil textures. We thus use a constant $\overline{D}_{p0}$ as an approximation for arid regions. From Fig. 1b, we summarize the relationship between $\overline{D}_p$ and $f_{silt+clay}$ as:

$$\overline{D}_p = \begin{cases} \Psi_0 + \Psi_1 f_{silt+clay}, & for \text{ LAI} > \text{LAI}_{thr} \\ \overline{D}_{p0}, & for \text{ LAI} \leq \text{LAI}_{thr} \end{cases} \tag{14}$$

where $\Psi_0 = 7.81 \pm 3.70$ μm, $\Psi_1 = 124 \pm 36$ μm, $\overline{D}_{p0} = 127 \pm 47$ μm, and LAI$_{thr} = 1$ as specified in Eq. 11. This empirical formula suggests that some models' assumptions of the relationship between $\overline{D}_p$ and soil texture was possibly inaccurate (e.g., Table 2 of Laurent et al., 2008 assumed $\overline{D}_p$ decreases with $f_{silt+clay}$), and this result could substantially simplify model parameterizations. Additionally, our diameter of 127 μm over arid regions is larger than Z03's assumption of a globally constant optimal diameter of 75 μm. This translates to a modest increase of $u_{*ft0}$ from 0.204 m s$^{-1}$ to 0.216 m s$^{-1}$ (given $\rho_a = 1.225$ kg m$^{-3}$), which slightly decreases global dust emissions by 18 % (see Sect. 4). The uncertainty in $\overline{D}_{p0} = 127 \pm 47$ μm translates to an uncertainty of $u_{*ft0}$ between 0.204 m s$^{-1}$ to 0.234 m s$^{-1}$.

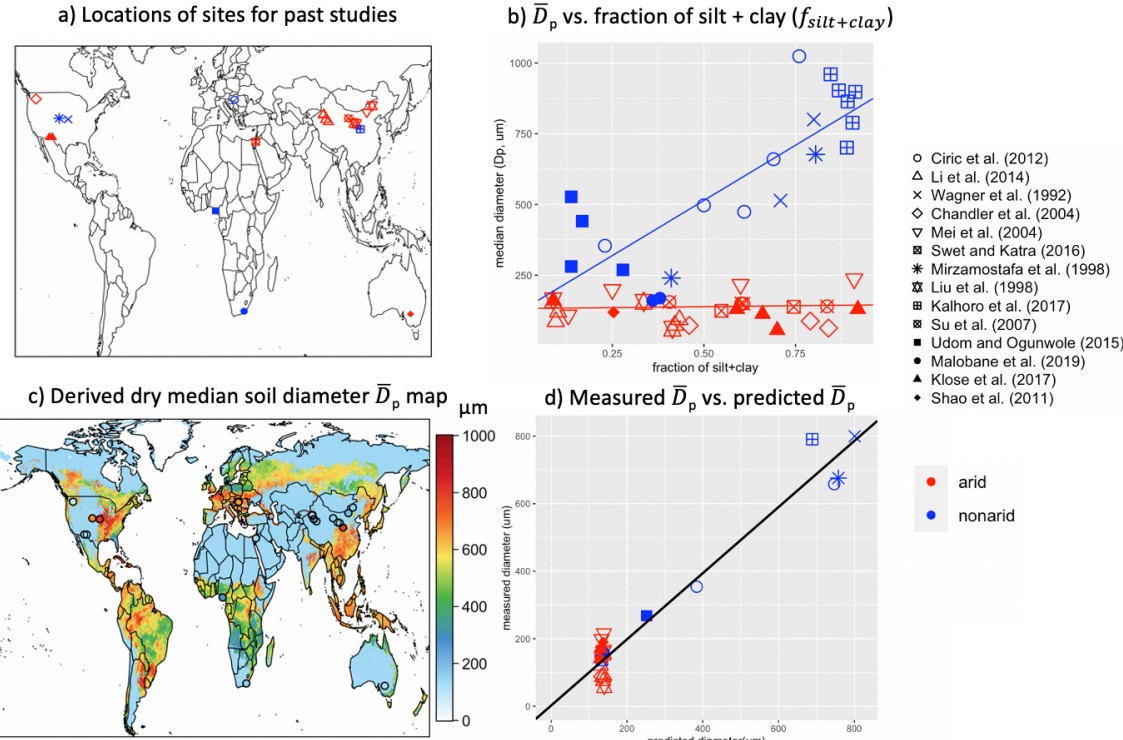

Figure 1. Constructing a global map of the median diameter $D_p$ of aggregated soil particles using soil particle size and texture data. (a) The locations of literature measurements, with symbols indicating the names of the studies and the color indicating the aridity of the locations; a site is classified as arid (red color) if its location has MERRA-2 LAI < 1 and otherwise nonarid (blue color). (b) Literature measurements of soil dry median diameter $D_p$ versus silt + clay fraction ($f_{silt+clay}$). (c) The predicted soil dry median diameter $\overline{D}_p$ map (in μm) derived by projecting our derived $\overline{D}_p-f_{silt+clay}$ relationship of Eq. 14 on the SoilGrids (Hengl et al., 2017) soil texture data (Fig. S3). The circles represent the locations of the sites same as panel (a) and their colors show the measured median diameter $\overline{D}_p$ at those sites. (d) The predicted $\overline{D}_p$ using Eq. 14 versus measured $\overline{D}_p$ from past studies.

We then project our derived relation between $\overline{D}_p$ and $f_{silt+clay}$ on the available soil texture and properties database. We employ global soil properties data from the SoilGrids database (Hengl et al., 2014, 2015, 2017), a global soil mapping project that used machine learning (random forest) to regress in-situ measurements of soil variables (moisture, temperature, nutrients, etc.). SoilGrids provides global maps of soil texture and other soil properties with a horizontal resolution of 250 m and eight soil depths down to 200 cm (Hengl et al., 2017). We use SoilGrids instead of other available soil databases as it shows better performance against observed soil profiles than other soil databases (Dai et al., 2019). Figure S3 shows the SoilGrids relative fractions of sand, silt, and clay with a 0.1° × 0.1° horizontal resolution for the topmost soil layer. Fig. 1c shows our global 0.1° × 0.1° soil median diameter $\overline{D}_p$ map. Following Eq. 14, the arid and semi-arid regions are set to have a $\overline{D}_{p0}$ of 127 $\mu m$, whereas for nonarid regions $\overline{D}_p$ increases with $f_{silt+clay}$. Our derived $\overline{D}_p$ are largely consistent with the site $\overline{D}_p$ measurements from past studies





(overlaid points), showing a similar spatial distribution, with a fit-line slope of 0.98 (*p*-value = 0.007), and an $R^2$ of 81% (Fig. 1d).

**3.2 A wind drag partition scheme for decreasing wind stress and erosion**

We now present a methodology to account for the wind drag partition effect due to nonerodible roughness elements including vegetation and rocks that protect the bare soil by absorbing part of the surface wind stress. We calculate the rock drag partition $f_{eff,r}$ using $z_{0a}$ since global $z_{0a}$ observations are
available, and calculate the vegetation drag partition $f_{eff,v}$ using vegetation cover which is a proxy of $\lambda$ (e.g., Shao et al., 1996; Okin, 2008) since gridded plant cover is often parameterized in GCMs (e.g., Wu et al., 2016; Foroutan et al., 2017; Meier et al., 2022). Here we use two separate drag partition schemes (Marticorena and Bergametti, 1995; Okin, 2008) to quantify the roughness effect of rocks (Sect. 3.2.1) and vegetation (Sect. 3.2.2), respectively. Then, we propose a unifying approach to combining the two
effects into a hybrid factor $F_{eff}$ (Sect. 3.2.3).

**3.2.1 Drag partition due to roughness of rocks**

In this study, we use the aeolian roughness length $z_{0a}$ to quantify the drag partition effect due to rocks. Whereas the smooth $z_{0s}$ and the aerodynamic momentum $z_{0m}$ can be derived from pre-existing
datasets, it is more challenging to quantify the aeolian $z_{0a}$. Existing efforts employed satellite and field measurements to quantify the roughness over deserts (e.g., Greeley et al., 1997; Roujean et al., 1997; Marticorena et al., 2004; Laurent et al., 2005; Prigent et al., 2005; Marticorena et al., 2006; Prigent et al., 2012). For instance, Marticorena et al. (1997) and Callot et al. (2000) developed a 1° × 1° $z_{0a}$ map over Africa and the Middle East by combining topographic data, geological information, aerial pictures and in-
situ observations. Prigent et al. (2005) and Prigent et al. (2012) further used radar measurements to yield global maps of backscatter coefficient, which is a measure of surface roughness because rougher surfaces generally scatter more radar signals to different directions and reduce the backscattering. Comparisons between satellite backscattering signals and field measurements of $z_{0a}$ yielded an empirical formula for extrapolating a global dataset of backscattering signal to global $z_{0a}$. We use here the global aeolian $z_{0a}$
dataset from Prigent et al. (2005) (hereafter Pr05), which contains the climatological monthly mean $z_{0a}$ (12 monthly values per grid) derived from the backscatter coefficient observed by the scatterometer at 5.3 GHz on board the European Remote Sensing (ERS) satellite. Since satellite $z_{0a}$ measurements could quantify both the roughness of rocks as well as vegetation, we take the minimum value out of the 12 months for all grids to obtain a static aeolian $z_{0a}$ map to eliminate as much as possible the vegetation
effect on the inferred roughness. Furthermore, we apply this map over arid regions only (LAI < 1), where the backscatter signal is mostly generated by rocks with little contribution from vegetation roughness. The resulting 2-D map of $z_{0a}$ (in cm) thus mostly represents time-invariant rock roughness and is plotted in Fig. 2a.

Marticorena and Bergametti (1995) derived a parameterization to quantify the drag partition effect
using both $z_{0a}$ and $z_{0s}$. They assumed that this equation is valid for roughness elements that are not too closely spaced (small wake), i.e., $z_{0a}$ < 1 cm (Darmenova et al., 2009). Here we use their semi-empirical equation to quantify the drag partition due to rocks, $f_{eff,r}$ (also see Eq. 8):

$$f_{eff,r} = 1 - \frac{\ln\left(\frac{z_{0a}}{z_{0s}}\right)}{\ln\left[a\left(\frac{X}{z_{0s}}\right)^b\right]} \tag{15}$$

where $X$ is the distance downstream the point of discontinuity in roughness, a length parameter that scales
with the IBL height $\delta$ behind the obstacle in Eq. 8, i.e., $\frac{\delta}{z_{0s}} = a\left(\frac{X}{z_{0s}}\right)^b$ following Marticorena and





Bergametti (1995), and $a = 0.7$ and $b = 0.8$ are empirical constants (King et al., 2005; Darmenova et al., 2009). $X$ should be a function of land type and implicitly space and time, but thus far most dust modeling studies have used a global constant for $X$ (e.g., Darmenova et al., 2009 used a globally constant $X = 0.1$ m). We use a globally constant $X = 10$ m in this study, which is different from what the past studies
suggested, because the scale of the rocks and plants we focus on in deserts are larger and are of the order of $10^0$–$10^1$ m. Some studies considered even larger roughness and used $X \sim 122$ m for vegetated deserts (MacKinnon et al., 2004). We then obtain $z_{0s}$ from our derived global dataset of $\overline{D}_p$ in Sect. 3.1 using Eq. 7. When nonerodible roughness elements are abundant over a surface, $z_{0a} \gg z_{0s}$ and $f_{eff,r} \ll 1$, causing the sheltering of the bare soil from the wind; when there are few roughness elements, $z_{0a}$ is small and
close to $z_{0s}$ and thus $f_{eff,r}$ approaches 1. In Fig. 2a, the red areas with very small $z_{0a}$ are the most susceptible regions for dust emission. Fig. 2a also shows that most arid and semiarid regions have $z_{0a} < 0.2$ cm, such that Eq. 15 can be well following the criterion ($z_{0a} < 1$ cm) in Darmenova et al. (2009). Fig. 2b shows the global $f_{eff,r}$ over arid regions, which is dominated by the spatial pattern of $z_{0a}$ in Fig. 2a given $f_{eff,r}$ is governed purely by $z_{0a}$.


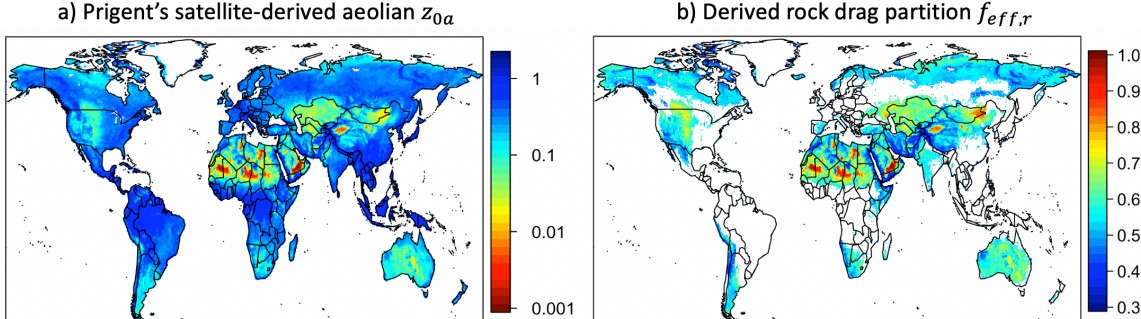

Figure 2. Global roughness length and rock drag partition factor maps at a horizontal resolution of 0.25°
× 0.25°. (a) Global mesoscale aeolian roughness length $z_{0a}$ (in cm) derived by Prigent et al. (2005). (b) Global static rock drag partition factor $f_{eff,r}$ derived by Eq. 15 following Marticorena and Bergametti (1995), derived over arid and semi-arid regions defined as MERRA-2 LAI < 1 in this study. The color schemes are set such that the most erodible regions appear red.


### 3.2.2 Drag partition due to roughness of vegetation

Unlike the very static and slowly evolving rock roughness, vegetation changes temporally. To include the effect of these dynamic vegetation changes on the drag partition, we follow the approach of
Okin (2008) (hereafter O08), which uses unvegetated gap size (the distance between neighboring plants) to characterize the variability of the reduced wind stress. O08 argued that his scheme represents an advancement over the classical R93 scheme, since R93 uses the roughness density (or lateral cover) $\lambda$, which only quantifies how much roughness is on a surface but not how that roughness is spatially distributed. O08 pointed out that, given the same $\lambda$, roughness elements divided into small blocks spread
over the soil surface would be more effective than elements stacked up like a telephone pole in partitioning wind stress (see Fig. 3 in Okin, 2008). Okin argued that since Raupach's model neglects the spatial variability of $\lambda$, the resulting simulated emission flux using the R93 scheme in Okin's paper decreased rapidly with increasing $\lambda$ and unrealistically reached zero at relatively low $\lambda$. To partially compensate this





error, R93 introduced a tuning parameter $m$ (Eq. 9a), serving to reduce the effective $\lambda$ and thereby
reducing the rapid decrease in dust flux. However, $m$ is a tuning parameter not derived from first
principles, and it is not clear how $m$ changes over different surface conditions. Therefore, we use the O08
model here to better characterize the spatial variability of wind stress and the resulting dust emissions.

Here we describe the O08 scheme and adapt it for use in LSMs and GCMs. O08 assumes $u_*$ drops
significantly when encountering a roughness element (plant), and gradually recovers at the lee (downwind
region) of the plant as a function of distance $x$, following:

$$u_{*s}(x/h) = u_*[f_0 + (1 - f_0)(1 - e^{-\frac{x/h}{c}})] \qquad (16a)$$

where $x/h$ is the dimensionless downwind distance from an obstacle normalized by vegetation height $h$
(m), $f_0 = \frac{u_{*s}}{u_*}|_{x=0}$ is the friction velocity ratio immediately behind the obstacle, and $c$ is the dimensionless
e-folding distance (normalized by $h$) over which $u_{*s}$ locally recovers to $u_*$. In this formulation, the *local*
drag partition factor due to vegetation as a function of distance $x$ is:

$$f_{local}\left(\frac{x}{h}\right) = \frac{u_{*s}}{u_*} = f_0 + (1 - f_0)(1 - e^{-\frac{x/h}{c}}) \qquad (16b)$$

Note that in the limit of $x/h \to \infty$, $u_{*s} \to u_*$. O08 used measurements from Bradley and Mulhearn (1983)
and fitted $f_0 = 0.32$ and $c = 4.8$ (i.e., the e-folding distance of $u_{*s}$ recovery to $u_*$ is 4.8 times the plant
height $h$) for semiarid regions.

In order to use Eq. 16b to obtain the drag partition $f_{eff,v}$ relevant to a regionally vegetated area
that is more applicable to GCMs, one needs to calculate an integral for the averaged and aggregated effect
of drag partitioning $f_{eff,v}$ (see Eq. 20a) instead of a locally varying $f_{local}$ (Eq. 16b). Therefore, Okin
employed a probability distribution function as a function of distance $x/h$ to indicate the importance (or
weight) of $f_{local}$ at any $x/h$ to the averaging of $f_{eff,v}$ (McGlynn and Okin, 2006; Okin, 2008). The PDF
is an exponential decay such that the weight of $f_{local}$ decreases with distance $x/h$, so $f_{local}$ at the
immediate lee of the obstacle (which is smaller and close to $f_0$) has more weight than the $f_{local}$ farther
away (which is larger and tends to one). From McGlynn and Okin (2006), the PDF is a function of
normalized distance $x/h$:

$$P_d(x/h) = \frac{1}{K} e^{-\frac{x/h}{K}} \qquad (17a)$$

$$K \equiv L/h \qquad (17b)$$

where $L$ (m) is the mean gap length between obstacles (plants), which is conceptually related to $f_v$; and $K$
is the normalized gap length, which is the gap length $L$ scaled by the plant height $h$. Physically, $P_d$ is the
probability that there is not another obstacle present within a downwind distance $x/h$. This exponential
decay implies that, the farther away from a plant (larger $x/h$), the higher the likelihood that there is another
plant present within the downwind distance $x/h$, with the normalized gap length $K$ quantifying the e-
folding distance of the probability. This PDF governs the spatial domain over which $u_{*s}$ recovers.

For O08, the mean gap length between obstacles $K$ is the only required input for calculating the
drag partition, since $f_0$ and $c$ are assumed to be invariant to surface conditions and desert biome. $K$ can
be expressed as a function of $f_v$ using some simple assumptions. First, O08 argued that vegetation cover
fraction is simply $f_v \equiv \frac{W}{L+W}$, where $L$ is the mean gap length and $W$ is the mean width of the plants within
that vegetated area. Rearranging gives

$$L = W\left(\frac{1}{f_v} - 1\right) \qquad (18a)$$

Then we assume plants in arid regions (e.g., shrubs) are approximately hemispheres with radius $R$. Then,
the plant height $h = R$ and width $W = 2R$ are related by $W = 2h$, which can be substituted into Eq. 18a
to yield

$$K \equiv \frac{L}{h} = 2\left(\frac{1}{f_v} - 1\right) \qquad (18b)$$





and thus we related $K$ to $f_v$. $f_v$ could be measured at the local level and thus O08 was frequently applied in field studies (e.g., Li et al., 2013; Pierre et al., 2014a). However, what is novel in our study is that we are the first to propose the implementation of O08 into LSMs, because Eq. 18b shows us that O08 could be formulated as a function of $f_v$, which is a grid-level parameter. Here we propose to follow the Mahowald et al. (2006) assumption in Eq. 11 and approximate vegetation cover fraction as $f_v = 1 - f_{bare} = \mathrm{LAI}/\mathrm{LAI}_{thr}$, Eq. 18b becomes

$$K \equiv \frac{L}{h} = 2\left(\frac{1}{f_v} - 1\right) = 2\left(\frac{1}{\mathrm{LAI}/\mathrm{LAI}_{thr}} - 1\right) \tag{18c}$$

where we assume $\mathrm{LAI}_{thr} = 1$ (in Eq. 11). The assumption of $f_v \sim \mathrm{LAI}$ is valid if we reasonably assume that leaf areas over arid regions overlap relatively little with each other. We note that by using LAI to quantify $f_v$ in Eq. 18c, we are only accounting for the vegetation drag partitioning due to green (photosynthetic) vegetation and miss that due to brown (non-photosynthetic) vegetation. In the future, it is warranted that Eq. 18b includes other proxies of brown vegetation drag partitioning, such as the vegetation cover quantified by Guerschman et al. (2015), which was adopted by later dust modeling studies such as Klose et al. (2021) and Huang and Foroutan (2022).

To estimate the reduced emission flux, O08 uses an integration approach without quantifying $f_{eff,v}$. O08 calculates the reduced dust emission flux $F_{red}$ (kg m$^{-2}$ s$^{-1}$) by locally integrating the emission $F_d$ following the spatially varying $u_{*s}$ over the normalized distance $x/h$:

$$F_{red} = \int_{x/h=0}^{\infty} P_d\left(\frac{x}{h}\right) F_d\left[u_{*s}\left(\frac{x}{h}\right)\right] d\left(\frac{x}{h}\right) \tag{19a}$$

where $F_d$ (kg m$^{-2}$ s$^{-1}$) is the local emission as a function of $u_{*s}$ which is itself a function of $x/h$. In the integration, $F_d$ needs to be weighted by $P_d$ (which means $F_d$ at large $x/h$ has proportionally less importance) because as $x/h$ increases, the likelihood of the presence of another obstacle gets larger and larger, which will hinder the recovery of $u_{*s}$ to $u_*$. Integrating the emission flux $F_d$ from zero to infinity gives a reduced emission flux $F_{red}$, which will be smaller than the emission flux without roughness elements, defined as $F_{bare} = \int_{x/h=0}^{\infty} P_d\left(\frac{x}{h}\right) F_d(u_*) d\left(\frac{x}{h}\right) = F_d(u_*)$, in which $F_d$ is a constant in space since $u_{*s} = u_*$ is a constant without obstacles.

However, since we need to also combine the vegetation drag partition with the rock partition effects, we need to quantify $f_{eff,v}$ in order to form a hybrid drag partition factor for LSMs. Instead of directly implementing Eq. 19a into LSMs, we require an alternative approach of quantifying $f_{eff,v}$ such that $F_d(f_{eff,v} u_*) = F_{red}$. Quantifying $f_{eff,v}$ for O08 can be useful for comparisons against $f_{eff,v}$ from other schemes such as R93 and Klose et al. (2021). In addition quantifying $f_{eff,v}$ for O08 makes it possible to generate a high-resolution, diagnostic $f_{eff,v}$ dataset for mechanistic models with different resolutions as a model input.

An approach of evaluating $f_{eff,v}$ from O08 was proposed by Pierre et al. (2014a). They obtained the expected value of the shear stress ratio $u_{*s}/u_*$ (SSR in Okin, 2008) between obstacles by evaluating the integral of $u_{*s}/u_*$ weighted by $P_d$, which represents the averaged $f_{local}$ (in Eq. 16b) across the vegetated area and perfectly fits our purposes for implementing $f_{eff,v}$ into LSMs:

$$f_{eff,v} = \int_{x/h=0}^{\infty} P_d\left(\frac{x}{h}\right) \left[\frac{u_{*s}\left(\frac{x}{h}\right)}{u_*}\right] d\left(\frac{x}{h}\right) = \int_{x/h=0}^{\infty} P_d\left(\frac{x}{h}\right) f_v\left(\frac{x}{h}\right) d\left(\frac{x}{h}\right) \tag{20a}$$

Substituting Eq. 17 for $P_d$ into Eq. 20a and analytically evaluating the integral gives a simple algebraic equation for $f_{eff,v}$ (Pierre et al., 2014a), representing the aggregated vegetation drag partition effect at the grid level:

$$f_{eff,v} = \frac{K + f_0 c}{K + c} \tag{20b}$$

This elegant formula conveys a clear physical intuition: If the obstacle does not effectively dissipate
momentum ($f_0 \to 1$), $f_{eff,v} \to 1$; if land is densely covered by vegetation (gap length $K \to 0$), $f_{eff,v} \to f_0(= 0.32)$, the shear stress ratio at the immediate lee of the obstacle. An advantage of this approach is
that it can be easily adopted by gridded models since modelers only need to code an algebraic equation
instead of an integral.

We calculate $0.5° \times 0.625°$ global hourly $f_{eff,v}$ data for Okin's model, using Eq. 18c and Eq. 20b

with hourly MERRA-2 LAI. We note that MERRA-2 LAI is based on the Advanced Very High Resolution
Radiometer (AVHRR) observations (Reichle et al., 2017). Figure 3a shows the annually averaged
MERRA-2 LAI for year 2006 over arid regions with LAI < 1 (seasonal LAI maps are also shown in Fig.
S4), and Fig. 3b shows the corresponding mean $f_{eff,v}$ for areas where LAI < 1. The LAI plot shows the
most erodible regions on Earth.


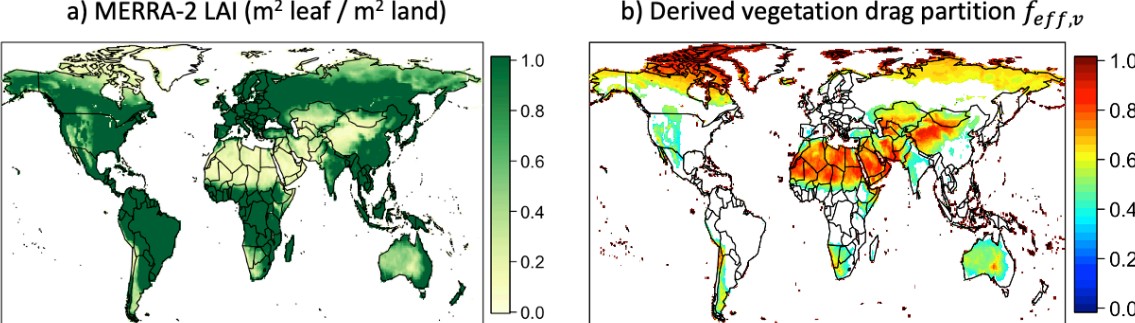

Figure 3. Vegetation drag partition factor $f_{eff,v}$ derived from the Okin (2008) and Pierre et al. (2014) drag

partition model for year 2006 on a $0.5° \times 0.625°$ grid. (a) Annual mean MERRA-2 LAI, with colorbar
saturated at a value of 1. (b) Annually averaged $f_{eff,v}$ derived using the Okin (2008) and Pierre et al.
(2014) drag partition model. White areas indicate water body, ice/snow, or LAI > 1.

### 3.2.3 Combining drag partition factors of rocks and vegetation

After obtaining both the static $f_{eff,r}$ map of rocks and the time-varying $f_{eff,v}$ map of vegetation,
we now propose a methodology to combine the two drag partition sources to capture and represent the
total drag partition effect for dust emission. LSMs need a single drag partition factor capturing all

roughness effects to estimate the total reduction of the surface winds. Thus, we compute a hybrid drag
partition factor map $F_{eff}$ that can be used as input for dust modules in GCMs. To achieve this, we need
to know the fractions of a grid that consists of areas dominated by rocks and areas dominated by plants,
which can be obtained from several recent studies (Lawrence et al., 2016; ESA, 2017; Klein Goldewijk et
al., 2017; Kobayashi et al., 2017). We obtained this data from the European Space Agency Climate Change

Initiative (ESA CCI) dataset (https://www.esa-landcover-cci.org/?q=node/164, last access: 21 June 2022).
The land cover product classifies the land cover of the whole globe into 37 categories (Li et al., 2018),
with relevant land cover over arid regions such as shrub, herbaceous, sparse vegetation, cropland,
grassland, as well as consolidated (gravels and rocks) and unconsolidated (soil) bare land. This dataset
has a horizontal resolution of 300 m, making the dataset capable of counting the portion of the grid

consisting of rocks and vegetation over a larger MERRA-2 $0.5° \times 0.625°$ gridbox (a MERRA-2 grid box
consists of ~35000 grids of 300 m). This dataset gives a representation to the annually varying land covers,



so the rock and vegetation area fractions we use are a function of space only within the simulation year of 2006. We describe our approach to synthesizing the ESA CCI land cover maps and drag partition datasets in the following.


We incorporate the drag partition effects by identifying two roughness regimes using the ESA CCI dataset. The first regime is the rock regime (Fig. 4a), for which we combine the consolidated (gravel and rocks) and unconsolidated (soil) bare land types (types 34–36). This regime is subject to the rock drag partition effect. The second regime is the vegetation regime (Fig. 4b), which includes different vegetation types such as shrubland and herbaceous (types 19–23, 28–29, 32), sparse vegetation (types 26–27), cropland (types 2–5), grassland (type 24), mixed vegetation (type 18), and other vegetation mosaic (types 6–7). Since O08 does not specify the differences in drag partition for different plant functional types (PFTs), here we assume all PFTs produce the same drag partition effect. The overall drag partition effect $F_{eff}$ for a grid is thus defined by the summation of emissions, with emission $F_{d,r}$ over the rock regime with a fractional area of $A_r$, and emission $F_{d,v}$ over the vegetation regime with another fractional area $A_v$:

$$F_d\left(u_* F_{eff}\right) = A_r F_{d,r} + A_v F_{d,v} = A_r F_d(u_* f_{eff,r}) + A_v F_d(u_* f_{eff,v}) \tag{21a}$$

Given that dust emissions approximately scale with the cube of $u_{*s}$ (Zender et al., 2003a; Kok et al., 2014b) and neglecting the effect of the dust emission threshold, Eq. 21a can be simplified to

$$F_{eff}^{\ 3} = A_r\, f_{eff,r}^{\ 3} + A_v\, f_{eff,v}^{\ 3} \tag{21b}$$

such that $F_{eff}$ is simply the weighted mean of drag partition effects. The fractional areas are simply calculated by counting the total occupied area of the ESA CCI land cover corresponding to a certain regime, and then dividing by the total area of the gridbox. We use Eq. 21b to obtain the spatiotemporally-varying $F_{eff}(\mathrm{lon, lat}, t)$, given $f_{eff,v}(\mathrm{lon, lat}, t)$ as well as $A_r$, $A_v$, and $f_{eff,r}$ as functions of $(\mathrm{lon, lat})$. We then apply the obtained $F_{eff}$ here to Eq. 6 to yield $u_{*s}$ for the dust emission equation. We discuss in Sect. 5.2 the caveats and limitations of this hybrid drag partition scheme.

Figures 4a–b show the fractional areas of the two regimes. The rock regime (Fig. 4a) is located mostly over the Sahara, the Middle East, and the Asian deserts. The vegetation regime (Fig. 4b) is concentrated mostly over Australia, the United States, South America and South Africa. Fig. 4c shows the resulting annually averaged $F_{eff}$ using Eq. 21b. The regions with the highest $F_{eff}$ are the Bodélé Depression, El Djouf, the Arabian Desert, and Taklamakan due to high $f_{eff,r}$. The Strzelecki–Sturt Stony Deserts in Australia, the Kyzylkum and Patagonia also have high $F_{eff}$ ($\sim 0.7$) due to high $f_{eff,v}$. Regions with both high rock and vegetation roughness are located in parts of the Middle East and North America with low $F_{eff}$ values.

825

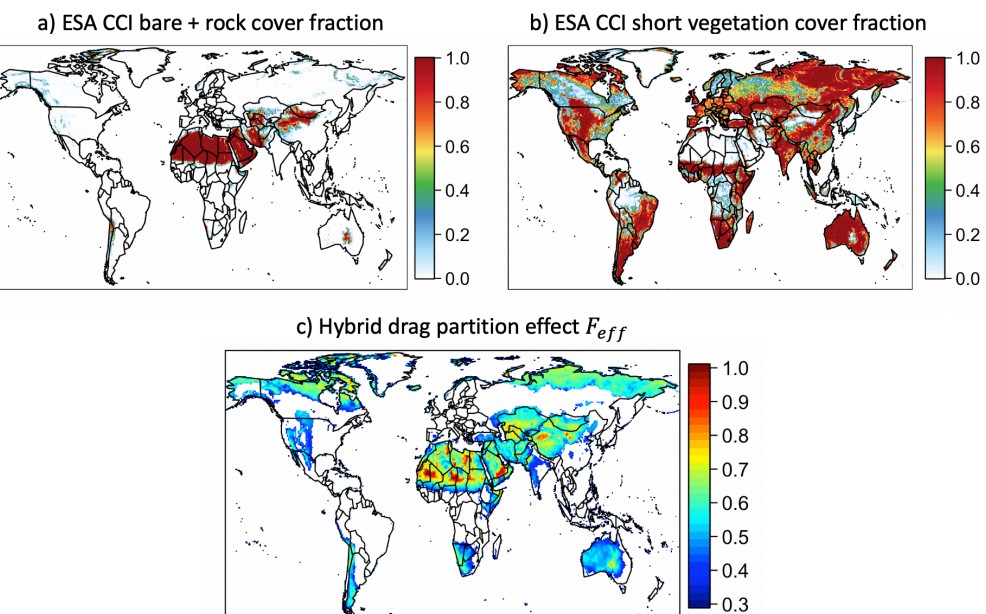

Figure 4. The $0.5° \times 0.625°$ hybrid drag partition factor $F_{eff}$ incorporated using the European Space Agency Climate Change Initiative (ESA CCI) dataset. (a–b) The fractional areas of (a) the rock regime (consolidated/unconsolidated land) and (b) the vegetation regime (shrubs, herbaceous plants, croplands, grassland, and sparse vegetation) over arid regions. (c) The hybrid drag partition factor $F_{eff}$ by a combination of rock drag partition $f_{eff,r}$ and vegetation drag partition $f_{eff,v}$ for year 2006.

### 3.3 Parameterizing the dust emission intermittency

The above improvements enable a more accurate calculation of emission when wind speeds are sufficient to initiate dust emission. Next, we will improve the calculation of the resulting dust emission flux by accounting for the effects of boundary-layer turbulence on dust emission intermittency. Dust emission intermittency exists because saltation is driven by turbulent surface winds, which exhibit strong spatiotemporal fluctuations in speed and direction. Instantaneous winds can thus pass within short timescales across the emission thresholds for initiating or ceasing saltation (Martin and Kok, 2018). Consequently, saltation can be highly intermittent (Comola et al., 2019b), with pronounced variability on timescales of seconds to hours (Dupont et al., 2013). In contrast, existing dust emission parameterizations describe saltation as uniform in time and space and driven by a constant downward momentum flux within a model time step. The disconnect between the reality of intermittent dust emissions and uniform emissions in current theories is likely contributing to the poor performance of dust emission simulations (Barchyn et al., 2014; Todd et al., 2008). Comola et al. (2019b) argued that the intermittency effect is more prevalent for regions with low intensity dust emissions when $u_{*s}$ is regularly fluctuating around the threshold to turn on or shut off dust emissions. Neglecting intermittent dust emissions in current models thus likely degrades the accuracy of dust emission simulations for arid regions during low-wind periods as well as for marginal dust source regions such as semi-arid areas (since soil cohesion increases $u_{*ft}$ but



does not affect $u_{*it}$), which dominate in much of the Southern Hemisphere (Ginoux et al., 2012; Ito and Kok, 2017).

Accounting for the intermittency effect on dust emission fluxes is complicated by the hysteresis of dust emission due to the existence of double thresholds for dust emission physics. The instantaneous wind at the saltation level $\tilde{u}_s$ (we use the tilde to denote instantaneous quantities and take away the asterisk to
denote winds at the saltation level of $z_{sal}\sim0.1$ m instead of a velocity scale) needs to exceed the fluid threshold $u_{ft}$ (also defined at the saltation level) to initiate saltation, but only needs to exceed a smaller impact threshold $u_{it}$ to sustain it (Kok et al., 2012; Martin and Kok, 2018; Comola et al., 2019b). When $\tilde{u}_s$ at a moment lies between both thresholds ($u_{it} < \tilde{u}_s < u_{ft}$), saltation is active if transport was more recently initiated ($\tilde{u}_s > u_{ft}$) and inactive if transport was more recently terminated ($\tilde{u}_s < u_{it}$). This
process is known as hysteresis (Kok, 2010; Martin and Kok, 2018; Comola et al., 2019b). As a result, if $u_s$ (mean of $\tilde{u}_s$ within a model time step) is between $u_{it}$ and $u_{ft}$, there will be fluctuating emission fluxes in reality, while models using a fluid threshold scheme would predict zero emission within a model time step, thereby underestimating the emissions. Meanwhile, models using an impact threshold scheme without considering turbulence will have uniform positive dust emission within the time interval. However,
because in reality high-frequency winds can pass below $u_{it}$ and shut off dust emissions, using average $u_s$ in an impact threshold scheme will overestimate dust emissions. It is thus important for GCMs to account for the effects of turbulence causing both intermittency and hysteresis of dust emission.

As GCMs have a relatively large time step (e.g., ~30 minutes for a 1° GCM) and cannot dynamically resolve high-frequency (~0.1–5 minutes) turbulent wind speed fluctuations, models cannot
directly simulate the dust emission intermittency. Therefore, accounting for intermittent dust emission requires a parameterization that links the low-frequency (~30 minutes) variables of boundary-layer turbulence that are resolved in GCMs to the high-frequency intermittency dynamics. Comola et al. (2019b) formulated a parameterization (hereafter the C19 scheme) of intermittent saltation fluxes by quantifying wind fluctuations due to both shear-driven and buoyancy-driven turbulence in terms of resolved model
parameters, including $u_{it}$ and the Monin-Obukhov length $L$. C19 showed that when a dust emission equation employs $u_{it}$ and accounts for the intermittency effect, it can successfully capture the magnitudes of small dust fluxes otherwise missed by models using $u_{ft}$ (Fig. 3 of Comola et al., 2019b). The C19 scheme will thus moderate the temporal variability of modeled dust emissions due to diurnal wind cycles continuously crossing the thresholds. Additionally, it will also capture more lower intensity emissions
over marginal sources missed by many current models (Zhao et al., 2022).

In the C19 scheme, the dust emission flux $F_d$ is calculated using $u_{*it}$ instead of $u_{*ft}$. We update K14 (Eq. 13) with $u_{*it}$ as the threshold (see Sect. 2 for the description of K14 and dust emission thresholds), giving:

$$F_d = C_{tune}C_d f_{bare} f_{clay} \frac{\rho_a(u_{*s}^2 - u_{*it}^2)}{u_{*it}}\left(\frac{u_{*s}}{u_{*it}}\right)^{\kappa} \qquad \text{for } u_* > u_{*it} \qquad (22a)$$

where $C_d$ is still a function of $u_{*st}$ and $u_{*st} = u_{*ft}\sqrt{\rho_a/\rho_{a0}}$ is the same standardized fluid threshold as in the default K14, and $u_{*it}$ is computed using Eq. 5. Because $u_{*it} < u_{*ft}$, this modified equation allows more small dust fluxes over the marginal source regions that are otherwise missed by employing $u_{*ft}$ as the threshold (see Fig. 7g).

Next, we account for the intermittency effect by introducing the intermittency factor $\eta$, which is
the fraction of time that saltation is active in a model time step (e.g., ~30 minutes). $\eta$ corrects the horizontal sand saltation flux, which scales with dust emission flux (Shao et al., 1993), thereby also representing the fraction of time that dust emission is active in a model time step. C19 accounts for the effect of intermittency by multiplying dust emission by $\eta$:

$$F_{d,\eta} = \eta F_d \qquad (22b)$$



where $\eta \in [0,1]$. Note that C19 parameterizes $\eta$ using wind information at the typical saltation height of $z_{sal} = 0.1$ m instead of the velocity scales (and thus there are no asterisks). $\eta$ is formulated as:

$$\eta = 1 - P_{ft} + \alpha(P_{ft} - P_{it}) \tag{22c}$$

where $P_{it}$ is the cumulative probability that the instantaneous wind (at 0.1 m) $\tilde{u}_s$ does not exceed $u_{it}$, $P_{ft}$ is the cumulative probability that $\tilde{u}_s$ does not exceed $u_{ft}$, and $\alpha$ is the fluid threshold crossing fraction. $\alpha$,
$P_{ft}$, and $P_{it}$ are functions of $\tilde{u}_s$, $u_{it}$, $u_{ft}$, as well as the standard deviation $\sigma_{\tilde{u}_s}$ of the instantaneous $\tilde{u}_s$ (see their full form in Sect. S2, Eqs. S3–S6). $\sigma_{\tilde{u}_s}$ is defined given $\tilde{u}_s$ can be described by a normal distribution (Chu et al., 1996), with its mean being the model time step mean (at 0.1 m) $u$ and its standard deviation $\sigma_{\tilde{u}_s}$. From Eq. 22c, within a model timestep, dust emission is continuous for the fraction of time $1 - P_{ft}$ when $\tilde{u}_s > u_{ft}$, and for the fraction of time $P_{ft} - P_{it}$ dust emission is in the hysteresis regime ($u_{it} < \tilde{u}_s <$
$u_{ft}$) where dust emission can only be active for a fraction of time $\alpha$. With Eqs. S3–S6, we computed $\eta$ to yield the dust emission with intermittency effect $F_{d,\eta}$ as the final dust emission for the LSM. The full C19 intermittency scheme is summarized in Sect. S2 and also discussed in Comola et al. (2919b). $\sigma_{\tilde{u}_s}$ is parameterized using the similarity theory (Panofsky et al., 1977):

$$\sigma_{\tilde{u}_s} = u_{*s} \left(12 - 0.5 \frac{z_i}{L}\right)^{1/3} \qquad \text{for } 12 - 0.5 \frac{z_i}{L} \geq 0 \tag{23}$$

where $L$ is the Obukhov length and $z_i$ is the modeled PBL height. Note that MERRA-2 does not provide $L$ output, and in this study we computed $L$ from the MERRA-2 outputs of $u_*$, $\rho_a$, sensible heat flux $H$, and temperature $T$ for our simulations (see Sect. S3). In boundary-layer dynamics, turbulence is generated by mechanical shear and buoyancy (Stull, 1988). From Eq. 23, high-frequency wind fluctuations $\sigma_{\tilde{u}_s}$ increase with shear ($u_{*s} > 0$) and buoyancy ($L < 0$). For larger $\sigma_{\tilde{u}_s}$, it is easier for $\tilde{u}_s$ to sweep across $u_{it}$
and shut off dust emission, leading to $\eta < 1$. If $u_s \gg u_{ft}$, $\tilde{u}_s$ will be less likely to sweep across $u_{it}$ and $\eta$ will approach 1. If $u_{it} < u_s < u_{ft}$, $\eta$ will be much smaller than one, leading to a small emission flux when other parameterizations predict a zero emission flux. When $u_s < u_{it}$, $\eta$ could also be greater than zero when $\sigma_{\tilde{u}_s}$ is large enough so that the instantaneous $\tilde{u}_s$ crosses through $u_{it}$, but C19 would not generate any emission anyway according to Eq. 22a (which is a technical flaw of C19; see a discussion in Sect.
5.3). We note that Eq. 23 is not the traditional Monin-Obukhov similarity theory as the zonal fluctuation was shown to correlate poorly with $z/L$ but relates much better with $z_i/L$ (Panofsky et al., 1977). We also note that Eq. 23 only applies to the convective PBL, but dust emission often occurs during daytime within the convective boundary layer (Yu et al., 2021). See a discussion to the limitations of this scheme in Sect. 5.3.

Here we show some significant results from the intermittency scheme. Figure 5 shows the global dust emission thresholds in 2006 computed using MERRA-2 fields. Figure 5a shows $u_{*it}$, computed using a globally constant $\overline{D}_{p0} = 127$ $\mu$m (Sect. 3.1). Its spatial variability is purely a function of $\rho_a$ (Eq. 5). $u_{*it}$ is around ~0.16–0.24 m s⁻¹, and higher $u_{*it}$ implies higher altitude. A lower $u_{*it}$ leads to a smaller
aerodynamic drag force from the airflows given the same wind speed; conversely, there will be a large $u_{*it}$ for soils over low $\rho_a$ regions. Figure 5b shows $u_{*ft}$ (Eq. 1). It varies between 0.2 and 0.9 m s⁻¹. Its spatial variability is dictated by the spatial variability of soil moisture $w$ (see Fig. S1). Regions with the lowest $u_{*ft}$ are the driest places in the world, which are all deserts. Regions with the highest $u_{*ft}$ are wet soils covered by rainforests, boreal forests, tundras/permafrosts, and snow. Figure 5c shows the ratio of
$u_{*ft}/u_{*it}$, for which the spatial variability is again dictated by that of $w$. The magnitude of the ratio conveys not only the strength of the soil moisture effect on the threshold but also the width of the hysteresis regime. Deserts with $u_{*ft}/u_{*it} \sim 1/B_{it}$ have a narrow hysteresis regime ($u_{*it} < u_{*s} < u_{*ft}$) and smaller thresholds, and thus tend to have more continuous dust emissions. Semiarid and nonarid regions with larger $u_{*ft}/u_{*it}$ tend to have a wide hysteresis regime and thus dust emissions will be more intermittent.




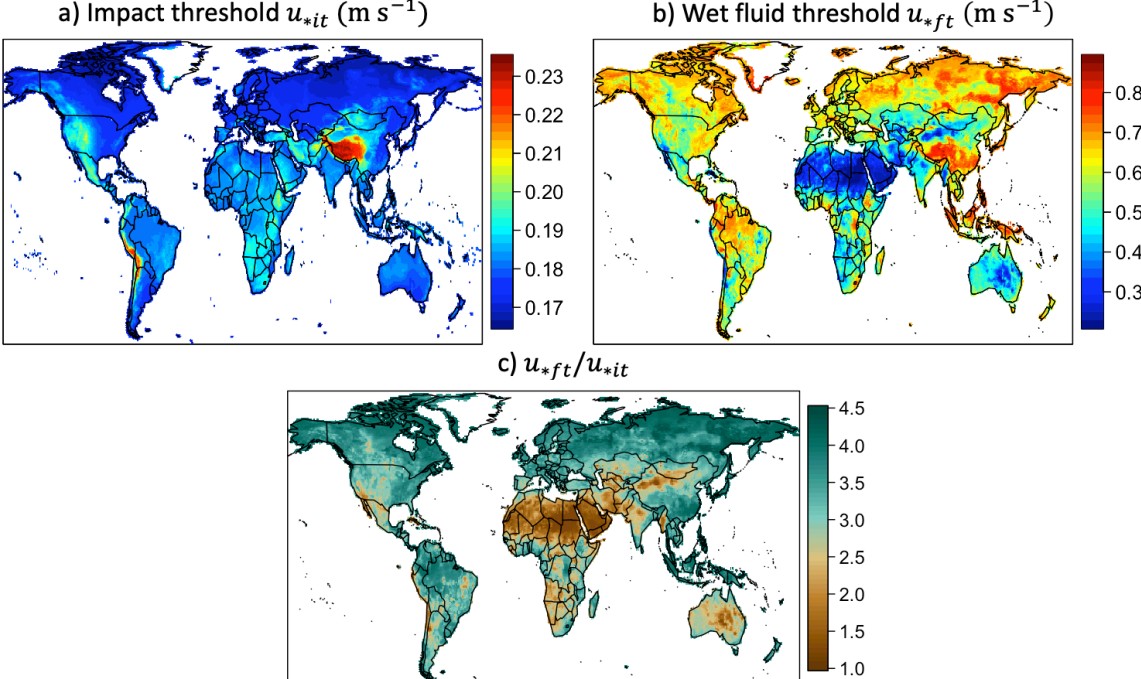

Figure 5. The dust emission thresholds using the (Shao and Lu, 2000) scheme for the year 2006 on a 0.5°
× 0.625° grid. (a) the impact threshold $u_{*it}$ calculated using $\bar{D}_{p0} = 127\ \mu m$ (Eq. 4), (b) the wet fluid
threshold $u_{*ft}$ (Eq. 1), and (c) the ratio between the wet fluid threshold and impact threshold which is the
$f_m / 0.81$, where $f_m$ is the moisture effect on $u_{*ft}$ (Eq. 3). The larger this ratio, the wider the range of wind
speeds for which hystesis in dust emission occurs and the more important it is to account for intermittency
in dust emissions.


Figure 6a shows the 2006 annual mean intermittency effect over the Bodélé Depression as an
example. Figure 6a shows hourly mean $\eta$ as a function of the hourly mean $u_s$. It demonstrates the
properties of $\eta$ discussed above: e.g., when $u_s > u_{ft}, \eta \to 1$, and when $u_s < u_{it}, \eta \to 0$. In both regimes,
the behaviour of the dust emission intermittency is asymptotic to dichotomous (0 or 1) activity which is
the same as that of the conventional emission schemes. Near the intermittency or hysteresis regime $u_{it} <
u_s < u_{ft}, \eta$ is intermediate between zero and one, and thus a scheme using $u_{*it}$ gives a small finite
emission flux while conventional schemes using $u_{*ft}$ give a prediction of zero. The color code shows the
strength of convection $-z_i/L$. $-z_i/L$ is positive (red) when buoyant convection is active ($L < 0$), and is
negative (blue) when the PBL is statically stable ($L > 0$). The color shows that there is a modest correlation
between $-z_i/L$ and $u_s$ but the correlation is not necessarily strong, and the strongest buoyancy (dark red)
often happens when $u_s$ (or shear $u_{*s}$) is moderate. The strongest buoyancy associates with moderate $\eta$
values of ~0.5 only, and for the highest $\eta$ values $-z_i/L$ is mildly unstable (light red). This means that the
turbulent fluctuation is primarily governed by shear $u_{*s}$ instead of controlled by buoyancy $-z_i/L$, and the
intermittency behavior is dictated by shear-driven instead of buoyancy-driven turbulence. Eq. 23 could
essentially be simplified into $\sigma_{\tilde{u}_s} \approx 12^{1/3} u_{*s}$.



Figure 6b shows the global spatial distribution of the annual mean $\eta$ for year 2006, averaged across time steps during which saltation and dust emissions are occurring over the grid (and thus $\eta$ during time steps when $F_d = 0$ are not counted). Most marginal sources have small $\eta < 0.3$ (red color), indicating dust emissions are fluctuating and intermittent. These emissions may not be existent in other LSMs employing

$u_{*ft}$ in the dust emission equation (but also dependent on their threshold tuning). Over these regions, because of the intermittent shut offs of emissions, the emissions need to be scaled down by the fraction of time $\eta$ which is also missed by other LSMs. Intermittency is thus critical in accounting for emissions over semiarid regions. Regions with high $\eta$ (more continuous emission; light blue color) include El Djouf, the Bodélé Depression, the Libyan–Nubian Desert over Libya, Egypt and Sudan, the Rub' al-Khali Desert

within the greater Arabian Desert, the Lut Desert in Iran, the Taklamakan Desert, and the Strzelecki Desert in Australia. During saltation, these regions tend to have wind episodes further away from the thresholds, leading to a high $\eta$. However, regions with high $\eta$ are not necessarily regions with the highest dust emissions (Figs. 7a–b) because their saltation frequency could be low, or the strength of their emission fluxes is limited by other factors such as the fragmentation exponent and soil moisture. In Fig. S5, we

further show the factor $\eta$ averaged over all time steps of 2006, including periods of no emissions over the grid. Since $\eta$ is close to 0 when there is no emission, Fig. S5 shows much smaller $\eta$ compared to Fig. 6b.


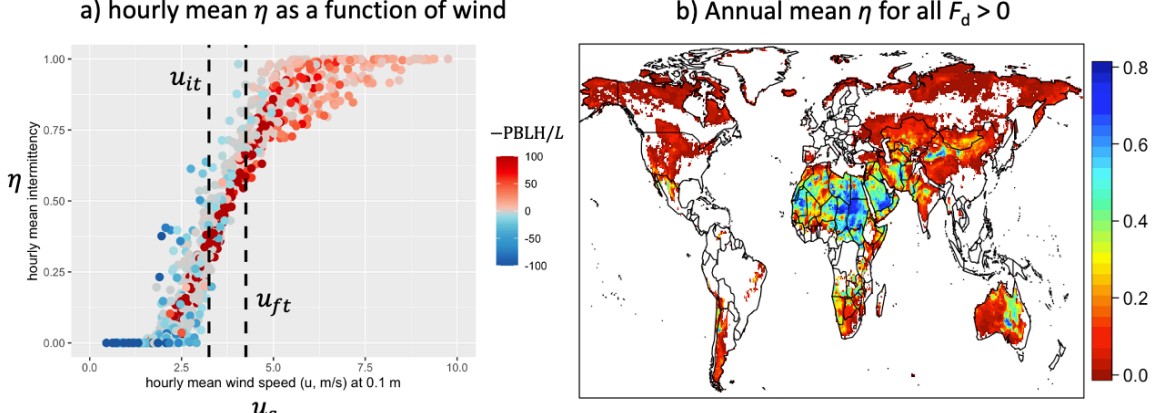

Figure 6. Simulated dust emission intermittency effect for year 2006. (a) The hourly mean intermittency factor $\eta$ versus hour mean wind speed (m s$^{-1}$) at 0.1 m height over the Bodélé Depression. The color indicates $-z_i/L$, with red indicating $-z_i/L > 0$ (unstable), blue indicating $-z_i/L < 0$ (stable), grey

indicating $-z_i/L \sim 0$ (neutral). The two vertical dotted black lines indicate the annual mean impact threshold (left) and the fluid threshold (right) wind speed (m s$^{-1}$) at 0.1 m. (b) The annual mean intermittency factor $\eta$, or equivalently the fraction of time within a time step that emission is active, averaged over times when emission is active ($F_d > 0$).


## 4 Results of our new dust emission scheme

In this section, we implement the three new parameterizations of key dust emissions processes upon the K14 model into R to investigate the resulting spatial variability of dust emissions. The MERRA-

2 data are for year 2006, and other input data are regarded as climatological datasets. In Sect. 4.1, we then





compute the dust emissions and analyze their spatial characteristics. We examine the effects of each new modification on the simulated dust emissions by conducting different simulations with different individual parameterizations added. Then, in Sect. 4.2, we inspect the effects of the grid-resolution of input data on simulating dust emissions, and propose a simple method to calibrate the spatial variability of low-
resolution dust emissions to match the spatial variability of high-resolution emissions.

### 4.1 Effects of different new physics on global dust emissions

In this subsection, we show the effects of different modifications on the resulting dust emissions (in kg m$^{-2}$ yr$^{-1}$) in Fig. 7. We demonstrate the effect of each modification by creating a suite of sensitivity
experiments as follows: (I) First we simulate emissions using the default K14 scheme; (II) then we use the K14 scheme with our derived soil diameter of $\overline{D}_{p0} \sim 127$ $\mu m$ for simulation; (III) we further add in the hybrid drag partition physics on top of (II); (IV) we switch from using the fluid threshold to the impact threshold in (III); and finally (V) we include the intermittency effect on top of (IV) for simulation. We note that past studies derived the approximate magnitude of the global total emission (e.g., Tegen and
Fung, 1995; Zender et al., 2004; Evan et al., 2014; Kok et al., 2021), but there are no global observations of dust emissions. In the field of dust modeling, there are currently no first principles that can derive the essential dust emission proportionality constants to constrain modeled emissions at a correct order of magnitude, which means scientists still have insufficient knowledge in aeolian physics to generate predictions in the correct order of magnitude. Recognizing that the spatiotemporal characteristics of the
predictions are more credible, it is very common for dust modelers to rescale the emissions according to the known constraints of observed atmospheric dust mass or the global DAOD. For instance, Li et al. (2022) scaled all their simulations to achieve a global mean DAOD of 0.03, based on Ridley et al. (2016). Thus, what matters the most is how each modification changes the spatial variability and the relative magnitudes of dust emitted from one region compared to the others, and the absolute magnitude changes
are of secondary importance. Here in Fig. 7, the left panels (Figs. 7c, e, g, i) show the unnormalized absolute changes in emissions from one experiment to another, and then the right panels (Figs. 7d, f, h, j) show the normalized emission ratios from one experiment to another. In calculating the ratios, we normalize all simulations to a global total of 5000 Tg yr$^{-1}$, which is around the current constraint of global PM$_{20}$ dust emission flux from past studies (Evan et al., 2014; Kok et al., 2021a, b). The maps of ratios on
the right panels thus show the changes in the dust emission spatial variability better than the left panels. Fig. S6 shows the unnormalized emission maps for all experiments.

Figure 7a shows the unnormalized emissions from the default K14 scheme (experiment I) using MERRA-2 inputs, giving a global total of ~29300 Tg yr$^{-1}$. The emission map shows a similar spatial pattern compared with the K14 simulations in Kok et al. (2014b) and Li et al. (2022) despite differences
in the data used for different land surface fields (e.g., for LAI and soil moisture). The most significant emissions are over the Bodélé Depression in Chad, the Nubian Desert in Sudan and Egypt, the whole Arabian Peninsula, most of Iran, the Taklamakan Desert in China, and the Strzelecki Desert in Australia. Over regions with emissions, the spatial variability of the emissions is dictated by the moisture $w$ (see Fig. S1 and Fig. 5b), which fundamentally shapes the threshold $u_{*ft}$, the soil erodibility $C_d$, and the
intermittency $\eta$. Smaller emissions occur over South Africa, South America, and the western U.S.

Figures 7c–d show the effect of changing the global soil diameter from the current standard of 75 $\mu$m to our new constraint of $\overline{D}_{p0} = 127$ $\mu m$ (experiment II). As described in Sect. 3.1, a larger $\overline{D}_p$ leads to heavier particles, resulting in higher thresholds and lower emissions. Therefore, the global emission flux is ~23900 Tg yr$^{-1}$, 18 % less than the unnormalized emission flux of experiment I. Figure 7c (and
subsequent difference maps) shows that larger emissions reductions occur over the major sources simply because the emission magnitudes are larger there. The light blue color in Fig. 7c shows that the effect of employing a new global $\overline{D}_{p0}$ is relatively mild compared to the drag partition effect and intermittency (Figs. 7e and g). Then, Fig. 7d shows the ratio map of the normalized emissions of experiment II to those

of experiment I, which shows the changes in solely the spatial variability of the emissions. Employing a new globally constant $\overline{D}_{p0}$ does not strongly impact the spatial pattern of the emissions, so the ratio map is of order one around the globe. While Fig. 7c shows that the largest absolute emission changes occur over the major sources, Fig. 7d shows that after rescaling there are some mild emissions reductions (in blue) over marginal regions (e.g., the Arctic) compared to the minimal increases over the major sources (in light red). The major sources are less affected since in the high $u_{*s}$ regime $F_d$ becomes more sensitive to $u_{*s}$ than to $u_{*ft}$. Thus, a uniform increase in $u_{*ft}$ around the globe tends to eliminate small emissions more than large ones. The main effect of employing a larger $\overline{D}_p$ is therefore a very modest shift of emissions from the marginal regions toward the arid areas.

Figures 7e–f show the effect of including the hybrid drag partition effect $F_{eff}$ (experiment III). Rocks and plants ($F_{eff} < 1$) reduces $u_{*s}$ and thus the global total emission decreases drastically by 85 % relative to that of experiment II to ~2880 Tg yr$^{-1}$ (Fig. S6c and Fig. 7e). Many significant emissions reductions (in dark blue) occur over the Sahara where $F_{eff}$ is 0.7 or smaller, such as Egypt, Sudan, and Western Sahara. K14 struggles to distinguish major sources (e.g., the Bodélé) from less significant sources (e.g., Sudan) and predicts similar levels of emissions (Fig. 7a). $F_{eff}$ effectively introduces the effect of surface roughness on mitigating emissions over secondary sources, reducing the magnitude of emissions by at least one order of magnitude. A clearer contrast between major and marginal sources is shown in the ratio map in Fig. 7f, which mirrors the spatial pattern of $F_{eff}$ in Fig. 4c. For example, normalized emissions increase (in red) over major sources such as the Bodélé Depression, El Djouf, and the Rub' al Khali Desert; normalized emissions over the western U.S. and western Australia are significantly reduced. El Djouf is a significant dust source over Africa (Yu et al., 2018), yet K14 fails to represent its high emissions because of the strong moisture effect (see Fig. 5c) compared to other major sources such as the Bodélé Depression. However, $F_{eff}$ highlights El Djouf as a highly erodible surface and helps mitigate the low emission issue over there. Similarly, the underrepresented emissions over Taklamakan and the Arabian Desert by K14 is partially mitigated by accounting for the drag partitioning (light red in Fig. 7f).

Figures 7g–h show the effects of implementing the C19 intermittency scheme. It consists of employing $u_{*it}$ in K14 (experiment IV) and further multiplying the dust emission flux by the intermittency factor $\eta$ (experiment V). Figures 7g–h show their combined effects by comparing experiment V with experiment III. First, Fig. 7g first shows the effects of employing $u_{*it}$ instead of $u_{*ft}$ since the unnormalized total emission vastly increases to ~13200 Tg yr$^{-1}$ (Fig. S6d), more than four times that of experiment III. As seen from the ratios in Fig. 7h, emission schemes employing $u_{*it}$ will have slightly higher emissions over major sources but much stronger emissions over marginal sources. This is because not only $u_{*ft}/u_{*it}$ but also the fragmentation exponent $\kappa$ (which scales with $u_{*ft}$) is also greater over marginal sources. As a result, the main feature in the spatial pattern of Fig. 7h is that the marginal sources (in red color) now have more emissions than experiments II and III. The most apparent emissions increases are over Patagonia, the Horn of Africa (HOA), and western U.S. deserts. Another observable change is that there are many more high-latitude dust emissions, such as over the Arctic, Canada, and Alaska. Studies reported that many models greatly underestimate high-latitude dust emissions (Bullard et al., 2016; Meinander et al., 2021), and the use of $u_{*it}$ will mitigate this issue.

On the other hand, the effect of multiplying the emission flux with the intermittency factor $\eta$ is less dramatic than the effect of using $u_{*it}$. As shown in Fig. 5b, $\eta$ tends to scale down small emissions, so fluxes from major sources are only moderately reduced. In contrast, fluxes from marginal source regions (e.g., high-latitude boreal forests) are typically reduced by ~1–3 orders of magnitude (see the blue areas in Fig. S6e). However, Experiment V has a global total of ~11700 Tg yr$^{-1}$, which is only 11 % smaller than experiment IV, because those remote regions as such already have small emissions. All in all, accounting for both the impact threshold and the intermittency factor will increase the global total emission from ~2880 Tg yr$^{-1}$ in experiment III to ~11700 Tg yr$^{-1}$ in experiment V, which is about a four-



fold increase. Fig. 7h shows that C19 mainly increases marginal emissions; the overall effect of C19 is thus to move emissions from the hyperarid regions to semiarid regions.

Figure 7b shows the final emission map of our new dust emission scheme with all new physics, and Figs. 7i-j show the resulting emission changes and ratios from K14 to our scheme. Our scheme's global total emission of 11700 Tg yr$^{-1}$ is ~60 % smaller than the K14 emission. Figures 7i-j show that compared to K14, our scheme's emission fluxes over densely vegetated regions (e.g., equatorial Africa and northern Australia) are reduced due to the drag partition effect, while there are increases in marginal sources like the Arctic and mid-latitude boreal forests due to the intermittency effect. Figure 7j shows the corresponding ratios of our scheme's emissions (experiment V) to the K14 emissions (experiment I), which is essentially a combination of Fig. 7f and Fig. 7h. Major sources are more affected by the drag partition effect (e.g., the Bodélé Depression and El Djouf), while marginal sources are more dominated by turbulence and intermittency (e.g., the Arctic). For regions where both effects take place, more vegetated semiarid regions are more affected by the drag partition (e.g., western U.S.) while less vegetated semiarid regions are more affected by the intermittency (e.g., Patagonia, the Great Plains of the U.S., and southern Australia). A notable feature shown in Fig. 7i is that the new mechanisms favor the emissions over the Horn of Africa (HOA) the most, with an emission increase of ~2 kg m$^{-2}$ yr$^{-1}$ (as seen in Fig. 7g), because in C19 the HOA has low $u_{*it}$, high summertime $u_*$, low roughness element cover (and thus high $F_{eff}$) and moderate soil moisture (and thus high dust emission coefficient $C_d$). This issue could be problematic since it could introduce too much dust in the GCM over the HOA, which is further discussed in Sect. 5.5.

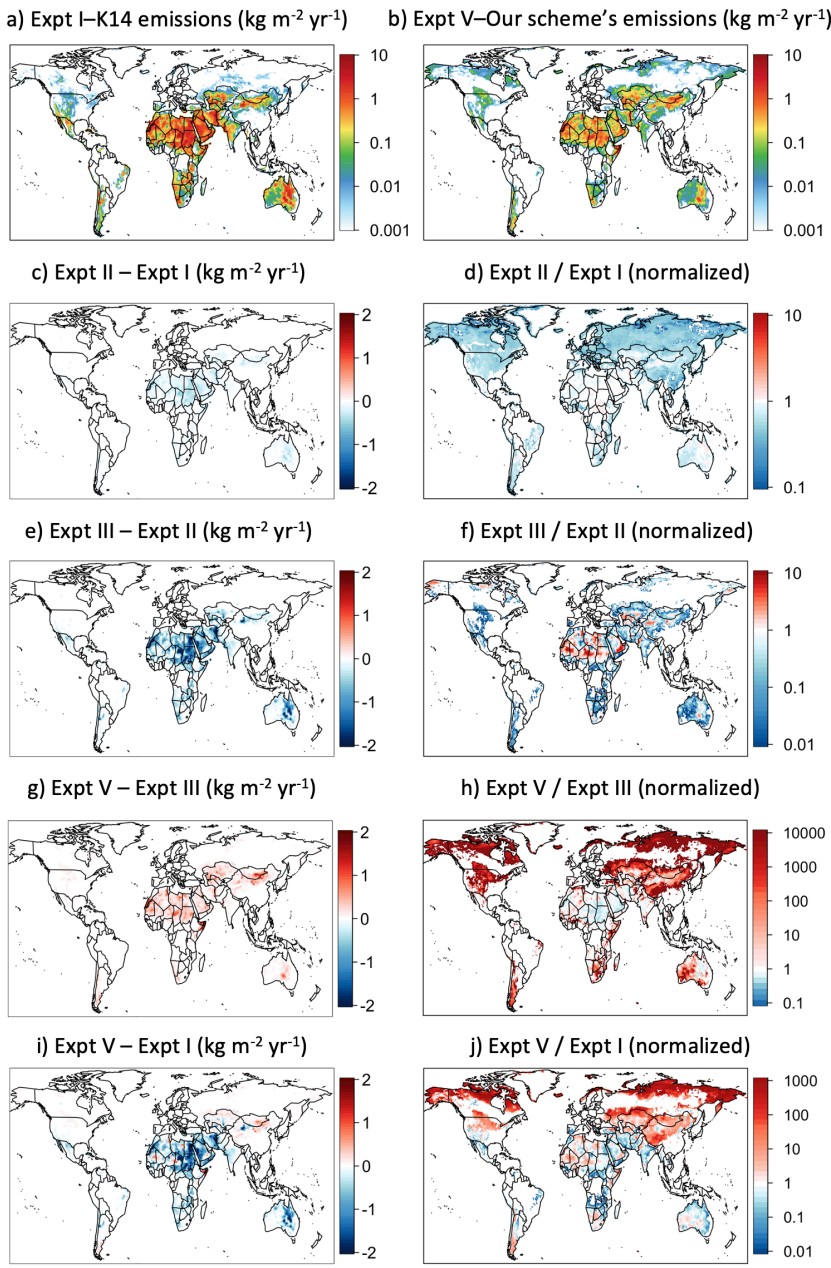

Figure 7. The effects of the proposed improvements to the parameterization of dust emissions on the default Kok et al. (2014a, b) dust emission scheme. (a, b) Global unnormalized dust emission fluxes simulated by (a) the default K14 scheme (expt. I) and (b) our new scheme (expt. V). (c-j) Maps of (c,e,g,i) unnormalized emission differences and (d,f,h,j) of the ratios of the normalized emissions, with individual improvements added on top of the default K14 scheme. The individual improvements are respectively (c,d) changing the soil median diameter (expt. II), (e,f) including the drag partition effect (expt. III), and (g,h) employing the Comola et al. (2019) intermittency scheme (expt. V). (i,j) Maps of (i) unnormalized emission differences and (j) emission ratios for our new scheme and the K14 scheme.





### 4.2 The grid scale-dependence of our new dust emission scheme


The spatial resolution of a GCM strongly affects the budget and spatiotemporal variability of dust emissions because modeled emissions scale nonlinearly with input meteorological fields (Ridley et al., 2013; Meng et al., 2021). Many current GCMs have a horizontal resolution of $\sim 1°$–$2°$ (Zhao et al., 2022). Most of the datasets employed in this paper, such as the 0.5° MERRA-2 fields and the even finer aeolian

roughness length $z_{0a}$, are datasets of higher horizontal resolutions. Thus, GCMs need to regrid the datasets to the model native grid resolution as model input. Since dust emission has a nonlinear dependence on multiple variables such as $u_*$ and $w$, using a simple, area-weighted spatial average of $u_*$ to calculate dust emission would be inaccurate as it is different from an area-weighted average of high-resolution dust emissions per se, i.e., for any $n > 1$ $\bar{u}_{*s}^n < \overline{u_{*s}^n}$ leads to $F_d(\bar{u}_{*s}^n) < \overline{F_d(u_{*s}^n)}$ (Ridley et al., 2013) (here we

use the bar as a spatial average from a finer to a coarser grid). This inequality applies to our model and datasets as well: simply taking the area-weighted mean of high-resolution $u_{*s}$ or $w$ in a model grid box will omit the locally high $u_{*s}$ (or low $w$) values that can produce locally extremely high emissions, resulting in an underestimation of emissions relative to direct area-weighted averaging of the emissions. Additionally, the presence of thresholds in dust emission parameterizations further intensifies the scale

dependence, because spatially averaging $u_{*s}$ might cause $\bar{u}_{*s} < \bar{u}_{*it}$ which leads to zero emission over a coarse gridbox, whereas fine emissions could be large than zero when $u_{*s} > u_{*it}$ in any fine grid. Dust emission flux will thus be strongly dependent on model resolution more than other linear processes (Zender et al., 2003a; Ridley et al., 2013; Meng et al., 2021), which is undesirable. There is a need to better upscale low-resolution dust emissions to match the variability of high-resolution emissions such

that dust emissions tend to be less resolution dependent. To address the problem of missing emissions due to the smoothing of the subgrid wind maxima, a common approach is to employ a grid-by-grid Weibull distribution to the GCM winds to represent the subgrid wind maxima and thus obtain the subgrid emission peaks (Cakmur et al., 2004; Grini et al., 2005; Cowie et al., 2015; Zhang et al., 2016; Menut, 2018; Tai et al., 2021). However, the shape factor of the distribution needs to be empirically determined and thus might

not capture the interannual variability and changes in climate. In this subsection, we examine the scale-dependence of our dust emission scheme and then propose an alternative approach, which is to derive a simple spatial map and upscale the spatial variability of dust emissions from low resolution ones to high resolution ones. A discussion about our proposed approach versus the more common Weibull approach is detailed in Sect. 5.4.


We first examine the scale dependence of our dust emission scheme. We achieve this by performing an area-weighted mean of all input meteorological and land-surface variables to various coarser resolutions. Starting from the native 0.5° × 0.625° resolution of MERRA-2, we regrid fields to 0.9° × 1.25°, 1.9° × 2.5°, and 4° × 5°. Then we use these input fields with our new scheme to compute

hourly dust emissions for 2006 across these four resolutions. We then compare the emission outputs across these resolutions.

We examine the global, regional, and grid-level scale dependence of our dust emission scheme in Fig. 8. At the regional level, Fig. 8a shows the unnormalized annual total emissions (in Tg yr⁻¹) of nine major dust source regions. The source regions are defined following Kok et al. (2021a) (see Fig. S7 and

Figs. 8c-d). The regional dust abundance in Fig. 8a is mainly consistent with the regional dust emissions in Kok et al. (2021b, see Fig. 2). The highest emissions occur over the Middle East/Central Asia, followed by the Sahel and northern Africa. Smaller emissions are over East Asia, North and South America, Australia, and southern Africa. The dust emission scheme also shows scale dependence across different resolutions. Some regions may have a sharper decrease in emissions from 0.9° × 1.25° to 1.9° × 2.5° (e.g.,



northeastern Africa), and some regions may have a sharper decrease from 1.9° × 2.5° to 4° × 5° (e.g.,
       Sahel). This difference is contingent upon the degree of smoothing of the input fields such as $u_{*s}$ at a
       particular resolution. The emissions will drastically drop when the local extrema of $u_{*s}$ are smoothed out
       and no longer can be represented at a particular resolution, and over different regions this cut off may
       occur at different resolutions depending on the spatial heterogeneity of the local-scale meteorological
fields. Nevertheless, in general, the coarser the resolution, the worse the model can represent the local
       variability of $u_{*s}$ and other input fields and subsequently the emissions, and thus the magnitudes of the
       emissions decrease with resolution. Figure 8a shows that the relative differences in regional emissions can
       be different in different resolutions (e.g., Sahel can have larger emission than the Middle East/Central
       Asia in the 4° × 5° simulation), which will subsequently affect the spatial variability of other major dust
cycle variables (such as DAOD or deposition). Therefore, it is always preferential for GCMs to simulate
       the dust cycle in high resolutions. Figure 8b shows the scale dependence of the global emissions, which
       also shows the same decrease in emissions from fine to coarse grid resolutions.
              We also examine the scale dependence of dust emissions at the grid level. Figures 8c and 8d show
       the spatial distributions of 0.5° × 0.625° and 1.9° × 2.5° unnormalized emissions (with a global total of
11700 Tg yr$^{-1}$ and 5450 Tg yr$^{-1}$, respectively). The 0.5° × 0.625° simulation shows a more detailed local
       spatial variability compared to the coarser 1.9° × 2.5° simulation. The 1.9° × 2.5° simulation fails to
       capture some of the high emission regimes in the 0.5° × 0.625° simulation such as over Taklamakan, and
       fails to simulate the local emission peaks over marginal sources such as the Chihuahuan Desert and
       Patagonia. It can be seen that coarse-gridded simulations lose emissions in both major and marginal dust
sources. We calculate in Fig. 8e the grid-by-grid ratio of unnormalized 0.5° × 0.625° emissions to 1.9° ×
       2.5° emissions to show the emissions missed by the 1.9° × 2.5° simulation in each grid. Both the coarse
       (y-axis) and fine (x-axis) emissions are plotted in the $\log_{10}$ scale to show the emission ratios in terms of
       the order of magnitude only. It can be seen that most of the points are above the 1:1 line (black dashed
       line), meaning the coarse emissions are underestimating some local dust fluxes accounted by the fine
emissions. The low-resolution simulation can capture the largest emissions well (top right hand corner),
       as predicted by the high-resolution simulation. However, the smaller the emission, the more significant
       the difference in emissions, and for minimal emissions (< 10$^{-5}$ kg m$^{-2}$ yr$^{-1}$) the difference can be up to three
       orders of magnitude. There are very few grids with low-resolution emissions higher than high-resolution
       emissions. Those are exceptional cases due to the spatial variability of moisture $w$ more heterogeneous
than that of $u_{*s}$, leading to the smoothing of local maxima of $u_{*ft}$ instead of $u_{*s}$ and thus the
       overestimation of $F_d$ in the coarser simulation. In conclusion, the coarse resolution models omit many
       local input features and thus fail to represent the correct spatial variability of dust emissions.

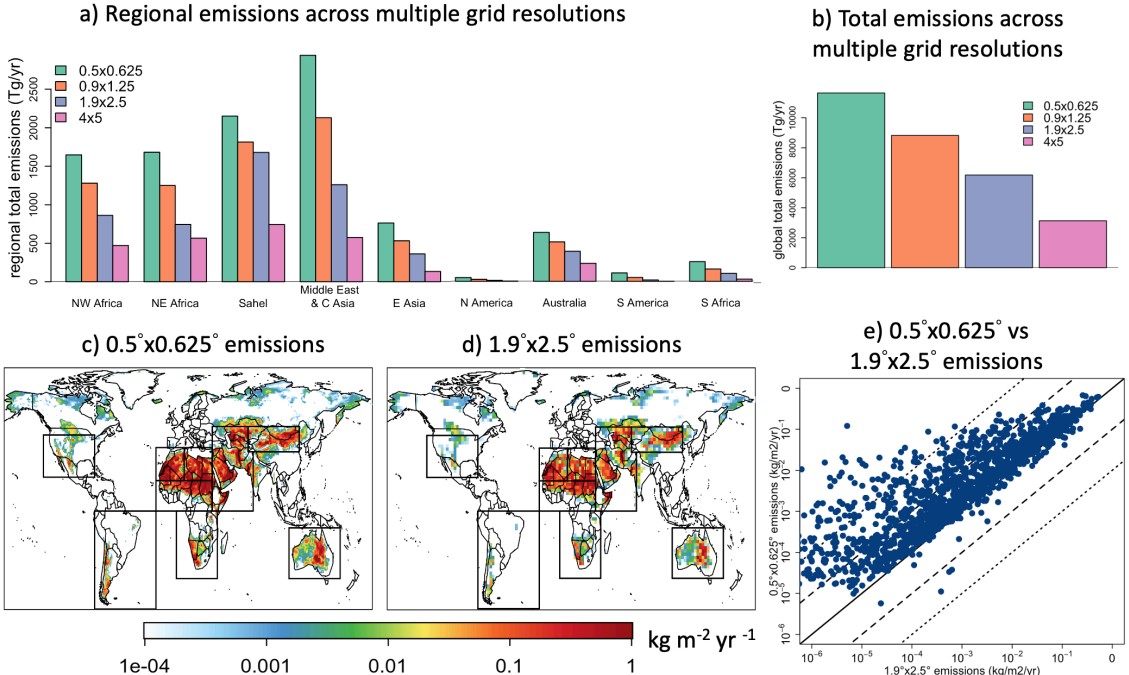


Figure 8. The dependence on horizontal resolution of dust emissions simulated with our new dust emission scheme. (a, b) Bar plots of dust emissions as a function of grid resolutions for (a) nine major dust emission regions and (b) the globe. (c) 0.5° × 0.625° unnormalized emissions and (d) 1.9° × 2.5° unnormalized emissions. e) A scatterplot of 0.5° × 0.625° versus 1.9° × 2.5° unnormalized dust emissions. The rectangular boxes show the nine source regions in (a) defined as in Fig. S7.


To mitigate the scale dependence of our dust emission simulations, here we propose a simple approach to upscale the simulated low-resolution emissions to match the high-resolution emissions. This approach assumes that the fine emissions have a more adequate magnitude and spatiaotemporal variability
than the coarse emissions so that the coarse emissions are calibrated to match the fine emissions. Our approach is that, by dividing the normalized high-resolution emission map by the low-resolution emission maps, we obtain a map of scaling factors to account for the differences in the spatial variability of dust emissions between high- and low-resolution simulations. We obtain a climatological map of correction factors $\widetilde{K}_c$, and apply this map to all simulated coarse emissions from both historical and future simulations,
correcting their spatial variability to match that of the high-resolution emissions. We show the seasonal variability of $\widetilde{K}_c$ in Fig. S8. This map of scaling factors contains some temporal and seasonal variability, so it would be preferable to apply a seasonally varying dataset of $\widetilde{K}_c$. But, as shown in Fig. S8, $\widetilde{K}_c$ varies modestly across season because the subgrid variability of the meteorological and land-surface fields are partly determined by spatial structures such as orography or land use/land cover, which change across a
relatively long enough time scale (decadel to multidecadal or longer). We note that theoretically this approach will fail to work if, over the remote regions, the low-resolution emission is zero throughout the entire simulation period while the high-reoslution emission is a small positive definite, since $\widetilde{K}_c$ will go to infinity. We also note that employing different input fields or different emission schemes will change the subgrid variability and thus the spatial representation of this correction map. For instance, one will obtain
a *slightly* different correction map if one uses ERA-Interim meteorology instead of MERRA-2, or a





*moderately* different map if one employs the Z03 or any other dust emission equations instead of K14 or our new scheme. Therefore, although here we present a standard correction map which is likely accurate and realistic, we suggest each model should make their own correction maps for their specific model configurations. We discuss more caveats of this approach in Sect. 5.4.

Figure 9 shows the ratio maps normalized fine emissions to coarse emissions. Figure 9a shows the ratio of normalized $0.5° \times 0.625°$ emissions ($F_{d,0.5}$) to constrained $1.9° \times 2.5°$ emissions ($F_{d,2}$), and Figure 9b shows the ratio of constrained $F_{d,0.5}$ to constrained $0.9° \times 1.25°$ emissions ($F_{d,1}$). Since all emissions are constrained to match the global dust budget, the correction maps display the relative changes in spatial variability between two resolutions. From both Figs. 9a and 9b, it can be seen that $0.5° \times 0.625°$ emissions

tend to generate relatively fewer emissions (blue color) over the major sources, such as the Sahara, the Arabian Desert, and the Taklamakan Desert. $0.5° \times 0.625°$ emissions also tend to have relatively more emissions (red color) over the peripheries of the major sources, such as Algeria, Yemen, and the peripheries of Taklamakan. This is in line with the above discussion, because the high-resolution simulations tend to be more capable of representing the local peaks of $u_{*s}$ and therefore can more likely

pass the thresholds and produce $F_d$; the low-resolution simulations would miss a lot of marginal emissions because the low-resolution wind will be smaller than the low-resolution threshold and yield a zero $F_d$. The ratios over the marginal sources can be up to 100 or even more because of the much smaller emissions in low-resolution simulations. The contrast is smaller (with ratios of 0.5–0.9) over major sources where coarser simulations are more capable of representing the large-scale emission fluxes. Due to the same

reason, the correction values in Fig. 9a for $1.9° \times 2.5°$ ($\widetilde{K}_{c,2}$) generally have bigger magnitudes than those in Fig. 9a for $0.9° \times 1.25°$ ($\widetilde{K}_{c,1}$). These maps indicate that in coarse-gridded simulations, dust emissions are overall underrepresented over marginal sources and overrepresented over major sources. Therefore, we propose implementing these maps into the coarser GCMs to correct the dust emission spatial variability accordingly.


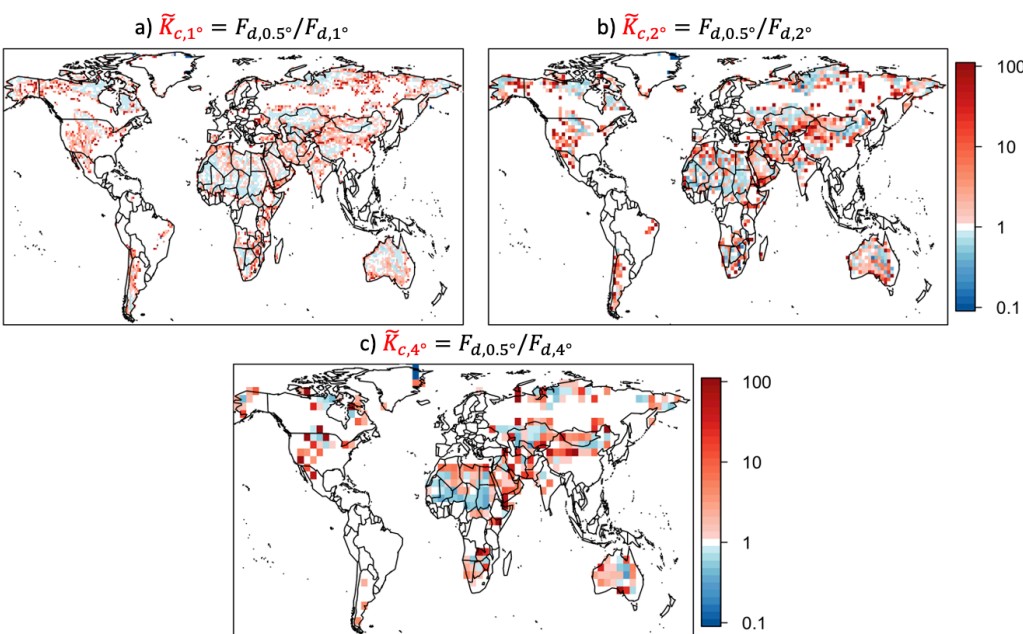

Figure 9. Gridded maps of a $0.9° \times 1.25°$, $1.9° \times 2.5°$, and $4° \times 5°$ scaling factor to rescale the coarse dust emission simulations to match the spatial variability of high-resolution dust emissions. This is achieved



by calculating the ratios of (a) 0.5° × 0.625° to 1.9° × 2.5° emissions, (b) 0.5° × 0.625° to 0.9° × 1.25° emissions, and (c) 0.5° × 0.625° to 4° × 5° emissions. Emissions of all resolutions are constrained and normalized to have a global total of 5000 Tg yr$^{-1}$ before calculating the ratios.


### 4.3 Comparison of our emission against other emissions estimates

To validate the emissions produced by our new scheme, this subsection focuses on comparing the resulting 0.5° × 0.625° emissions from our new scheme against other existing emission data. Since there
are no globally gridded observations of dust emissions, GCMs and ESMs mostly evaluate their schemes using observable atmospheric dust products such as satellite and ground-based DAOD data, as well as dust surface concentrations and dust deposition flux measurements (Ridley et al., 2012; Kok et al., 2014b; Pu and Ginoux, 2018; Parajuli et al., 2019; Klose et al., 2021). Since this study focuses on simulating dust emissions, not their subsequent transport and deposition, we compare our results against constraints on
the fraction of annual dust emission contributed by nine major source regions (see Fig. 10a and Table 1 in Kok et al., 2021b). These constraints on regional emission fluxes were obtained from the Dust Constraints from joint Observational-Modelling-Experimental analysis (DustCOMM) data set (Kok et al., 2021a, b), which used inverse modeling to combine an ensemble of model simulations of the global dust cycle with constraints on the regional DAOD near major dust source regions (Ridley et al., 2016), the dust size
distribution (Adebiyi and Kok, 2020), and dust extinction efficiency (Kok et al., 2017). The DustCOMM constraints on regional dust emissions include error estimates that account for the spread in model results and the uncertainties in the constraints on dust properties and abundance. Comparisons against independent measurements of dust surface concentrations and deposition fluxes indicated that the DustCOMM product is more accurate than GCM simulations and the MERRA-2 dust reanalysis, and that
uncertainties are realistic (Kok et al., 2021a, b). The total emissions from all nine source regions obtained by the DustCOMM dataset was 4.7 (3.4–9.1) × 10$^3$ Tg/year for dust PM$_{20}$. The global total of ~4700 Tg yr$^{-1}$ is close to the 5000 Tg yr$^{-1}$ we adopted in this study for normalization, and we again normalized the DustCOMM global emissions to 5000 Tg yr$^{-1}$. Note that Kok et al. (2021a, b) only constrained the emissions and other dust variables for each broad region, but its subregional spatial distribution of dust is
still a multimodel mean.

Fig. 10a shows the gridded global spatial distribution of DustCOMM dust emissions (Kok et al., 2021a). Here we compare the gridded simulations of K14 (experiment I; Fig. 7a) and our scheme (experiment V; Fig. 7e) against the gridded K21 DustCOMM emissions. Our new scheme's simulation in
Fig. 7e successfully captures most of the major peaks in DustCOMM emissions, except that there are more northern U.S. and high-latitude emissions in our new scheme which was not represented and constrained in DustCOMM's inverse analysis. Our scheme and DustCOMM emissions have a gridded spatial correlation coefficient of $r = 0.71$, showing the resemblance of the two emission maps and our scheme's ability in physically capturing the emission peaks. On the other hand, K14 emissions (Fig. 7a) also share
a similar spatial distribution with DustCOMM emissions but show more emissions over central Africa, central India, and northern Australia. Its gridded spatial correlation coefficient with DustCOMM is $r = 0.61$, indicating it does not match DustCOMM emissions in spatial variability as well as our scheme does.

We also conducted simulations using the Z03 scheme (Eq. 10) for comparison with our scheme's simulation. The Z03 scheme requires a source function $S$, and in this study we adopted two popular source
functions: one is the Zender et al. (2003b) geomorphological source function (e.g., used in CESM; Oleson et al., 2013) and the other one is the Ginoux et al. (2001) source function (e.g., used in GEOS-Chem; Fairlie et al., 2007). Both source functions are plotted in Fig. 2 in Kok et al. (2014b). Fig. S9a shows the simulations of the Z03 scheme with Ginoux et al. (2001) $S$ (Z03–G), which shows almost an identical



pattern with the Z03–G scheme in the GEOS-Chem simulations (e.g., top panel of Fig. 1 in Meng et al.,
2021). The Z03–G scheme is mostly consistent with the DustCOMM multimodel emissions ($r = 0.57$),
but captures more African emissions in the northwest Africa such as Algeria, Morocco, and Western
Sahara. On the other hand, the Z03–Z scheme employs a geomorphic $S$ that possesses a spatial distribution
of the upstream area where surface runoff is collected, from Zender et al. (2003b). Fig. S9b shows the
simulations of the Z03–Z scheme, which shows a highly similar pattern with the Z03–Z scheme in the
CESM simulations (e.g., Fig. 2a in Li et al., 2022). It captures the emission peaks across the globe but is
quite spatially heterogenous, yielding a relatively low $r = 0.35$ with DustCOMM.

In Fig. 10b, we compare the simulated dust emissions summed over each of nine source regions
(rectangular boxes in Fig. 10a) against the corresponding DustCOMM emissions. We also summed the
emissions outside all rectangular boxes as the high-latitude emissions, to yield the ten data points in Fig.
10b. From Fig. 7e, high-latitude emissions from our scheme mainly includes emissions from Alaska,
Canada, Greenland, and Iceland, and there are no emissions from Antarctica because of the lack of
necessary input data there (e.g., soil texture and roughness due to rocks). However, since K21 does not
provide emission estimates outside of the nine defined source regions, we compare against the estimate of
high-latitude dust emissions from Bullard et al. (2016) (B16) obtained from GCM results. Their definition
of high-latitude emission not only includes emissions from the abovementioned regions but also from
Patagonia, so we add Patagonia (39°–56°S of S America) emissions as part of the high-latitude emissions
in Fig. 10b. Bullard et al. (2016) estimated that high-latitude emissions (without error estimates) accounted
for 4-5 % of their assumed 2000 Tg/yr of global total emissions. We normalized their estimate to match
our global total of 5000 Tg yr$^{-1}$, yielding a high-latitude emission range of 200–250 Tg yr$^{-1}$. We take the
mean value which is 225 Tg yr$^{-1}$.
It can be seen from Fig. 10b that the results with our scheme (blue color) better match these
estimates of regional emissions than results with the K14 scheme, lying substantially closer to the 1:1
(black) line over most regions including Africa, Asia, and Australia. There are two notable exceptions,
namely North America and the high-latitude emissions. Our scheme generates fewer dust emissions over
the Mojave–Sonoran–Chihuahuan over the U.S.–Mexico border compared to the K14 emissions (Fig. 7a),
because of the high LAI (annual mean > 0.4; Fig. 3a) over the western U.S that leads to the strong wind
drag partitioning. Meanwhile, for the rest of the world, our scheme generates a significant amount of high-
latitude emissions over the Arctic, which were not captured by K14 emissions due to the very high $u_{*ft}$.
Using the emission intermittency parameterization, our scheme represents one of the earliest attempts of
successfully capturing a significant amount of high-latitude emissions. Our high-latitude emissions
account for 262 Tg yr$^{-1}$ (without Patagonia) and 308 Tg yr$^{-1}$ (with Patagonia), in total accounting for 5–6
% of a global sum of 5000 Tg yr$^{-1}$ which is very close to what B16 suggested. Our scheme has an R$^2$ of
89 % and an RMSE of 141 Tg yr$^{-1}$, which is substantially better than K14's performance with an R$^2$ of 65
% and an RMSE = 259 Tg yr$^{-1}$. On the other hand, the Z03–Z scheme has a similar level of performance
compared with K14, with a higher RMSE of 317 Tg yr$^{-1}$ and an R$^2$ of 64 %. The Z03–G simulation has a
higher R$^2$ of 83 % and also a smaller RMSE of 237 Tg yr$^{-1}$ against DustCOMM compared with K14 and
Z03–Z. In conclusion, our scheme outperforms all the aforementioned simulations in terms of matching
against the DustCOMM estimates of regional dust emissions.






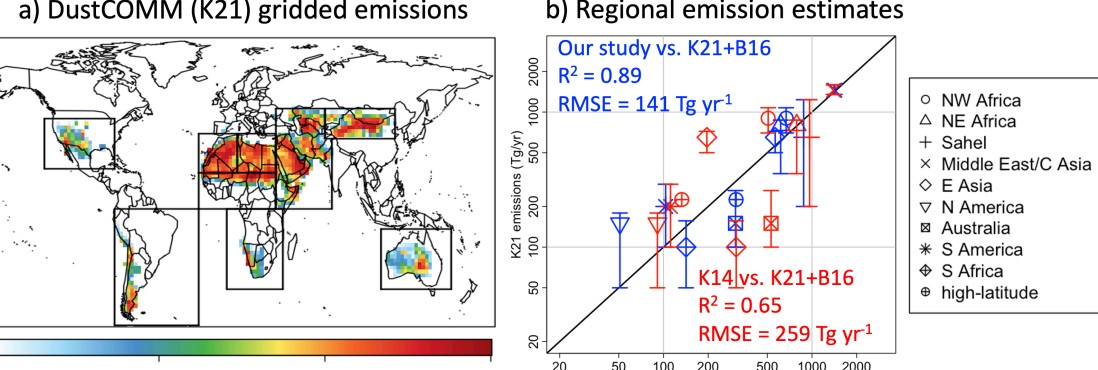

Figure 10. Dust emissions simulated using our new scheme and the older Kok et al. (2014b) scheme compared against DustCOMM (K21) and Bullard et al. (2016) constraints on regional dust emissions. (a) Globally gridded DustCOMM emissions (kg m$^{-2}$ yr$^{-1}$) based on emissions from six different models that were adjusted using inverse modeling to match constraints on particle size distribution, extinction efficiency, and regional dust aerosol optical depth. The black rectangular regions specify the nine source regions defined by Kok et al. (2021a) as also shown in Fig. 8c-d. (b) DustCOMM regional emissions (based on fractional emissions reported in the fifth column of Table 1 in K21b scaled to a global total of 5000 Tg yr$^{-1}$) versus the regional emissions computed by the K14 scheme (Fig. 7a) and our new scheme (Fig. 7e). The regional emissions are obtained from DustCOMM following the nine source regions in panel (a), with one extra data point representing the "high-latitude" emissions estimated by Bullard et al. (2016). The error bars show one standard error, except that the B16 high-latitude emission does not include an error estimate. The black line shows the 1:1 line.


## 5. Discussion of the caveats and limitations of the new parameterization

Because of the complexities of simulating the fine-scale process of dust emission in a large-scale gridded model, the parameterizations of dust emission processes presented in Sect. 4 and the dust schemes employed from all past studies necessarily made a number of assumptions. Below, we discuss the most important assumptions and the corresponding limitations.

**5.1. Soil median diameter representation**

For our derived parameterization for the dry soil median diameter $\overline{D}_p$, we obtained a relationship between $\overline{D}_p$ and silt+clay fraction for non-arid regions (LAI > 1), and a constant of $\overline{D}_{p0} = 127 \pm 47$ $\mu$m for arid regions (LAI < 1). In theory, $\overline{D}_p$ should be a function of soil properties (Hillel, 1980) and therefore implicitly of space and time, but we obtained a simple relationship for $\overline{D}_p$ over arid regions because 1)
different studies provided measurements of different soil properties such that data are limited and insufficient for a more detailed statistical analysis, 2) the uncertainties of the measured soil PSDs are moderately large, and 3) most $\overline{D}_p$ measurements over arid regions found $\overline{D}_p$ within 40–250 $\mu$m (Fig. 1c), which limited the dry fluid threshold $u_{*ft0}$ to vary within the relatively small range of ~0.204–0.268 m s$^{-1}$ (from Eq. 2 assuming $\rho_a = 1.225$ kg m$^{-3}$). This range is about ten times smaller than the global range





of the wet fluid threshold $u_{*ft}$ (~ 1 m s⁻¹). Thus, using a single $\overline{D}_{p0} \sim 127 \ \mu m$ across all arid regions appears to be a reasonable approach given the current data availability.

A number of previous studies have also compiled their global $\overline{D}_p$ maps for calculating the dust emission threshold (Laurent et al., 2008; Tegen et al., 2002; Menut et al., 2013). These studies used global soil texture data (with known fractions $f_a$ of soil components including sand, silt, and clay) to determine

gridded soil types using the soil texture triangle diagram (e.g., FAO/IIASA/ISRIC/ISS–CAS/JRC, 2012; Chatenet et al., 1996; Shirazi and Boersma, 1984), and calculated gridded $\overline{D}_p$ maps by combining the soil component fractions $f_a$ with the estimated aggregated particle size $\overline{D}_a$ of the soil components by a weighted mean $\overline{D}_p = \sum_a f_a \overline{D}_a$ (Menut et al., 2013). Since these $\overline{D}_a$ values are based on measurements from Chatenet et al. (1996), these studies also took into account the aggregated sizes of the individual soil

components. However, owing to insufficient data worldwide, these maps have not been verified against the measurements of the in-situ soil PSD. On the other hand, our results based on observations suggest that the spatial variability of $\overline{D}_p$ over deserts are relatively small compared to $\overline{D}_p$ over nonarid regions, and does not significantly correlate with soil texture and soil properties (see Eq. 14). Our results thus surprisingly suggest that the $\overline{D}_p$ parameterization can reasonably be much simplified into a global constant

over arid regions. This approach is consistent with another current approach of employing a globally constant soil particle diameter (e.g., Zender et al., 2003a; Mahowald et al., 2006) and significantly different from another approach of employing a $\overline{D}_p$ map (e.g., see Fig. S10 for the input $\overline{D}_p$ map for the CHIMERE CTM; Mailler et al., 2017; Menut, 2018; Menut et al., 2021). Furthermore, the $\overline{D}_p$ measurements showed a positive relationship with the silt+clay fraction $f_{silt+clay}$ over the non-arid

regions, which we attributed to the increasing cohesion between soil particles with the increasing silt+clay content. This is different from the negative $\overline{D}_p - f_{silt+clay}$ relationship assumed by most past studies (Tegen et al., 2002; Laurent et al., 2008; Menut et al., 2013) as they assumed that $\overline{D}_p$ increased with sand content ($f_{sand} = 1 - f_{silt+clay}$). We argue that our $\overline{D}_p - f_{silt+clay}$ relationships are based on observations and should thus represent an improvement over the past assumed $\overline{D}_p$–texture relationships. We anticipate that as more

measurements emerge in the future, more statistical or machine learning modeling approaches can more robustly decipher the intricate relationships between the aggregated $\overline{D}_p$ and the soil properties in order to further improve the representation of the global $\overline{D}_p$ map.

**5.2. Hybrid drag partition scheme**

For our drag partition scheme, we combined the drag partition for rocks ($f_{eff,r}$) from the Marticorena and Bergametti (1995) scheme with the drag partition for vegetation ($f_{eff,v}$) from the Okin (2008) scheme using a weighted mean approach employing the land cover fractions of rocks and vegetation, which is a novel approach. Nonetheless, our scheme still contains several major limitations.

First and foremost, we calculate the rock $f_{eff,r}$ using the Prigent et al. (2005) aeolian roughness length data derived from the ERS microwave sensor, but we could not completely separate rock roughness from vegetation roughness in the dataset. Marticorena et al. (2006) argued that ERS sensor measurements has relative small local incident angles such that the contribution from vegetation roughness is relatively small compared to the contribution from rock roughness. We also removed as much vegetation influence as

possible on $z_{0a}$ by taking the minimum $z_{0a}$ out of the 12 available monthly data for all grids, but it is still possible that a small fraction of vegetation roughness remains. Thus, our approach of combining the time-invariant Pr05 $z_{0a}$ (and $f_{eff,r}$) with the time-varying $f_{eff,v}$ will probably result in a small degree of double-counting of vegetation roughness in our drag partition scheme. We also note that the Pr05 data used the microwave backscatter coefficient as a proxy for inferring $z_{0a}$, but it does not perfectly correlate



with the $z_{0a}$ measured by the ground sites, and so there will be some corresponding errors in the Pr05 $z_{0a}$
used in our drag partition scheme.

Second, the land cover fraction areas used are obtained from the ESA CCI dataset in our drag partition scheme, which are annual mean and thus not season-dependent. The land cover of vegetation $A_v$ in Eq. 21b should also be a function of time mainly because of the seasonally changing vegetation cover.

However, most current land cover datasets provide annual data but not seasonal data, and thus $A_r$ and $A_v$ in Eq. 21b are only spatially varying within the year 2006. Furthermore, most land cover datasets only provide near-term land cover data (the farthest data goes back to the year 850; e.g., Klein Goldewijk et al., 2017). ESA CCI has a temporal coverage of 1992–2019, and so the dataset will represent relatively well for present-day scenarios but become less representative of paleoclimatological scenarios, for which the

vegetation distributions differ from that in the present day, such as during the Green Sahara > 6000 years ago (Kutzbach et al., 1996; Bonan, 2015).

A third limitation of our new drag partition scheme is that the Okin (2008) drag partition scheme requires the vegetation cover fraction $f_v$ as a proxy for vegetation density. In this study, we proposed LAI ~ $f_v = 1 - f_{bare}$ as discussed in Eq. 18b. Since LSMs do not have accurate simulations of $f_v$, there

are only preliminary equations for calculating the bare and vegetated fractions proposed by previous papers (e.g., Mahowald et al., 2006; see discussions in Eq. 11 and Eq. 18b). However, Eq. 11 (and thus $f_v$ ~ LAI) only applies to regions with low LAI because leaves are mostly not overlapped; over regions with higher LAI, leaves start to overlap and LAI > $f_v$. Thus, this assumption could overestimate $f_v$ and thus $f_{eff,v}$, thereby underestimating dust emissions over vegetated regions (e.g., underestimations over western

U.S. deserts in Fig. 7e). Thus, LSMs need a more accurate parameterization of $f_v$ to get a more accurate vegetation drag partition effect regardless of whether the Okin (2008) or the Raupach et al. (1993) scheme is used. Alternatively, LSMs could also read in observed $f_v$ from available satellite-derived products such as MODIS or AVHRR $f_v$, as done by Wu et al. (2016) for instance. We used the observed/modeled LAI instead of observed $f_v$, because LSMs actively simulate LAI, allowing the scheme to be used in past and

future climates when vegetation cover observations are not available.

A fourth important limitation is that O08 does not fundamentally distinguish the drag partitioning between different plant functional types (PFTs). O08 assumes that all short plants are hemispheric in shape and produce the same $f_{eff,v}$ if they have the same $f_v$ or LAI. In reality, short plants such as shrubs, herbaceous plants, and crops have vastly different shapes; some are far from hemispheric and can hardly

be approximated by a simple geometry or shape. Nonetheless, this limitation is not unique to O08 but shared by other drag partition schemes such as Raupach et al. (1993), as their $f_{eff,v}$ equations are also functions of $f_v$ only and not functions of PFTs. More research is thus warranted in the future to better quantify the plant shapes and $f_{eff,v}$ of different PFTs.


### 5.3. Dust emission intermittency scheme

There are two important caveats about the Comola et al. (2019b) intermittency scheme. The first caveat is that it has exponential dependences on $u_{*s}$, $\sigma_{\tilde{u}_s}$, $u_{*it}$, and $u_{*ft}$ (see Sect. S2) and is thus very sensitive and vulnerable to the accuracy of the GCM simulations of the four variables. For instance, if the

thresholds are overestimated by the threshold schemes, not only will emissions be underestimated but $\eta$ from C19 will also be close to zero and further worsen the low bias of the dust emissions. Therefore, a prerequisite of employing the C19 scheme is that the wind $u_{*s}$ and the thresholds should be adequately simulated and have reasonable ranges of variability throughout the year. If simulated well, $\eta$ will have reasonable day-to-day and seasonal fluctuations. Otherwise, $\eta$ can constantly fall on one or zero and

become unindicative of the boundary-layer dynamics temporal variability, and the resulting $\eta$ time series will further impact and worsen the temporal variability of the $F_{d,\eta}$ time series.





The second caveat is that Comola's theory allows turbulence to generate instantaneous winds $\tilde{u}_s$ that exceed the impact and fluid thresholds and generate some emissions, even when the averaged $u_{*s} < u_{*it}$ across the model timestep. Thus, C19 allows $\eta > 0$ (per Eq. 22c) when $u_{*s} < u_{*it}$. However, the C19 parameterization as such still depends on conventional dust emission equations such as Kok et al. (2014b) or Zender et al. (2003a), which prohibit emissions when the mean $u_{*s} < u_{*it}$ within a model timestep. Thus, the C19 theory and its dust emission parameterization contain an internal logical inconsistency, and the C19 scheme per se still does not generate emissions $F_{d,\eta}$ (Eq. 22b) for $u_{*s} < u_{*it}$ ($\eta > 0$ per Eq. 22c), but $F_{d,\eta} = \eta F_d = 0$ since $F_d = 0$ per Eq. 22a). But, because the turbulent emissions in the $u_{*s} < u_{*it}$ regime is small, the C19 formulation is still a good approximation for turbulent emissions, performing much better than the conventional timestep-constant models as demonstrated in Comola et al. (2019b).

**5.4 Reducing the grid scale-dependence of dust emission simulations**

We produced the correction maps to scale the spatial variability of the low-resolution dust emission simulations, matching the spatial variability of the high-resolution emissions (Fig. 9) to reduce the scale-dependence of dust emission simulations. As such, it is an alternative to the computationally expensive but more fundamental solution of simulating dust emissions in the highest model resolution possible and then regrid to coarser resolution. Employing the scaling map $\widetilde{K}_c$ is different from the fundamental solution in the sense that the maps in Fig. 9 are time-independent and derived by matching the annual total high-resolution emissions (Sect. 4.2). As seen in Fig. S8, the scaling map exhibits a moderate degree of seasonality, but employing an annual scaling map like Fig. 9 will already address a large part of the scale-dependence problem. We suggest GCMs and CTMs, which focus on present-day simulations, perform multiyear simulations in both high and native grid resolutions to obtain monthly climatological maps of scaling factors for present-day dust emission simulations. Afterwards, ESMs only need to read in the climatological monthly scaling maps to rescale the native grid dust emissions every month before passing the dust emissions to the atmospheric model component. If desired, instead of generating climatological maps, models can even choose to obtain transient (e.g., monthly) $\widetilde{K}_c$ maps as a time series for the past decades of the historical period (e.g., 1980–2020) so that the scaling maps contain a much better temporal variability in terms of seasonality, interannual and decadal variability, of the historical dust emissions than compared with the climatological scaling maps.

Our proposed approach is an alternative to a more common approach, which is to employ a Weibull distribution to the GCM winds (Cakmur et al., 2004; Grini et al., 2005; Cowie et al., 2015; Zhang et al., 2016; Menut, 2018; Tai et al., 2021). In the Weibull approach, in each time step the model assumes a Weibull PDF for each grid that is characterized by the modeled mean wind speed and a shape parameter $k$ representing the subgrid wind variability. $k$ could be a global constant (e.g., k = 4 in Menut, 2018), a parameterized function of the model mean wind speed (e.g., Grini et al., 2005), or a globally gridded map obtained by comparing coarse winds against high-resolution winds (Ridley et al., 2013; Tai et al., 2021). A distinction between our approach and the Weibull approach is that while previous studies derived the shape parameter by comparing high- and low-resolution winds (e.g., Tai et al., 2021), we directly make use of the high- and low-resolution input fields to calculate dust emissions (Fig. 8c and d) and then compare between the high- and low-resolution emissions. Therefore, the correction map $\widetilde{K}_c$ (Fig. 9) captures subgrid wind variability due to subgrid spatial characteristics just like $k$ in the Weibull approach, and thus we anticipate some intrinsic spatial correlations between the $\widetilde{K}_c$ and $k$ maps.

However, there are three distinctions and advantages of our approach over the Weibull approach. The first and the most important one is that our approach accounts for not only the subgrid variability of wind but all other fields ($u_{*s}$, $w$, LAI, etc). $\widetilde{K}_c$ is obtained via comparing emissions across resolutions and thus its magnitude is a result of the subgrid variability of all input fields, whereas in Tai et al. (2021) the



$k$ is obtained via comparing winds across resolutions and thus only captures subgrid wind variability. The second advantage is that our approach makes no assumption about the distributions of the subgrid variability of forcings. The Weibull approach is a parametric statistical method, which means the dust models need to assume the subgrid winds follow a Weibull distribution, but the subgrid spatial wind variability can deviate substantially from a Weibull distribution due to several reasons, such as complex

terrain (Jiménez et al., 2011). Our method is non-parametric and thus more robust in capturing subgrid variability of multiple input fields at once. The third advantage is that our approach saves computational cost because it only needs to (1) find the $\widetilde{K}_c$ map and (2) directly apply $\widetilde{K}_c$ to correct the coarse emissions. Meanwhile, past studies needed to (1) find the $k$ map, (2) iteratively calculate emissions by looping across the Weibull PDF (3) sum across the PDF to yield the total emission, and (4) update the Weibull PDF for

each grid and time step. Repeating step (2) for all times and grids is computationally very expensive.

     There is yet another alternative approach to account for subgrid variability of winds and other parameters. Some CTM studies (e.g., Meng et al., 2021) proposed to simulate dust emissions at the highest resolution possible and then store the results as a gridded emission inventory. Meng et al. (2021) proposed that CTMs do not need to simulate dust emissions and instead only need to regrid the stored gridded dust

emissions to the desired grid resolutions. An advantage of their approach is clearly that their regridded dust emissions will have the correct spatial and temporal correlations with their high-resolution dust emission inventory, which means there will be no grid scale-dependence problem in their approach. Their approach also saves time and computational resources because they do not simulate but just read in and regrid dust emissions. This approach is particularly favorable for the CTM simulations for air quality

forecasts and hindcasts in which models need to ensure the near-term input meteorological fields are very accurate to generate an accurate dust emission inventory, such as the air quality forecasts conducted by the Environmental Protection Agency (EPA) using CMAQ (Appel et al., 2013, 2017), or other CTM studies such as GEOS-Chem (Zhang et al., 2013; Meng et al., 2021). However, for GCMs/ESMs that care about long-term simulations and aerosol–climate interactions, there is a need to actively simulate dust

emissions and allow a full coupling between meteorology and dust, which could not be achieved by feeding the models with an inventory. In that case, our scaling method is likely more desirable for GCMs and ESMs to reduce the scale-dependence problem.

     Also, we note that although our approach alleviated the grid-scale dependence of dust emissions, the grid-scale dependence problem also appears in other component of dust cycle simulations, such as in

dust transport. The grid-scale dependence can be due to not just the averaging problem but also other problems, such as numerical diffusion which worsens with increasing grid size in an Eulerian GCM (Rastigejev et al., 2010; Eastham and Jacob, 2017; Zhuang et al., 2018). It has to be solved in the atmospheric model component such as by some adaptive mesh refinement approaches (e.g., Semakin and Rastigejev, 2016) or machine learning methods (e.g., Zhuang et al., 2021).


### 5.5. Dust emission simulations

     Last but not least, all maps produced in this paper depend on the accuracy of the representations of the input meteorological fields and land-surface variables in various datasets. Our results are

particularly sensitive to the soil moisture simulated by MERRA-2 mainly because dust schemes are very sensitive to soil moisture. As a result, the final simulation of our scheme (expt. V; Fig. 7e) outperforms other dust emission schemes employed in this paper (Fig. 10 and Fig. S9), yet some features in the map are not ideal. First, for instance, the Australian dust emissions are of a comparable order of magnitude to the East Asian emissions (and even larger than East Asian dust emissions in coarser resolutions, per Fig.

8a), which might be because the soil moisture over Australia is slightly underestimated by MERRA-2. Second, northeastern China in Fig. 7e has more emissions than northwestern Chinese deserts, and similar spatial variability is also seen in other past studies (e.g, Kok et al., 2014b), which might be due to the





stronger friction velocity over northeastern China (annual mean > 0.3 m s$^{-1}$ in MERRA-2) than over
northwestern China (annual mean ~0.2 m s$^{-1}$ in MERRA-2) in the GCMs or input MERRA-2 fields (see
Fig. S11 for the spatial distribution of MERRA-2 $u_*$ over northeastern vs. northwestern China). Third, our
scheme generates very high summertime emissions over Sudan and the Horn of Africa because of the very
high MERRA-2 $u_*$ (~ 1 m s$^{-1}$ in the summer), low soil moisture and aeolian roughness (see Fig. 7i).
However, the emission peak are not seen by the dust aerosol optical depth (DAOD) observations. There
might be several reason for the HOA emission peak: i) the input fields are biased over the HOA, e.g., ; ii)
some unknown mechanisms are responsible for suppressing the HOA dust and we need to implement into
the GCMs; iii) dust emissions elsewhere except the HOA are suppressed too much by the drag partitioning
and the intermittency effect.

**6. Conclusions and significance of our new parameterization**

This study presented a new desert dust emission scheme for GCMs and CTMs. The major advances
of our scheme compared with existing schemes are the following: 1) we obtained improved
parameterizations for several key aspects of dust emission, 2) these improved parameterizations were
informed by multiple observations that constrained critical parameters, and 3) we proposed a method to
reduce the grid resolution-dependence of the emission scheme that is a common problem to many other
existing schemes.

To achieve these advances, we have implemented the following modifications to the existing dust
emission scheme of Kok et al. (2014a, b): Our first improvement involved the use of soil particle size
distributions from multiple past studies to estimate and constrain the soil median diameter $\overline{D}_p$ as a critical
parameter that determines the dust emission thresholds $u_{*it}$ (impact) and $u_{*ft}$ (fluid). We found that over
the arid desert regions (LAI < 1), $\overline{D}_p$ can be approximated as a global constant of $127 \pm 47$ $\mu$m, and over
non-arid regions $\overline{D}_p$ increases linearly with silt and clay content. This finding indicates that past dust
modeling approaches which parameterized $\overline{D}_p$ as a function of soil types can be simplified.

Second, we presented a parameterization of the effects of surface rocks and vegetation on the wind
drag partition effects, which is not included by many of the current GCMs and CTMs. In particular, a
major advance of our drag partition scheme is that we propose a novel method to combine the effects of
rocks and vegetation by getting a weighted mean of both effects according to the globally gridded rock
and vegetation land-cover area fractions from land-cover datasets (e.g., Klein Goldewijk et al., 2016; ESA,
2017; Kobayashi et al., 2017). Many dust modeling studies only attempted to include the drag partition
effect of either one of these roughness elements, and this study represents one of the earliest attempts
along with a few other papers (e.g., Darmenova et al., 2009; Foroutan et al., 2017; Klose et al., 2021) to
combine and unify the effects on the wind partition of both kinds of roughness elements. Future work
should also account for the time-varying vegetation drag partition effect to further enhance the realistic
coupling of dust emissions to vegetation dynamics and variability.

Third, we incorporated the boundary-layer turbulence effects on dust emission intermittency by
coupling the intermittency scheme formulated by Comola et al. (2019b) to our Kok et al. (2014b) dust
emission schemes. The C19 scheme is formulated based on field measurements of simultaneous high-
frequency measurements of sand transport and the turbulent wind. This is one of the first studies to have
included the turbulence effects on dust emissions, amongst others which focused more on the turbulence
effects on convective dust emissions (e.g., Klose et al., 2014; Li et al., 2014). Including the turbulence-
driven intermittency effect is important for marginal dust emission sources where the wind speed is
normally below the fluid threshold, e.g., dust emissions from high-latitude regions. The C19 scheme also
allows dust emission physics to couple better with boundary-layer dynamics and variability, such that





simulated dust will have a day-to-day and seasonal variability that is physically linked to the characteristics of the turbulent boundary-layer.

        Fourth, we proposed a simple scaling method to reduce the inconsistencies in the spatial distributions of the high-resolution and low-resolution dust emission simulations within a LSM. We propose to rescale the low-resolution dust emissions to match the spatial variability of the high-resolution

emissions by comparing the spatial distributions of the high- and low-resolution dust emission maps, thereby obtaining a climatological map of scaling factors. The correction maps can thus be applied to other simulations of similar settings, e.g., experiments with the same meteorological and land-surface inputs but different sea/land ice, ocean, stratospheric, or plant physiological forcings. This approach can alleviate the long-standing problem of grid-resolution dependence and spatial distribution inconsistencies of dust

emissions across grid sizes among a GCM (e.g., Ridley et al., 2013). Although grid-scale dependence exists in most physical variables in the GCMs, dust emission is exceptionally vulnerable to the grid-resolution dependence problem because of the very strong nonlinearity (a power of 3–5 or more) of dust emissions to the meteorological fields.

        These new approaches act synergistically to improve dust emission modeling. Our new scheme's

dust emission simulation, driven by the MERRA-2 meteorological and land-surface fields, shows higher consistency with the Kok et al. (2021a, b) DustCOMM multimodel mean emissions (Fig. 10 and Fig. S9), which were observationally constrained by an inverse modeling approach and thus contain a realistic regional distribution of dust emissions. Our scheme shows the best agreement against the multimodel mean dust emissions in terms of regional characteristics with $R^2 = 89$ %, meanwhile other schemes, i.e.,

Kok et al. (2014a, b) and Zender et al. (2003a, b), respectively yielded $R^2 = 65$ % and the $R^2 = 64$ %. This indicates that adding the missing aeolian physics into the existing emission schemes is critical to correctly capturing the dust emission spatial variability, and that our scheme displayed almost identical regional charactistics as the inverted multimodel emissions. Our emission map also show more distinctively the major dust sources including the Bodélé Depression, El Djouf, the Arabian Desert, the Australian Desert,

and Patagonia. In our companion paper (Leung et al., in prep.), we will examine the dust cycle simulations of this newly proposed dust emission scheme in CESM and evaluate its performance with other dust cycle variables such as dust PM concentration, dust AOD, lifetime, and deposition flux.

        Finally, we note that although our scheme employs a specific emission threshold scheme (i.e., Shao and Lu, 2000) and a specific dust emission equation of Kok et al. (2014b), the modifications we

proposed could be applied to different dust emission schemes. For instance, one could use Iversen and White (1982) as a threshold scheme with our newly proposed soil median diameter $\overline{D}_p$. One could also use Ginoux et al. (2001), Tegen et al. (2002), or any other dust emission equation to combine with the Comola et al. (2019b) intermittency scheme and our hybrid drag partition scheme. Therefore, our formulation in this paper is highly versatile and adaptable to most of the existing GCMs and CTMs. As

such, the new dust emission parameterization presented here can improve the global dust cycle in most GCMs, ESMs, and CTMs.





**Acknowledgements**

D. M. Leung and J.F.K. are funded by the National Science Foundation (NSF) grants 1552519 and 1856389. L. L. and N. M. M. acknowledge support from the Department of Energy (DOE) DE-SC0021302 and the Earth Surface Mineral Dust Source Investigation (EMIT), a NASA Earth Ventures-Instrument (EVI-4) Mission. G. S. O. is funded by the Army Research Office award W911NF-21-1-0070. M. K. has received funding through the Helmholtz Association's Initiative and Networking Fund (grant agreement no. VH-NG-1533). C. P. G.-P. acknowledges support from the European Research Council under the European Union's Horizon 2020 research and innovation programme (grant n. 773051; FRAGMENT) and the AXA Research Fund (AXA Chair on Sand and Dust Storms based at the Barcelona Supercomputing Center). D. M. Leung and J. F. K. acknowledge the high-performance computing resources from Cheyenne and Casper provided by the National Center for Atmospheric Research (NCAR), sponsored by the NSF. D. M. Leung thanks Dr. Francesco Comola for the helpful comments on the implementation of the dust emission intermittency scheme.

**Author Contributions**

J. F. K. and D. M. Leung conceptualized the study. D. M. Leung performed the model development, conducted the simulations, analyzed the simulation results and conducted the evaluations. D. M. Leung wrote the original manuscript and plotted all figures under J. F. K.'s supervision. L. Li, G. S. O., C. P. G.-P. M. K., L. M., D. M. Lawrence, and N. M. M. assisted the conceptualization and model development. C. P. provided the roughness length satellite retrievals. M. C. assisted the implementation of the intermittency scheme. All authors contributed to the manuscript preparation, discussion and writing.

**Competing Interests**

The authors declare no competing interests.

**Data Availability**

MERRA-2 fields are available at https://disc.gsfc.nasa.gov/datasets?project=MERRA-2. SoilGrids soil texture and properties can be obtained from https://soilgrids.org/. ESA-CCI land cover can be obtained from https://www.esa-landcover-cci.org/?q=node/164. DustCOMM inverse analysis results are available at https://dustcomm.atmos.ucla.edu. The satellite-derived aeolian roughness data are available upon contacting Catherine Prigent.

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
