# Peer review of "A new process-based and scale-aware desert dust emission scheme for global climate models — Part I: description and evaluation against inverse modeling emissions"

_Atmospheric Chemistry and Physics, 2022_

## Author Comment (AC1)

Responses to Reviewers on "A new process-based and scale-respecting desert dust emission scheme for global climate models – Part I: description and evaluation against inverse modeling emissions" by Danny M. Leung et al. (MS No: acp-2022-719)

We thank the reviewers for their careful examinations and thoughtful comments. Our point-by-point responses are provided below. The reviewers' comments are *italicized*, and our new/modified text is highlighted in blue.

Response to Referee #1

*The paper titled "A new process-based and scale-respecting desert dust emission scheme for global climate models – Part I: description and evaluation against inverse modeling emissions" correctly identifies the known problems and challenges in current dust schemes used in GCM/RCM including the effect of threshold friction velocity u\*t, wind stress partitioning, boundary layer turbulence, and grid-size-dependency. I appreciate the author's effort in improving dust emission schemes. The proposed changes are reasonable and have some physical basis. The emission estimates are validated as well. The paper is overall well written. However, there are a few concerns that should be addressed before the paper is accepted.*
*The resulting emissions are validated using DUSTCOMM data which is reasonable. The author mentions that they plan to implement the code in CESM later, in which case, it should be possible to conduct more robust validation of the results, for example, using DOD data, dust concentrations, and deposition. However, I still think that additional validation of the proposed changes is necessary to justify the added changes.*
*The proposed changes are numerous, not to mention that the paper is too long. An ideal, effective dust model should be simple in description and should use commonly available input datasets. Such a complex treatment of parameterizations must be justified appropriately. A step-wise validation and analysis of each added term would justify the complexity of the model. In this context, I suggest addressing the following four major points:*

1. *\*Validation of threshold friction velocity: For example, threshold friction velocity can be compared with Pu and Ginoux, (2020) who retrieved u\*t from MODIS AOD data globally, which are publicly available:*
   *https://acp.copernicus.org/articles/20/55/2020/acp-20-55-2020.html*

We appreciate the reviewer's suggestion to compare our dust emission threshold simulation against the Pu et al. (2020) threshold derived from satellite data. We note that we did not propose any modification to the fluid threshold friction velocity $u_{*ft}$, and our $u_{*ft}$ would thus be similar to that computed by MERRA-2.

Pu et al. (2020) obtained their emission threshold for dust emission schemes by matching the frequency distribution of remotely sensed dust AOD to the frequency distribution of reanalyzed friction velocity $u_*$. We argue in this paper that dust emission equations should employ the impact threshold $u_{*it}$. So, given that the emission threshold defined by Pu et al. (2020), $u_{t,Pu}$, was obtained by matching dust AOD distributions against the distributions of wind speed $u$, $u_{t,Pu}$ should be related to the impact threshold $u_{it}$ and not the fluid threshold $u_{ft}$ (i.e., the moisture effect would be less relevant). However, in our paper's context, Pu et al. (2020) used wind speed $u$ instead of the soil surface wind speed $u_s$ ($= uF_{eff}$) in the analysis, and so $u_{t,Pu}$ captured a larger threshold that included the impact threshold $u_{it}$ as well as the drag partition effect $F_{eff}$ that inhibits saltation. To

make a fair comparison, we compare here $u_{t,Pu}$ from Pu et al. (2020) (with a DAOD threshold of 0.5) against our $u_{*it}$ divided by the drag partition factor, $u_{*it}/F_{eff}$, which is a higher threshold than $u_{*it}$. We used the log law of the wall to change $u_{*it}$ from a velocity scale to a velocity of $u_{it}$ at the 10 m to yield $u_{it}/F_{eff}$, since Pu et al. (2020) employed the 10-m wind speed to derive $u_{t,Pu}$.

[Figure]

a) Pu et al. (2020) threshold $u_{t,Pu}$  m s⁻¹
b) Our scheme's $u_{it}/F_{eff}$  m s⁻¹
c) $u_{t,Pu} - (u_{it}/F_{eff})$  m s⁻¹
d) $u_{t,Pu} / (u_{it}/F_{eff})$ in $\log_{10}$ scale

The top two panels show a) Pu et al. (2020) annual mean threshold ($u_{t,Pu}$) and b) our threshold of $u_{it}/F_{eff}$. Both maps share similar spatial variability and magnitudes over certain regions such as Africa and Australia. The bottom panels show the c) bias and d) ratio of the thresholds, showing that the larger discrepancies occur over East and Central Asia. E.g., for the Taklamakan Desert, $u_{t,Pu}$ is 2–3 m s⁻¹ higher than our threshold. In general, panel c shows that over most emission source regions, $u_{t,Pu}$ and our threshold are close to each other and differ by only ~1–2 m s⁻¹. $u_{t,Pu}$ tends to be smaller than our $u_{it}/F_{eff}$ in the downwind regions (e.g., the Sahel), which is possibly because the dust over downwind regions are more advected from the upwind source regions instead of emitted locally. We included this comparison in the supplementary Sect. S5 for reference and noted at the end of Sect. 4.3:

"To evaluate our simulations of the dust emission thresholds, we also compare our simulations of dust emission thresholds against observationally based threshold estimates from Pu et al. (2020). They compared reanalyzed wind speed distributions against observationally derived DAOD distributions to obtain a threshold wind speed for each gridbox that corresponds to a threshold DAOD value (e.g., 0.5 over arid regions and 0.05 over semiarid regions), above which is defined as a dust emission event. We show that our simulations of dust emission threshold overall match their derived threshold wind speed in magnitude and spatial variability (see Sect. S5 and Fig. S13)."

We note the caveat that $u_{t,Pu}$ was derived by using a dust AOD threshold as a proxy for defining an emission event, which will lead to a less reliable threshold wind speed in the

downwind regions, because there will be non-local dust advected from upwind regions instead of locally emitted dust that affects the dust AOD distribution of the downwind gridbox. The transport effect might cause a lower $u_{t,Pu}$ in the downwind regions, such as over the Sahel.

2.  *Since MERRA2 provides DOD as well, how about calculating the correlation between DOD and estimated emission flux for additional validation? Or how about comparing it with the dust emission fluxes from MERRA2 itself? Can we compare the estimates of dust emissions from this study with the MERRA estimates and show that the estimates from this study better correlate with DUSTCOMM compared to MERRA?*

That is another helpful suggestion. We think it is difficult to use observed dust AOD or MERRA-2 AOD data to evaluate emission data directly since their spatial distributions are by nature very distinct from each other. It would be better to compare emissions with emissions and AOD with AOD. But, it is a good idea to take MERRA-2 dust emissions and compare them directly against DustCOMM emissions. MERRA-2 uses GOCART dust, which means dust emissions are from the Ginoux et al. (2001) scheme. It is a great idea to include MERRA-2 in Sect. 4 for evaluations along with Zender's and Kok's schemes. We note that DustCOMM is compiled by assimilating observed dust AOD with data from several models including GOCART, so DustCOMM will share some spatial variability with MERRA-2 dust emissions. We downloaded MERRA-2 dust emissions and plotted them below:

[Figure]

Panel a shows 0.5° × 0.625° MERRA-2 global dust emissions, and panel b shows MERRA-2 versus DustCOMM regional emissions. MERRA-2 shares a high resemblance in spatial variability with DustCOMM ($R^2$ = 88 %), which is close to our scheme's performance ($R^2$ = 89 %). MERRA-2 has a slightly higher RMSE of 187 Tg yr$^{-1}$ than our scheme (RMSE = 141 Tg yr$^{-1}$). MERRA-2 is highly consistent with DustCOMM for most major sources. Compared to DustCOMM, MERRA-2 generally tends to underestimate emissions for smaller and secondary emission sources, such as East Asia, the United States, and Southern Hemisphere sources. We included the discussion on MERRA-2's emissions in Sect. 4.3, added the above plots to Fig. 10 and supplementary Fig. S12, and modified Sect. 4.3 in several paragraphs accordingly (see main text). For instance:

"… The Z03–G simulation (Fig. 10c) has a higher $R^2$ of 83 % and also a smaller RMSE of 237 Tg yr$^{-1}$ (NRMSE = 43 %) against DustCOMM compared with K14 and Z03–Z. Meanwhile, MERRA-2 (Fig. 10d) has a high regional correlation of $R^2$ of 88 % and RMSE of 187 Tg yr$^{-1}$ (NRMSE = 37 %) against DustCOMM regional variability. In conclusion, …"

3. *The relative importance of added terms: The proposed changes are exhaustive and the usefulness of their addition has to be clarified. For example, in desert regions, wind explains the most variance of resulting dust emission flux; the other parameters such as threshold friction velocity, soil moisture, clay content, etc., would explain less than 10% of the variance. Figure 1b also supports this point. Therefore, it is important to check which parameter contributes the most or identify in order the relative importance of each parameters for the arid and non-arid regions. I see that emission fluxes are compared on pages 26-28 but it would be useful to present these numbers in a table so that the readers can directly see the relative importance of each added term, separately for arid and nonarid regions. In that way, the model developers would be able to prioritize the inclusion of different terms depending on their relative importance based on the input data available. Also, present the total emissions resulting from the model for arid and nonarid regions (out of 29,300 Tg/year).*

We thank the reviewer for the suggestions. We do think some modifications are more important than others, and it is important to make this message as clear as possible. The submitted paper attempted to show the relative importance of the different modifications in Fig. 7, by showing the absolute change in emissions after each modification (left column) and the ratio of the normalized fluxes (i.e., rescaled to the global emission of 5000 Tg yr$^{-1}$) after each modification (right column). To make the effect of each modification even clearer, we have changed the left column to show the absolute difference after normalization (to 5000 Tg yr$^{-1}$) for each modification.
(Expt I: K14 scheme; Expt. II: changing $\overline{D}_p$ to 127 $\mu m$; Expt. III: multiplying $u_*$ by $F_{eff}$; Expt. V: using $u_{*it}$ in emission equations and multiplying $F_d$ by $\eta$.)

[Figure]

Subtracting the normalized emissions maps shows more clearly the impacts of each modification on changing the spatial patterns of the emissions. These maps are complementary to the relative change in the normalized emissions already shown in the right column of Fig. 7. The differences between Expt. II and Expt. I are small (of max.

magnitude of ~0.001 kg m$^{-2}$ yr$^{-1}$). The changes in (b) Expt. II and (c) Expt. III are much larger in magnitude (with the largest changes of ~0.1 kg m$^{-2}$ yr$^{-1}$). We quantified the total change caused by a modification by summing up all changes in magnitude by taking the absolute values, and defined this number as the "relative importance" as suggested by the reviewer. In panel a, out of 5000 Tg yr$^{-1}$ there is 250 Tg yr$^{-1}$ dust redistributed among regions. The drag partitioning (panel b) redistributed 3611 Tg yr$^{-1}$ dust, and the intermittency effect (panel c) redistributed 3163 Tg yr$^{-1}$ dust. In this sense, the drag partition effect and the intermittency effect are much more important than changing the soil diameter. Dust modelers can prioritize the inclusion of different terms, knowing that the intermittency and drag partition effect significantly change the dust emission behavior in models. We put these figures in Fig. 7 and described the relative importance of each modification in Sect. 4.1 after describing each modification. For instance, for the intermittency scheme (expt. V), we added at the end of the paragraph:

"… which is about a four-fold increase. Fig. 7h shows that C19 mainly increases marginal emissions; the overall effect of C19 is thus to move emissions from the hyperarid regions to semiarid regions. Summing up the absolute magnitude of changes in Fig. 7g, out of 5000 Tg yr$^{-1}$ there are 3163 Tg yr$^{-1}$ of dust redistributed within the source regions, indicating that the intermittency scheme induces a similar magnitude of changes compared to employing $F_{eff}$. Both the hybrid drag partition scheme and the intermittency scheme lead to > 60 % of dust emissions redistributed, showing that both effects modify the modeled emission behavior much more strongly compared to the effect of changing the value of $\overline{D}_p$ (experiment II)."

Please also refer to Sect. 4.1 for the relevant descriptions of other modifications.

Per the reviewer's suggestion, we added a table in the supplement (Table S2) that reports the total changes and separates the emissions by arid and nonarid regions. All modifications change dust emissions over arid regions more than nonarid regions, with changes over nonarid regions constituting ~10 % of the total redistributions. The drag partition effect tends to suppress emissions over less arid regions, hence shifting emissions toward hyperarid regions. The intermittency effect overall encourages emissions over semiarid and nonarid regions, but the shifts toward nonarid and high-latitude regions are not very high (87 Tg yr$^{-1}$).

|  | Using $\overline{D}_p = 127$ $\mu$m | Including $F_{eff}$ | Including the intermittency | Our scheme compared with K14 |
|---|---|---|---|---|
| Arid | 216 Tg yr$^{-1}$ | 3257 Tg yr$^{-1}$ | 3075 Tg yr$^{-1}$ | 3663 Tg yr$^{-1}$ |
| Nonarid | 34 Tg yr$^{-1}$ | 354 Tg yr$^{-1}$ | 87 Tg yr$^{-1}$ | 395 Tg yr$^{-1}$ |
| Globe | 250 Tg yr$^{-1}$ | 3611 Tg yr$^{-1}$ | 3163 Tg yr$^{-1}$ | 4058 Tg yr$^{-1}$ |

Per the reviewer's request, we also summarized in Table S3 the arid and nonarid dust emissions for K14, our scheme, Z03–Z, and Z03–G (given our input data in the main text Table 1), as well as MERRA-2's dust emissions:

|  | Original emissions | Normalized emissions | % of emissions from arid regions | % of emissions of nonarid regions |
|---|---|---|---|---|
| K14 | 29254 Tg yr$^{-1}$ | 5000 Tg yr$^{-1}$ | 92.1 % | 7.9 % |

| | | | | |
|---|---|---|---|---|
| Our scheme | 11494 Tg yr$^{-1}$ | 5000 Tg yr$^{-1}$ | 97.8 % | 2.2 % |
| MERRA-2 | 1561 Tg yr$^{-1}$ | 5000 Tg yr$^{-1}$ | 97.3 % | 2.7 % |
| Z03–Z | 424 Tg yr$^{-1}$ | 5000 Tg yr$^{-1}$ | 100 % | 0 % |
| Z03–G | 442 Tg yr$^{-1}$ | 5000 Tg yr$^{-1}$ | 100 % | 0 % |

*4. The paper is exhaustively long to read. Some sections are not very relevant to the main theme of the paper which is to describe the processes of dust emission parameterization proposed. For example, turbulence-driven intermittency (section 3.3) and scaling up of emission (section 4.2) is meant to match the GCM outputs with high-resolution estimates, so do not fall in the core theme of the paper. These two sections could probably form another paper so that this paper can better focus on emission parameterization and its validation alone.*

We thank the reviewer for the suggestion, as the reviewer is correct that the paper is longer than ideal. We have chosen to retain the intermittency scheme and the spatial rescaling in this paper, due to several reasons. First, is that the turbulence-driven intermittency scheme is also a process-based parameterization of a dust emission process, hence falling into the core theme of "dust emission processes" of this paper. Second, the spatial rescaling also falls within the main title of "scale-aware" dust scheme, and Sect. 4.2 itself is not enough content to stand as a separate scientific paper. Finally, leaving out the intermittency scheme will worsen our dust emission scheme's performance against DustCOMM emissions.

We agree with the reviewer that some of the descriptions are less relevant to the main theme of the paper. Combining with another reviewer's suggestion, we have shortened some of the longer sections such as Sect. 2 and greatly shortened Sect. 5 by deleting paragraphs or moving them to the supplement. We trimmed ~4000 words. Please see the track changes manuscript for the strikethroughs and the modified text, and the clean manuscript for the revised version of the main text.

***Detailed comments:***

*Title: scale-aware or scale-invariant instead of scale-respecting?*
*The title seems self-contradictory to the content because the paper itself compares the emissions at different resolutions which are very different (lines 165-195). So in what sense is the model scale-respecting? And why limit it only to GCMs, it should be applicable to RCMs as well. Perhaps a better title could be 'Towards a scale-invariant process-based dust emission scheme for climate models ……*

We thank the reviewer for the suggestion. The reason for using the term "scale-respecting" in the title is because we have derived the maps of scaling factor to correct the spatial variability of low-resolution emission simulations to match the high-resolution emission simulations. The use of the scaling maps then became part of our proposed new emission scheme. With the scaling adjustment, our scheme therefore moves toward a grid size-independent emission scheme. However, we agree with the reviewer that the term scale-respecting is not very clear. So, we followed the reviewer's suggestion and changed the title to:

"A new process-based and scale-aware desert dust emission scheme for global climate models – Part I: description and evaluation against inverse modeling emissions"

Also, in Sect. 4.2, we clarified the paragraph by adding more details at the end: "…These maps indicate that in coarse-gridded simulations, dust emissions are overall underrepresented over marginal sources and overrepresented over major sources. Therefore, we propose implementing these maps into GCMs of ~1° or coarser resolution to correct the dust emission spatial variability accordingly. Scaling all simulations across different spatial resolutions to the finest spatial resolution will move our dust emission scheme toward a scale-aware and grid-independent formulation."

*Line 43: I understand that you scaled up your high-resolution data so that it can be compared to GCM outputs. However, I don't see why scaling up high-resolution gridded dust emissions to the coarse resolution of GCMs would be so important so as to develop such an additional methodology, which also diverts the focus of the paper. The world is moving towards a high-resolution era and actually, the opposite would be more beneficial – to convert GCM outputs to high-resolution emissions.*

As pointed out in the previous comment, in Sect. 4 we correct low-resolution emissions to match the spatial variability of the high-resolution simulations. We thus indeed do the "opposite" – our proposed method converts coarse gridded simulations toward higher resolutions. We think the sentence line 43 could be a bit confusing, so we rephrased it as follows:
"We further propose (4) a simple methodology to rescale lower-resolution dust emission simulations to match the spatial variability of higher-resolution emission simulations in GCMs. The resulting dust emission simulation…"

*Line 51-52, this may not be entirely true, for example, Osipov et al. (2022) show that anthropogenic aerosols contribute more than 90% of PM:*
*https://www.nature.com/articles/s43247-022-00514-6*

Thanks for the good comment. Osipov et al. stated in the abstract: "In the Middle East, … hazardous fine particulate matter is to a large extent of anthropogenic origin (>90%), …". Osipov meant that anthropogenic aerosols contribute >90% of fine PM (defined as PM1 in their paper) in the Middle East. Therefore, their statement was not conflicting against lines 51–52 here, considering Kinne et al. (2006) and Kok et al. (2017) showed that desert dust globally contributes to >50 % of total aerosol mass particularly since desert dust dominates aerosol in the coarse mode.

*Line 55: Literature on dust-climate interaction should be expanded to include the most recent developments: e.g.,*
1. *Jin et al. (2021): Interactions of Asian mineral dust with Indian summer monsoon: Recent advances and challenges,*
   *https://www.sciencedirect.com/science/article/pii/S0012825221000611*
2. *Froyd et al. (2022): Dominant role of mineral dust in cirrus cloud formation revealed by global-scale measurements, https://www.nature.com/articles/s41561-022-00901-w*
3. *Parajuli et al. (2022): Effect of dust on rainfall over the Red Sea coast based on WRF-Chem model simulations, https://acp.copernicus.org/articles/22/8659/2022/*

We thank the reviewer for expanding our literature discussion. We expand the first paragraph of Sect. 1 by adding the mentioned papers as follows:

"For instance, dust changes Earth's radiative budget and atmospheric dynamics directly by scattering and absorbing radiation (Sokolik and Toon, 1996; Miller and Tegen, 1998) and indirectly by mediating cloud formation (Rosenfeld et al., 2001; Shi and Liu, 2019; McGraw et al., 2020; Froyd et al., 2022). These dust–radiation interactions and dust–cloud interactions also drives day-to-day variability in large-scale circulation patterns and local weathers such as monsoons and rainfall (Jin et al., 2021; Parajuli et al., 2022). Dust further impacts biogeochemistry by delivering nutrients such as iron and phosphorus to ocean and land ecosystems (Mahowald et al., 2010; Hamilton et al., 2020)."

*113-116: I am not sure about it because I think the main challenge is to represent small-scale roughness elements of grounds and rocks, not vegetation. The vegetation roughness effect is already taken into account in calculating friction velocity in most models in terms of displacement height and ground surface roughness, although crudely. So we need to be careful not double-accounting the effect of roughness.*

There are two major reasons for using a drag partition factor ($F_{eff}$) to represent the effect of both rocks and plants.

First, vegetation is included in the aerodynamic roughness, but that represents large-scale structures as explained by Menut et al. (2013) and will not account for the momentum absorption at the scale of concern for the aeolian processes. Large-scale vegetation properties only represent canopy roughness over densely covered vegetated areas but not sparse vegetation over arid regions. Our scheme and parameterization focus on drag partition effects due to aeolian roughness $z_{0a}$ of individual shrubs and sparse vegetation over the deserts, which could not be parameterized using large-scale aerodynamic roughness $z_{0m}$ of canopies. The large-scale $z_{0m}$ accounts for the large-scale roughness interacting with the winds and indeed slows down the wind. This roughness determines the wind stress exerted on the surface, summed over the entire grid box. Then the drag partition effect determines what fraction of the total wind is exerted onto the bare soil and drives dust emission. So, in that sense, there is no double-counting.

Second, many ESMs calculate global $z_{0m}$ by separating the globe into different regimes dominated by different types of surfaces. CLM5 distinguishes between vegetation, bare soil, snow, glacier, lake, and urban area in calculating $z_{0m}$ (Meier et al., 2021); most deserts are classified as bare soils in CLM and $z_{0m}$ there represents only soils, and vegetation is not at all represented over regimes other than vegetation (e.g., Lawrence et al., 2018; Meier et al., 2021). There is still a need to include further modifications to account for sparse vegetation over deserts.

Both reasonings support that we are not double counting the vegetation drag partition effect and indeed need to implement extra modifications in Sect. 3.2.

*122-123: that is true for GCMs but not for RCMs (e.g. WRF) which typically use model time steps in seconds.*

We focused this paper on GCMs and hoped we did not comment too much on RCMs. However, per the reviewer's question, it might be helpful to comment on our scheme's use in RCMs in general. Even regional climate models (RCMs) with a finer time step

(e.g., ~ 10 seconds) almost exclusively run in the Reynolds-averaged Navier–Stokes (RANS) mode and do not resolve eddies and turbulence. RCMs will only resolve turbulence in LES mode (i.e., WRF–LES). For RCMs in RANS mode, the issue of neglecting turbulence-driven intermittency exists no matter how long the time step the RCMs use. To comment on this issue, we added in the same paragraph in the introduction:

"… Even more importantly, wind turbulent fluctuations can sweep across the dust emission threshold multiple times and shut off dust emissions intermittently within one model time step, resulting in strong dust emission intermittency (Comola et al., 2019b). Even regional climate models (RCMs), which typically use a smaller time step (e.g., < 1 minute), do not resolve turbulence unless they are run in the computationally expensive large-eddy simulation (LES) mode (e.g., WRF–LES). Omitting turbulence by GCMs and RCMs thus causes either an overestimate or an underestimate of dust emissions, …"

*125: Are you talking about wind gusts that could be better represented in a sub-grid scale? Turbulence and subgrid variability of winds are two things. Wind gusts are probably better represented with the increase in resolution. Turbulence is usually understood differently in relation to large convective cells so it could be confusing.*

We are referring to turbulence-driven high-frequency wind fluctuations, not subgrid wind variability. To make this clearer, we modified the sentence here to:
"The third key piece of fundamental dust emission physics not accounted for by many models is the effect of turbulence-driven high-frequency wind variability on dust emissions. Most current GCMs assume a constant wind speed (and thus a constant emission flux) within the relatively large model time step, e.g., 30 minutes (e.g., Rahimi et al., 2019; Dunne et al., 2020). …"

*290-291: since Bit is constant u\*it is proportional to u\*ft0, I don't understand how it is going to change the simulated spatiotemporal distribution of dust as mentioned. u\*ft also includes soil moisture term fm in addition to u\*ft0 (which is only a function of Dp and air density) so u\*ft will show more variability, isn't it so?*

The reviewer is correct that $u_{*ft}$ includes effects of soil moisture, $D_p$, and air density, whereas $u_{*it}$ (and $u_{*ft0}$) contains only effects of $D_p$ and air density. Thus, $u_{*ft}$ and $u_{*it}$ have different spatial variability and $u_{*ft} > u_{*it}$. In a dust emission equation, switching the use of $u_{*ft}$ to $u_{*it}$ in the threshold term will substantially change the simulated spatiotemporal variability of dust emissions.

To clarify the sentence, we added in the sentence:
"… where $B_{it} = 0.82$ is approximately constant with soil properties and particle size (Bagnold, 1937; Kok et al., 2012). According to Eqs. 2–5, it can be seen that $u_{*ft} \geq u_{*ft0} > u_{*it}$ and that $u_{*ft}$ and $u_{*it}$ have different spatiotemporal variability. Also, the difference between $u_{*ft}$ and $u_{*it}$ could be much larger in nonarid regions because $f_m$ there can be much larger than one. In this study, we propose that dust emission models should use $u_{*it}$ instead of $u_{*ft}$ as the threshold term in the dust emission equations (e.g., Eq. 10 and Eq. 13), …"

*314-316: This is an interesting formulation but I am concerned that roughness could be double-accounted because roughness is already used in calculating u\* (law of the wall) in most GCM/RCM.*

We have responded to a similar concern of the reviewer's above for lines 113–116. Please refer to the above response.

*328: delete comma*

It is now deleted.

*384-385: Perhaps Chappel and Webb (2016) consider both since it uses satellite albedo, which has been recently implemented by Legrand et al. (2022).*
*Chappel and Webb (2016): https://gmd.copernicus.org/preprints/gmd-2022-157/*
*Legrand et al. (2022):*
*https://www.sciencedirect.com/science/article/pii/S1875963716300957?via%3Dihub*

Thank you for pointing this out. We have also noticed these two papers and indeed the measured albedo likely contains both rock and vegetation roughness, although the authors did not very explicitly discuss the relative importance of how albedo could capture vegetation versus rocks. We added a few sentences in the paragraph. In Line 377:

"It is thus very challenging to model vegetation drag partitioning using M&B95 by converting $\lambda$ to $z_{0a}$ when globally gridded $h$ (short vegetation height but not canopy height) is mostly unknown in GCMs or possesses strong subgrid variability. A more recent approach quantifies surface roughness by detecting the shadow (sheltered area) behind a roughness element using satellite-derived albedo (Chappell and Webb, 2016). This approach could potentially capture both rock and vegetation roughness and was also employed by later dust modeling studies (e.g., LeGrand et al., 2022). …"

*Eq 14: Using a constant Dp in arid regions is fine but it does not make much sense to use a variable Dp in vegetated areas which are not the dominant dust sources anyway. The question is: is adding such complexity worth that will likely not have any significant effect on the results of dust emission? In the vegetated areas, roughness/drag force will dominantly govern threshold friction velocity and the Dp will not likely have a remarkable effect on the resulting dust emission. Another concern is that fm already depends upon clay content so using clay content again in this formulation of Dp may double-account the effect of clay/silt content on dust emission.*

We agree with the reviewer that there is not much point to parameterize $\overline{D}_p$ over nonarid regions for modeling dust emissions, since dust emissions are zero over LAI>1 by definition anyway. In fact, in Sect. 4.1 we indeed proposed to set a global constant of $\overline{D}_{p0}$ = 127 $\mu$m in our simulations and for our dust emission scheme (line 1027) to simplify the complexity of the scheme. However, the relationship between $f_{silt+clay}$ and $\overline{D}_p$ over nonarid regions is still of scientific value especially for the fields of soil science, and we think it is still valuable to report the results in the main text.

To clarify, we added at the end of Sect. 3.1 that we suggest the use of a global constant in our final formulation of the scheme and simulations in Sect. 4.1, as follows:

"… Our derived $\overline{D}_p$ are largely consistent with the site $\overline{D}_p$ measurements from past studies (overlaid points), showing a similar spatial distribution, with a fit-line slope of 0.98 ($p$-value = 0.007), and an $R^2$ of 81% (Fig. 1d).

Since nonarid regions with LAI > 1 will generate zero emissions (Eq. 11), we simplify Eq. 14 and Fig. 1c by using a globally constant $\overline{D}_{p0} = 127 \, \mu$m.

For the reviewer's another concern, variables could affect dust emissions through multiple pathways. Increasing $f_{clay}$ content increases the dust emission threshold because it increases the size and weight of soil particles through the cohesion of microaggregates (Eq. 14). Meanwhile, increasing $f_{clay}$ also decreases the threshold by decreasing the moisture effect $f_m$, since water is adsorbed by clay particles and cannot create cohesive forces (Eq. 4). The two effects occur through different physical processes and have opposing effects to dust emissions. Therefore, this formulation of Eq. 14 is not double counting one single physical effect twice.

*Eq 15, again there is a possibility that zos is double-accounted as u\* already uses this term.*

We have responded to a similar concern of the reviewer's above for lines 113–116. Please refer to the response there.

*Eq 7, as far as I remember the equation is dp/30, not 2\*dp/30, e.g., see https://hal.archives-ouvertes.fr/hal-00677875/document*

The reviewer is right about the equation. Studies such as M&B95 and Laurent et al. (2006) used the same form of the equation. In this formulation, $\overline{D}_p$ represents the particle diameter of a monodisperse soil. The equation we presented with a factor of two was from (Sherman, 1992), representing a more classical sedimentology-based roughness formulation for mixed, polydisperse soils. Pierre et al. (2014) argued that including a factor of two is a better parameterization of soil roughness considering the whole undisturbed soil particle size distribution. This equation is thus adopted by more recent studies such as Pierre et al. (2014) and Klose et al. (2021). Many studies also proposed different relations, such as $D_p/10$ or $D_p/24$ (more of this is discussed in Pierre et al., 2014).

*Page 26 Line 38: You mentioned earlier that the K14 scheme is increasingly used in GCM/RCMs. The description of this model was already published and validated. Could you explain why this model still gives such a high estimate of dust emission flux (29,300 Tg/year)?*

We would like to clarify that, in GCM dust modeling, the current practice is that the global total dust emission is normalized such that the global mean dust AOD is approximately 0.03 (e.g., Klose et al., 2021; Li et al., 2022). A normalization or rescaling is needed because there are no a priori first principles to constrain the order of magnitude of the global total emission. It is normal for K14 or other schemes to have global total emission of higher or lower magnitudes, but it is usually fixed with a global tuning factor in GCMs. We plot in supplementary Fig. S8 all normalized emissions to a global total of 5000 Tg/yr, which should be the emissions to implement into GCMs and ESMs. We also clarify in Sect. 4.1:

"… The maps of ratios on the right panels thus show the changes in the dust emission spatial variability better than the left panels. Figure S6 shows the original, unnormalized emission maps for all experiments, and Fig. S7 shows the differences in unnormalized emissions between different experiments. Figure S8 shows the normalized emission maps for all experiments. …"

*313-315 Was Z03 also conducted at the same spatial resolution?*

Yes, Z03 was conducted at the same spatial resolution. To clarify, we added in the text (line 1335):
"We also conducted simulations using the Z03 scheme (Eq. 10) in 0.5° × 0.625° in comparison with our scheme's simulation. The Z03 scheme requires a source function S, …"

*313-315: Since this scheme uses an additional representation of drag partition, which is not in the Z03 scheme, wouldn't it make more sense to compare the results with another dust scheme that also contains a drag partition scheme, for example, Marticorena and Bergametti 1995 dust scheme which has been discussed in the paper:*
*https://agupubs.onlinelibrary.wiley.com/doi/epdf/10.1029/95JD00690*

We chose the Zender et al. (2003a) scheme (Z03) over the Marticorena et al. (1995) scheme (M&B95) due to the following considerations.

Z03 and M&B95 use the same dust emission equation (based on White, 1979) and threshold equation (based on Iversen and White, 1982). This is because Z03 adopted M&B95 in computing dust emissions. The differences between Z03 and M&B95 are mainly on i) the choice of source functions (since the original M&B95 does not use a source function), and ii) the size bin partitioning of the emissions (Z03 developed its own size bins different from M&B95). Meanwhile, both papers considered the drag partition effect, since Z03 took the idea from M&B95 (see Eq. 3 in Zender et al., 2003a). However, M&B95 did not state how gridded aeolian roughness could be obtained (which was accomplished by later studies such as Marticorena et al., 2006). Z03 simply assumed a globally constant drag partition factor, which means drag partitioning in Z03 is simply part of a global scaling factor. So, when ESMs nowadays state that they used Z03 versus M&B95, the only relevant difference is that Z03 employed a source function to further constrain global emission sources, or other differences in simulating transport. Therefore, it is safe to claim that M&B95 and Z03 are basically the same emission scheme.

Furthermore, in this paper, we wanted to show that our emission scheme does better than the schemes implemented in some of the current state-of-the-art ESMs, instead of better than the original M&B95/Z03. Although many current ESMs and CTMs employed M&B95 or Z03, they often left out the drag partition effect. Models such as HadGEM and UKESM employing M&B95 left out the drag partition effect and only considered the moisture effect on $u_{*ft}$ (e.g., Woodward, 2001; Fiedler et al., 2016; Woodward et al., 2022). Models employing Z03 such as CESM, E3SM, CMAQ, and GEOS-Chem, also dropped the drag partition effect (e.g., Wang et al., 2012; Oleson et al., 2013; Kok et al., 2014; Meng et al., 2021). We built the modified schemes as in current ESMs instead of strictly following the original M&B95/Z03, in order to show the improvements our scheme can achieve in these ESMs.

We mentioned Z03 instead of M&B95 in this study because we are further evaluating our scheme in CESM against its Z03 scheme in the companion paper.

Another reason for ESMs not using the drag partition effect in M&B95 is because Marticorena's later studies mainly focused on regional modeling, with only drag partition maps of specific regions such as Africa and Asia available (e.g., Laurent et al. 2006, 2008; Darmenova et al., 2009).

*Section 5: The limitation section is too detailed, which somehow undermines the value of the study itself. This section could be shortened to highlight only the key limitations associated with estimates of dust emission fluxes.*

We thank the reviewer for the suggestion. Per the reviewer's suggestion, we chose one major assumption/limitation for each modification and moved the rest into the supplement (Sect. S6). We kept the limitations associated with the estimates of dust emission fluxes. We shortened Section 5 by more than 2000 words. Please see the tracked changes manuscript for the strikethroughs to see how we shortened the article, and see the clean manuscript for the final version of the main text.

Response to Referee #2

*This paper concerns the improvement of the representation of dust emission processes in current global climate models and land surface models. The paper is novel, interesting, overall well written. I have some suggestions however, which might be helpful for improving the readability of the paper. Overall I share some similar thoughts as those from Anonymous Referee #1 as regards the paper's length. In particular, the length is not totally justified as some sections are really too long (more appropriate for a book/report/PhD thesis rather than for a research paper). I would suggest improving the readability and the novelty by focusing more on the validation of the proposed scheme(s), showing its performance against that obtained by other schemes available in the literature: as the RMSE is of the order of Tg/year, which might be considered high, how about adding also a normalized error (or percentage)? In summary, I would suggest focusing more on the results and their discussion trying to shorten a bit the considerations made to derive the scheme. This would make the reading much clearer and the novelty carried out might emerge more clearly.*

Thank you for the reviewer's suggestions. Combined with another reviewer's comments, we deleted ~4000 words from the main text and moved some of the text to the supplement. We also added more discussions to the proposed schemes and simulation results (see revised text and our response). We also changed the RMSE statistics in Fig. 10 to NRMSE. As the reviewer suggested, we present the NRMSE in percentage, by dividing the RMSE values by the mean of DustCOMM emissions (5000 Tg yr$^{-1}$ / 10 data points = 500 Tg yr$^{-1}$). For instance, in talking about Fig. 10b where we first mention RMSE and NRMSE, we added:

"… Our scheme has an $R^2$ of 89 % and a root-mean-squared error (RMSE) of 141 Tg yr$^{-1}$. We note in Fig. 10b the normalized RMSE (NRMSE) of 28 %, which is the RMSE divided by the mean of DustCOMM emissions (5000 Tg yr$^{-1}$ / 10 data points = 500 Tg yr$^{-1}$). Our scheme's performance is …"

*Specific comments:*

*Lines 53-56: This is true in general for all aerosols, and mineral dust does not make an exception.*

We agree with the reviewer. Along with another reviewer's suggestion, this paragraph is now modified to describe more impacts of dust:
"Desert dust accounts for more than half of the atmospheric mass loading of particulate matter (PM) (Kinne et al., 2006; Kok et al., 2017) and produces a wide range of important impacts on multiple components of the Earth system (Shao et al., 2011). Like other aerosols, dust changes Earth's radiative budget … by mediating cloud formation (Rosenfeld et al., 2001; Shi and Liu, 2019; McGraw et al., 2020; Froyd et al., 2022). These dust–radiation interactions and dust–cloud interactions also drive day-to-day variability in large-scale circulation patterns and local weathers such as monsoons and rainfall (Jin et al., 2021; Parajuli et al., 2022). Dust further impacts biogeochemistry by delivering nutrients such as iron and phosphorus to ocean and land ecosystems (Mahowald et al., 2010; Hamilton et al., 2020)."

*Lines 83-85: The listed properties are not only those from soil but also from the atmosphere and aerosol.*

We agree with the reviewer. We thus correct our sentence by adding the word atmospheric conditions, to:

"The dust emission threshold is a function of soil properties and atmospheric conditions like particle size distribution, soil moisture, and air density."

*Line 88: Which kind of soil properties and how do they affect the dust emission threshold?*

The dust emission threshold could be a function of soil properties such as soil texture, soil organic carbon, and pH value since these properties will change the cohesion between soil particles. We added in the details in line 88:

"First, many models assume a globally constant soil particle size in calculating a spatially varying dust emission threshold (Zender et al., 2003; Darmenova et al., 2009; Kok et al., 2014), whereas the actual soil particle size is likely a function of space and time and could depend on soil properties, such as texture, pH, and organic matter content since these variables modulate the cohesions between soil particles (Webb et al., 2016)."

*Line 172: This detail about R is probably not needed at this point of the work.*

We removed the sentence here as follows:

"In Sect. 4, we code the new dust emission scheme as a standalone sandbox model (see Sect. 2.4)  and examine the resulting spatiotemporal variability of the new dust emissions."

*Line 243, 245, 247, …: "moment" would be "momentum"?*

We meant to use here the word "moment" balance, in which moment means the forces that tend to cause a body to rotate about a certain point of axis.

*Lines 180-462: These subsections are quite long and descriptive. I would suggest to do some efforts in summarizing the description in these subsections.*

We thank the reviewer for the suggestion. We have shortened Sect. 2.1 by removing the background knowledge on saltation and moment balance. We also shortened the discussion on the contention between the "representative" approach versus the "independent" approach of saltation. We further shortened Sect. 2.2 by moving the discussion on other approaches on combining rock and vegetation drag partitioning into the supplementary text (Sect. S2). Because we do not list the trimmed numerous paragraphs here, but please see the tracked changes manuscript to see the relevant parts with strikethrough. Please see the final manuscript to see the shortened and finalized main text. We also shortened other sections in the main text, and the whole main text was shortened by ~4000 words.

*Lines 465-480: This subsection entitled "Input data and model description" is not appropriate as a subsection of Section 2 "Current dust emission schemes in climate models". Revise the structure.*

We thank the reviewer for correcting this mistake. We have changed the title of Sect. 2 to encompass Sect. 2.4:

"2 Current dust emission schemes and their input variables"

Sect. 2.4 is also renamed to link better to the previous subsections:
"2.4 Inputs required by the dust emission schemes"

*Figure 1 d: There is consistent spread for arid soils, which is masked by the goodness of the fit driven for nonarid soils. This might indicate reduced agreement for arid soils which are the more relevant for this study. Indeed, the fit-line slope is essentially good, but this might depend on the fact that the regression is driven by the non arid values.*

We agree with the reviewer's observation, and we hope we expressed the same idea and did not mask this message by stating that we used a constant of $\overline{D}_{p0}$ of 127 $\mu$m as our predictions in Fig. 1d. In this figure, we intended to evaluate Eq. 14 and make the point that using a constant for arid regions gave satisfactory agreement in predicting global $\overline{D}_p$ as such. A good agreement in predicting global $\overline{D}_p$ as such does not mean good performance in dust emission modeling using the predicted $\overline{D}_p$. We also pointed out in a few places, such as in Sect. 5, that it is indeed suboptimal to use a constant for $\overline{D}_p$ across all arid regions given insufficient soil PSD observations.

To make this message even clearer, we add a few more comments toward the end of the paragraph:
"… Following Eq. 14, the arid and semi-arid regions are set to have a $\overline{D}_{p0}$ of 127 $\mu$m, whereas for nonarid regions $\overline{D}_p$ increases with $f_{silt+clay}$. Our derived $\overline{D}_p$ are largely consistent with the site $\overline{D}_p$ measurements from past studies (overlaid points), showing a similar spatial distribution, with a fit-line slope of 0.98 (*p*-value = 0.007), and an $R^2$ of 81% (Fig. 1d). Note that, since the predictions for arid regions (red points) are a constant, the agreement between the predictions and observations are due to the linear $\overline{D}_p - f_{silt+clay}$ relation over nonarid regions (blue points). Fig. 1d shows that Eq. 14 gives satisfactory agreement in predicting global $\overline{D}_p$, but dust emission modeling will depend exclusively on the predicted $\overline{D}_p$ over arid regions. We anticipate that as more measurements emerge in the future, more statistical or machine learning modeling approaches can more robustly decipher the intricate relationships between $\overline{D}_p$ and various soil properties over arid regions."

*Lines 003-005: Please provide justifications for such statement (one year used as a climatological dataset?); also, please provide further details on "other input data".*

Here we should clarify that we are referring to all input data listed in Table 1. Apart from MERRA-2 and ESA-CCI datasets which contain 2006 data, other variables in Table 1, including SoilGrids, Prigent's aeolian roughness, and the source functions for Zender's emission scheme are 2-D spatial maps without time resolutions. SoilGrids soil texture was derived by using soil profiles in the recent decades; Prigent's roughness data used 1997 satellite retrievals; Zender's source function was obtained by averaging satellite retrievals across 1980–2001 (Zender et al., 2003); and the year of data that Ginoux's source function was obtained was not clear from Ginoux et al. (2001). These datasets were regarded as slowly time-varying variables and have been used for present-day and historical simulations for other years. We modify the paragraph to clarify the sentence and delete the word "climatological datasets":

"In this section, we implement the three new parameterizations of key dust emissions processes with the K14 model into R to investigate the resulting spatial variability of dust emissions. The MERRA-2 data and ESA CCI land cover data are for the year 2006. Other inputs in Table 1 (SoilGrids data, Prigent's roughness, and source functions) are 2-D spatial datasets with no 2006 data available, but they are slowly time-varying variables and are generally used for present-day simulations in other years. In Sect. 4.1, …"

*Line 224: Change "spatiaotemporal" to "spatio".*

It is corrected as suggested.

*Line 238: Change "infinity" to "infinite".*

We think the use of the noun infinity is more correct than the use of the adjective infinite. So, we would like to keep it as is.

*Lines 342-359: Perhaps this comparative analysis of performance would be better if provided in a table and discussed in the main text, providing reasons. It would be also nice to provide percentage values for comparison: for instance, as from figure 10 b it might be observed that the differences are more relevant in some regions, which might correspond or not to regions of significant dust emissions. As such, the introduction of a normalized or a percentage value could be helpful.*

To make the dust contributions from each region clear, we made a table specifying regional dust emissions simulated by different dust models and DustCOMM. We tried to discuss some major differences between emissions from K14 and our scheme in the main text, as the reviewer also suggested in the next comment in Figure 10. However, due to the size of the table and the many numbers presented in the table, we think it is more appropriate to put the table in the supplement. We put a sentence to direct interested readers to the supplementary Table S4.

"… We normalized their estimate to match our global total of 5000 Tg yr$^{-1}$, yielding a high-latitude emission range of 200–250 Tg yr$^{-1}$. We take the mean value, which is 225 Tg yr$^{-1}$. For all schemes and datasets discussed here, Table S4 provides a list of regional emission contributions to the global total emission."

Table S4. Regional contributions of dust emissions to the global total emission for different schemes. [a, b]

| | DustCOMM & B16[c] | K14 | Our scheme | Z03–Z | Z03–G | MERRA-2 |
|---|---|---|---|---|---|---|
| NW Africa | 18 % | 10.4 % | 14.4 % | 7.5 % | 23.2 % | 20.1 % |
| NE Africa | 16 % | 17.5 % | 14.5 % | 13.3 % | 17.9 % | 17.1 % |
| Sahel | 13 % | 17.4 % | 15.9 % | 20.8 % | 16.5 % | 21.5 % |
| Middle East / C Asia | 29 % | 29.4 % | 30.6 % | 32.5 % | 31.3 % | 29.2 % |
| E Asia | 13 % | 4.1 % | 11.9 % | 14.3 % | 4.0 % | 6.5 % |
| N America | 3 % | 1.8 % | 1.1 % | 0.1 % | 0.02 % | 0.5 % |
| Australia | 3 % | 10.8 % | 6.4 % | 9.7 % | 6.8 % | 2.6 % |
| S America | 4 % | 2.3 % | 2.2 % | 0.5 % | 0.07 % | 1.7 % |
| S Africa | 2 % | 6.3 % | 3.0 % | 1.4 % | 0.3 % | 0.7 % |

| high-lat (w/ Patagonia) | 5 % (from B16) | 2.8 % | 6.3 % | 0.2 % | 0.2 % | 1.7 % |
|---|---|---|---|---|---|---|

[a]All percentages from MERRA-2 and our simulations are rounded to 1 decimal place, except for smaller values where we rounded to 2 decimal places.

[b]Bullard et al. (2016) obtained 5 % including Patagonia emissions, which overlaps with the S. America domain defined in Kok et al. (2021a, b). We present the percentage here assuming the nine K21 source regions sum up to be 100 %, since K21 (DustCOMM) predicted zero emissions outside of the domains (including B16 will yield 105 %). We arrange the other columns the same way such that the percentages from the nine K21 regions sum up to 100 % also.

[c]Values are directly obtained from Table 2 of Kok et al. (2021b), which are rounded up to integers, except for high-latitude emissions that are obtained from Bullard et al. (2016).

*Figure 10: in figure 10 b it could be discussed better the reasons of the differences, more relevant in some regions than in ot*

Per the reviewer's suggestion, we added some discussions on the differences between K14 and our scheme for all source regions in Fig. 10b.

"Figure 10b shows that our scheme's emissions are in overall better agreement with DustCOMM emissions than the K14 scheme. Some of the most significant differences in emissions between our scheme and K14 are over regions including Australia, North America, and South Africa, where the vegetation drag partitioning causes strong reductions in winds and emission fluxes (Fig. 7f) from K14 to our scheme. Our scheme's East Asian emission is significantly higher than K14's (also shown in Fig. 7j), primarily due to the switch from using $u_{*ft}$ to $u_{*it}$ in the dust emission equation (Fig. 7h and Fig. S8d). Emissions over South America, the Middle East, and the three regions of North Africa have relatively small and negligible differences between K14 and our scheme. This occurs because both the drag partitioning and intermittency effects create only minimal changes to the emissions over these regions (Fig. 7j). From Fig. 10b, the results with our scheme (blue color) better match the DustCOMM regional emissions than results with the K14 scheme, lying substantially closer to the 1:1 (black) line over most regions including Africa, Asia, and Australia. There are two notable exceptions where our scheme has less agreement than K14 with DustCOMM, namely North America and the high-latitude emissions. Our scheme generates fewer dust emissions over the Mojave–Sonoran–Chihuahuan deserts over the U.S.–Mexico border compared to the K14 emissions (Fig. 7a), …"